# PINNACLE: A COMPREHENSIVE BENCHMARK OF PHYSICS-INFORMED NEURAL NETWORKS FOR SOLVING PDEs

## ABSTRACT

While significant progress has been made on Physics-Informed Neural Networks (PINNs), a comprehensive comparison of these methods across a wide range of Partial Differential Equations (PDEs) is still lacking. This study introduces PIN-Nacle, a benchmarking tool designed to fill this gap. PINNacle provides a diverse dataset, comprising over 20 distinct PDEs from various domains, including heat conduction, fluid dynamics, biology, and electromagnetics. These PDEs encapsulate key challenges inherent to real-world problems, such as complex geometry, multi-scale phenomena, nonlinearity, and high dimensionality. PINNacle also offers a user-friendly toolbox, incorporating about 10 state-of-the-art PINN methods for systematic evaluation and comparison. We have conducted extensive experiments with these methods, offering insights into their strengths and weaknesses. In addition to providing a standardized means of assessing performance, PINNacle also offers an in-depth analysis to guide future research, particularly in areas such as domain decomposition methods and loss reweighting for handling multi-scale problems and complex geometry. To the best of our knowledge, it is the largest benchmark with a diverse and comprehensive evaluation that will undoubtedly foster further research in PINNs.

## 1 INTRODUCTION

Partial Differential Equations (PDEs) are of paramount importance in science and engineering, as they often underpin our understanding of intricate physical systems such as fluid flow, heat transfer, and stress distribution (Morton & Mayers, 2005). The computational simulation of PDE systems has been a focal point of research for an extensive period, leading to the development of numerical methods such as finite difference (Causon & Mingham, 2010), finite element (Reddy, 2019), and finite volume methods (Eymard et al., 2000).

Recent advancements have led to the use of deep neural networks to solve forward and inverse problems involving PDEs (Raissi et al., 2019; Yu et al., 2018; Daw et al., 2017; Wang et al., 2020). Among these, Physics-Informed Neural Networks (PINNs) have emerged as a promising alternative to traditional numerical methods in solving such problems (Raissi et al., 2019; Karniadakis et al., 2021). PINNs leverage the underlying physical laws and available data to effectively handle various scientific and engineering applications. The growing interest in this field has spurred the development of numerous PINN variants, each tailored to overcome specific challenges or to enhance the performance of the original framework.

While PINN methods have achieved remarkable progress, a comprehensive comparison of these methods across diverse types of PDEs is currently lacking. Establishing such a benchmark is crucial as it could enable researchers to more thoroughly understand existing methods and pinpoint potential challenges. Despite the availability of several studies comparing sampling methods (Wu et al., 2023) and reweighting methods (Bischof & Kraus, 2021), there has been no concerted effort to develop a rigorous benchmark using challenging datasets from real-world problems. The sheer variety and inherent complexity of PDEs make it difficult to conduct a comprehensive analysis. Moreover, different mathematical properties and application scenarios further complicate the task, requiring the benchmark to be adaptable and exhaustive.

To resolve these challenges, we propose PINNacle, a comprehensive benchmark for evaluating and understanding the performance of PINNs. As shown in Fig. 1, PINNacle consists of three major

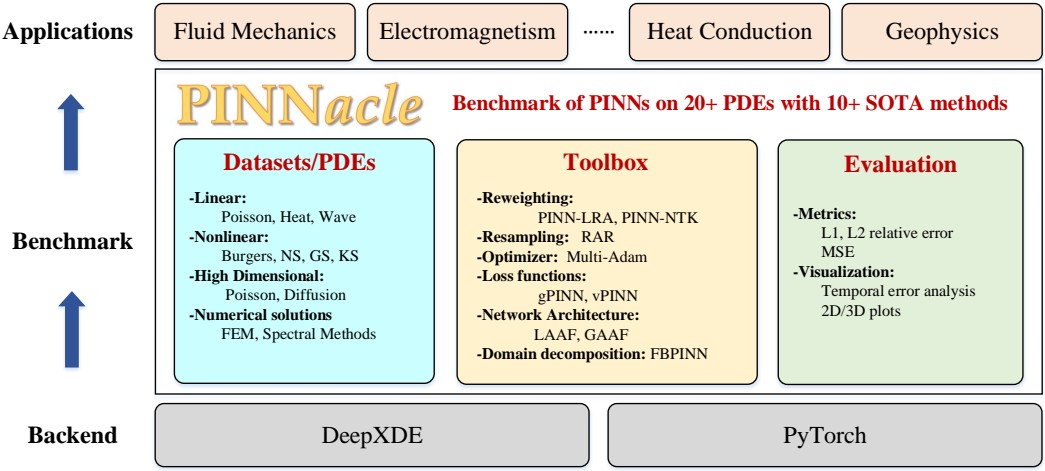

Figure 1: Architecture of PINNacle. It contains a dataset covering more than 20 PDEs, a toolbox that implements about 10 SOTA methods, and an evaluation module. These methods have a wide range of application scenarios like fluid mechanics, electromagnetism, heat conduction, geophysics, and so on.

components — a diverse dataset, a toolbox, and evaluation modules. The dataset comprises tasks from over 20 different PDEs from various domains, including heat conduction, fluid dynamics, biology, and electromagnetics. Each task brings its own set of challenges, such as complex geometry, multi-scale phenomena, nonlinearity, and high dimensionality, thus providing a rich testing ground for PINNs. The toolbox incorporates more than 10 state-of-the-art (SOTA) PINN methods, enabling a systematic comparison of different strategies, including loss reweighting, variational formulation, adaptive activations, and domain decomposition. These methods can be flexibly applied to the tasks in the dataset, offering researchers a convenient way to evaluate the performance of PINNs which is also user-friendly for secondary development. The evaluation modules provide a standardized means of assessing the performance of different PINN methods across all tasks, ensuring consistency in comparison and facilitating the identification of strengths and weaknesses in various methods.

PINNacle provides a robust, diverse, and comprehensive benchmark suite for PINNs, contributing significantly to the field's understanding and application. It represents a major step forward in the evolution of PINNs which could foster more innovative research and development in this exciting field. In a nutshell, our contributions can be summarized as follows:

- We design a dataset encompassing over 20 challenging PDE problems. These problems encapsulate several critical challenges faced by PINNs, including handling complex geometries, multi-scale phenomena, nonlinearity, and high-dimensional problems.

- We systematically evaluate more than 10 carefully selected representative variants of PINNs. We conducted thorough experiments and ablation studies to evaluate their performance. To the best of our knowledge, this is the largest benchmark comparing different PINN variants.

- We provide an in-depth analysis to guide future research. We show using loss reweighting and domain decomposition methods could improve the performance on multi-scale and complex geometry problems. Variational formulation achieves better performance on inverse problems. However, few methods can adequately address nonlinear problems, indicating a future direction for exploration and advancement.

## 2 RELATED WORK

### 2.1 BENCHMARKS AND DATASETS IN SCIENTIFIC MACHINE LEARNING

The growing trend of AI applications in scientific research has stimulated the development of various benchmarks and datasets, which differ greatly in data formats, sizes, and governing principles. For instance, (Lu et al., 2022) presents a benchmark for comparing neural operators, while (Botev et al., 2021; Otness et al., 2021) benchmarks methods for learning latent Newtonian mechanics.

Furthermore, domain-specific datasets and benchmarks exist in fluid mechanics (Huang et al., 2021), climate science (Racah et al., 2017; Cachay et al., 2021), quantum chemistry (Axelrod & Gomez-Bombarelli, 2022), and biology (Boutet et al., 2016).

Beyond these domain-specific datasets and benchmarks, physics-informed machine learning has received considerable attention (Hao et al., 2022; Cuomo et al., 2022) since the advent of Physics-Informed Neural Networks (PINNs) (Raissi et al., 2019). These methods successfully incorporate physical laws into model training, demonstrating immense potential across a variety of scientific and engineering domains. Various papers have compared different components within the PINN framework; for instance, (Das & Tesfamariam, 2022) and (Wu et al., 2023) investigate the sampling methods of collocation points in PINNs, and (Bischof & Kraus, 2021) compare reweighting techniques for different loss components. PDEBench (Takamoto et al., 2022) and PDEArena (Gupta & Brandstetter, 2022) design multiple tasks to compare different methods in scientific machine learning such as PINNs, FNO, and U-Net. Nevertheless, a comprehensive comparison of various PINN approaches remains absent in the literature.

## 2.2 SOFTWARES AND TOOLBOXES

A plethora of software solutions have been developed for solving PDEs with neural networks. These include SimNet (Hennigh et al., 2021), NeuralPDE (Rackauckas & Nie, 2017), TorchDiffEq (Chen et al., 2020), and PyDEns (Khudorozhkov et al., 2019). More recently, DeepXDE (Lu et al., 2021) has been introduced as a fundamental library for implementing PINNs across different backends. However, there remains a void for a toolbox that provides a unified implementation for advanced PINN variants. Our PINNacle fills this gap by offering a flexible interface that facilitates the implementation and evaluation of diverse PINN variants. We furnish clear and concise code for researchers to execute benchmarks across all problems and methods.

## 2.3 VARIANTS OF PHYSICS-INFORMED NEURAL NETWORKS

The PINNs have received much attention due to their remarkable performance in solving both forward and inverse PDE problems. However, vanilla PINNs have many limitations. Researchers have proposed numerous PINN variants to address challenges associated with high-dimensionality, non-linearity, multi-scale issues, and complex geometries (Hao et al., 2022; Cuomo et al., 2022; Karniadakis et al., 2021; Krishnapriyan et al., 2021). Broadly speaking, these variants can be categorized into: loss reweighting/resampling (Wang et al., 2021a;b; Tang et al., 2021; Wu et al., 2023; Nabian et al., 2021), innovative optimizers (Yao et al., 2023), novel loss functions such as variational formulations (Yu et al., 2018; Kharazmi et al., 2019; 2021; Khodayi-Mehr & Zavlanos, 2020) or regularization terms (Yu et al., 2022; Son et al., 2021), and novel architectures like domain decomposition (Jagtap & Karniadakis, 2021; Li et al., 2019; Moseley et al., 2021; Jagtap et al., 2020c) and adaptive activations (Jagtap et al., 2020b;a). These variants have enhanced PINN's performance across various problems. Here we select representative methods from each category and conduct a comprehensive analysis using our benchmark dataset to evaluate these variants.

## 3 PINNACLE: A HIERARCHICAL BENCHMARK FOR PINNS

In this section, we first introduce the preliminaries of PINNs. Then we introduce the details of datasets (tasks), PINN methods, the toolbox framework, and the evaluation metrics.

## 3.1 PRELIMINARIES OF PHYSICS-INFORMED NEURAL NETWORKS

Physics-informed neural networks are neural network-based methods for solving PDEs as well as inverse problems of PDEs, which have received much attention recently. Specifically, let's consider a general Partial Differential Equation (PDE) system defined on $\Omega$, which can be represented as:

$$\mathcal{F}(u(x); x) = 0, \quad x \in \Omega, \tag{1}$$
$$\mathcal{B}(u(x); x) = 0, \quad x \in \partial\Omega. \tag{2}$$

where $\mathcal{F}$ is a differential operator and $\mathcal{B}$ is the boundary/initial condition. PINN uses a neural network $u_\theta(x)$ with parameters $\theta$ to approximate $u(x)$. The objective of PINN is to minimize the following

loss function:

$$\mathcal{L}(\theta) = \frac{w_c}{N_c} \sum_{i=1}^{N_c} ||\mathcal{F}(u_\theta(x_c^i); x_c^i)||^2 + \frac{w_b}{N_b} \sum_{i=1}^{N_b} ||\mathcal{B}(u_\theta(x_b^i); x_b^i)||^2 + \frac{w_d}{N_d} \sum_{i=1}^{N_d} ||u_\theta(x_d^i) - u(x_d^i)||^2. \quad (3)$$

where $w_c, w_b, w_d$ are weights. The first two terms enforce the PDE constraints on $\{x_c^i\}_{1...N_c}$ and boundary conditions on $\{x_b^i\}_{1...N_b}$. The last term is data loss, which is optional when there is data available. However, PINNs have several inherent drawbacks. First, PINNs optimize a mixture of imbalance loss terms which might hinder its convergence as illustrated in (Wang et al., 2021a). Second, nonlinear or stiff PDEs might lead to unstable optimization (Wang et al., 2021b). Third, the vanilla MLPs might have difficulty in representing multi-scale or high-dimensional functions. For example, (Krishnapriyan et al., 2021) shows that vanilla PINNs only work for a small parameter range, even in a simple convection problem. To resolve these challenges, numerous variants of PINNs are proposed. However, a comprehensive comparison of these methods is lacking, and thus it is imperative to develop a benchmark.

## 3.2 DATASETS

To effectively compare PINN variants, we've curated a set of PDE problems (datasets) representing a wide range of challenges. We chose PDEs from diverse domains, reflecting their importance in science and engineering. Our dataset includes 22 unique cases, with further details in Appendix B.

- The **Burgers' Equation**, fundamental to fluid mechanics, considering both one and two-dimensional problems.
- The **Poisson's Equation**, widely used in math and physics, with four different cases.
- The **Heat Equation**, a time-dependent PDE that describes diffusion or heat conduction, demonstrated in four unique cases.
- The **Navier-Stokes Equation**, describing the motion of viscous fluid substances, showcased in three scenarios: a lid-driven flow (NS2d-C), a geometrically complex backward step flow (NS2d-CG), and a time-dependent problem (NS2d-LT).
- The **Wave Equation**, modeling wave behavior, exhibited in three cases.
- **Chaotic PDEs**, featuring two popular examples: the Gray-Scott (GS) and Kuramoto-Sivashinsky (KS) equations.
- **High Dimensional PDEs**, including the high-dimensional Poisson equation (PNd) and the high-dimensional diffusion or heat equation (HNd).
- **Inverse Problems**, focusing on the reconstruction of the coefficient field from noisy data for the Poisson equation (PInv) and the diffusion equation (HInv).

It is important to note that we have chosen PDEs encompassing a wide range of mathematical properties. This ensures that the benchmarks do not favor a specific type of PDE. The selected PDE problems introduce several core challenges, which include:

- **Complex Geometry**: Many PDE problems involve complex or irregular geometry, such as heat conduction or wave propagation around obstacles. These complexities pose significant challenges for PINNs in terms of accurate boundary behavior representation.
- **Multi-Scale Phenomena**: Multi-scale phenomena, where the solution varies significantly over different scales, are prevalent in situations such as turbulent fluid flow. Achieving a balanced representation across all scales is a challenge for PINNs in multi-scale scenarios.
- **Nonlinear Behavior**: Many PDEs exhibit nonlinear or even chaotic behavior, where minor variations in initial conditions can lead to substantial divergence in outcomes. The optimization of PINNs becomes intriguing on nonlinear PDEs.
- **High Dimensionality**: High-dimensional PDE problems, frequently encountered in quantum mechanics, present significant challenges for PINNs due to the "curse of dimensionality". This term refers to the increase in computational complexity with the addition of each dimension, accompanied by statistical issues like data sparsity in high-dimensional space.

These challenges are selected due to their frequent occurrence in numerous real-world applications. As such, a method's performance in addressing these challenges serves as a reliable indicator of its overall practical utility. Table 1 presents a detailed overview of the dataset, the PDEs, and the challenges associated with these problems. We generate data using FEM solver provided by COMSOL 6.0 (Multiphysics, 1998) for problems with complex geometry and spectral method provided by Chebfun (Driscoll et al., 2014) for chaotic problems. More details can be found in Appendix B.

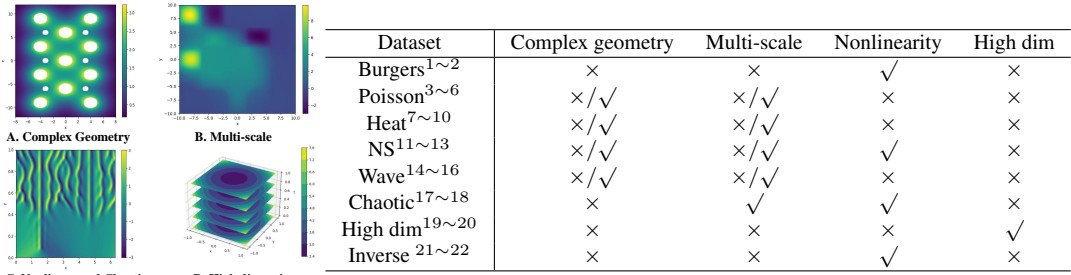

| Dataset | Complex geometry | Multi-scale | Nonlinearity | High dim |
|---|---|---|---|---|
| Burgers[1~2] | $\times$ | $\times$ | $\checkmark$ | $\times$ |
| Poisson[3~6] | $\times/\checkmark$ | $\times/\checkmark$ | $\times$ | $\times$ |
| Heat[7~10] | $\times/\checkmark$ | $\times/\checkmark$ | $\times$ | $\times$ |
| NS[11~13] | $\times/\checkmark$ | $\times/\checkmark$ | $\checkmark$ | $\times$ |
| Wave[14~16] | $\times/\checkmark$ | $\times/\checkmark$ | $\times$ | $\times$ |
| Chaotic[17~18] | $\times$ | $\checkmark$ | $\checkmark$ | $\times$ |
| High dim[19~20] | $\times$ | $\times$ | $\times$ | $\checkmark$ |
| Inverse[21~22] | $\times$ | $\times$ | $\checkmark$ | $\times$ |

Table 1: Overview of our datasets along with their challenges. We chose 22 cases in total to evaluate the methods of PINNs. The left picture shows the visualization of cases with these four challenges, i.e., complex geometry, multi-scale, nonlinearity, and high dimension.

## 3.3 METHODS AND TOOLBOX

After conducting an extensive literature review, we present an overview of diverse PINNs approaches for comparison. Then we present the high-level structure of our PINNacle.

### 3.3.1 METHODS

As mentioned above, variants of PINNs are mainly based on loss functions, architecture, and optimizer (Hao et al., 2022). The modifications to loss functions can be divided into reweighting existing losses and developing novel loss functions like regularization and variational formulation. Variants of architectures include using domain decomposition and adaptive activations.

The methods discussed are directly correlated with the challenges highlighted in Table 1. For example, domain decomposition methods are particularly effective for problems involving complex geometries and multi-scale phenomena. Meanwhile, loss reweighting strategies are adept at addressing imbalances in problems with multiple losses. We have chosen variants from these categories based on their significant contributions to the field. Here, we list the primary categories and representative methods as summarized in Table 2:

- **Loss reweighting/Resampling (2~4)**: PINNs are trained with a mixed loss of PDE residuals, boundary conditions, and available data losses shown in Eq 3. Various methods (Wang et al., 2021a; 2022b; Bischof & Kraus, 2021; Maddu et al., 2022; Rohrhofer et al., 2021) propose different strategies to adjust these weights $w_c$, $w_b$ and $w_d$ at different epochs or resample collocation points $\{x_c^i\}$ and $\{x_b^i\}$ in Eq 3, which indirectly adjust the weights (Wu et al., 2023; Nabian et al., 2021). We choose three famous examples, i.e., reweighting using gradient norms (PINN-LRA) (Wang et al., 2021a), using neural tangent kernel (PINN-NTK) (Wang et al., 2022b), and residual-based resampling (RAR)(Lu et al., 2021; Wu et al., 2023).

- **Novel optimizer (5)**: To handle the problem of multi-scale objectives, some new optimizers (Liu et al., 2022; Yao et al., 2023) are proposed. We chose MultiAdam, which is resistant to domain scale changes.

- **Novel loss functions (6~7)**: Some works introduce novel loss functions like variational formulation (Sirignano & Spiliopoulos, 2018; Khodayi-Mehr & Zavlanos, 2020; Kharazmi et al., 2021) and regularization terms to improve training. We choose hp-VPINN (Kharazmi et al., 2021) and gPINN (Yu et al., 2022; Son et al., 2021), which are representative examples from these two categories.

- **Novel activation architectures (8~10)**: Some works propose various network architectures, such as using CNN and LSTM (Zhang et al., 2020; Gao et al., 2021; Ren et al., 2022),

|  | Complex Geometry | Multi-scale | Nonlinearity | High dim |
|---|:---:|:---:|:---:|:---:|
| Vanilla PINN [1] | × | × | × | × |
| Reweighting/Resampling[2~4] | √ | √ | × | × |
| Novel Optimizer[7] | × | √ | × | × |
| Novel Loss Functions[5~6] | × | × | × | × |
| Novel Architecture[8~10] | √ | √ | × | × |

Table 2: Overiew of methods in our PINNacle. √ denotes the method is potentially designed to solve or show empirical improvements for problems encountering the challenge and vice versa.

custom activation functions (Jagtap et al., 2020a;b), and domain decomposition (Jagtap & Karniadakis, 2021; Shukla et al., 2021; Jagtap et al., 2020c; Moseley et al., 2021). Among adaptive activations for PINNs, we choose LAAF (Jagtap et al., 2020a) and GAAF (Jagtap et al., 2020b). Domain decomposition is a method that divides the whole domain into multiple subdomains and trains subnetworks on these subdomains. It is helpful for solving multi-scale problems, but multiple subnetworks increase the difficulty of training. XPINNs, cPINNs, and FBPINNs (Jagtap & Karniadakis, 2021; Jagtap et al., 2020c; Moseley et al., 2021) are three representative examples. We choose FBPINNs which is the state-of-the-art domain decomposition that applies domain-specific normalization to stabilize training.

### 3.3.2 STRUCTURE OF TOOLBOX

We provide a user-friendly and concise toolbox for implementing, training, and evaluating diverse PINN variants. Specifically, our codebase is based on DeepXDE and provides a series of encapsulated classes and functions to facilitate high-level training and custom PDEs. These utilities allow for a standardized and streamlined approach to the implementation of various PINN variants and PDEs. Moreover, we provided many auxiliary functions, including computing different metrics, visualizing predictions, and recording results.

Despite the unified implementation of diverse PINNs, we also design an adaptive multi-GPU parallel training framework To enhance the efficiency of systematic evaluations of PINN methods. It addresses the parallelization phase of training on multiple tasks, effectively balancing the computational loads of multiple GPUs. It allows for the execution of larger and more complex tasks. In a nutshell, we provide an example code for training and evaluating PINNs on two Poisson equations using our PINNacle framework in Appendix D.

### 3.4 EVALUATION

To comprehensively analyze the discrepancy between the PINN solutions and the true solutions, we adopt multiple metrics to evaluate the performance of the PINN variants. Generally, we choose several metrics that are commonly used in literature that apply to all methods and problems. We suppose that $\boldsymbol{y} = (y_i)_{i=1}^n$ is the prediction and $\boldsymbol{y}' = (y_i')_{i=1}^n$ to is ground truth, where $n$ is the number of testing examples. Specifically, we use $\ell_2$ relative error (L2RE), and $\ell_1$ relative error (L1RE) which are two most commonly used metrics to measure the global quality of the solution,

$$\text{L2RE} = \sqrt{\frac{\sum_{i=1}^n (y_i - y_i')^2}{\sum_{i=1}^n y_i'^2}}, \ \text{L1RE} = \frac{\sum_{i=1}^n |y_i - y_i'|}{\sum_{i=1}^n |y_i'|}. \tag{4}$$

We also compute max error (mERR in short), mean square error (MSE), and Fourier error (fMSE) for a detailed analysis of the prediction. These three metrics are computed as follows:

$$\text{MSE} = \frac{1}{n} \sum_{i=1}^n (y_i - y_i')^2, \ \text{mERR} = \max_i |y_i - y_i'|, \ \text{fMSE} = \frac{\sqrt{\sum_{k_{\min}}^{k_{\max}} |\mathcal{F}(\boldsymbol{y}) - \mathcal{F}(\boldsymbol{y}')|^2}}{k_{\max} - k_{\min} + 1}, \tag{5}$$

where $\mathcal{F}$ denotes Fourier transform of $\boldsymbol{y}$ and $k_{\min}, k_{\max}$ are chosen similar to PDEBench (Takamoto et al., 2022). Besides, for time-dependent problems, investigating the quality of the solution with time is important. Therefore we compute the L2RE error varying with time in Appendix E.2.

We assess the performance of PINNs against the reference from numerical solvers. Experimental results utilizing the $\ell_2$ relative error (L2RE) metric are incorporated within the main text, while a more exhaustive set of results, based on the aforementioned metrics, is available in the Appendix E.1.

| L2RE | Name | Vanilla | | | Loss Reweighting/Sampling | | | Optimizer | Loss functions | | Architecture | | |
|------|------|---------|--------|-------|------|------|------|-----------|-------|-------|------|------|-------|
| – | – | PINN | PINN-w | LBFGS | LRA | NTK | RAR | MultiAdam | gPINN | vPINN | LAAF | GAAF | FBPINN |
| Burgers | 1d-C | 1.45E-2 | 2.63E-2 | **1.33E-2** | 2.61E-2 | 1.84E-2 | 3.32E-2 | 4.85E-2 | 2.16E-1 | 3.47E-1 | 1.43E-2 | 5.20E-2 | 2.32E-1 |
| | 2d-C | 3.24E-1 | 2.70E-1 | 4.65E-1 | **2.60E-1** | 2.75E-1 | 3.45E-1 | 3.33E-1 | 3.27E-1 | 6.38E-1 | 2.77E-1 | 2.95E-1 | – |
| Poisson | 2d-C | 6.94E-1 | 3.49E-2 | NaN | 1.17E-1 | **1.23E-2** | 6.99E-1 | 2.63E-2 | 6.87E-1 | 4.91E-1 | 7.68E-1 | 6.04E-1 | 4.49E-2 |
| | 2d-CG | 6.36E-1 | 6.08E-2 | 2.96E-1 | 4.34E-2 | **1.43E-2** | 6.48E-1 | 2.76E-1 | 7.92E-1 | 2.86E-1 | 4.80E-1 | 8.71E-1 | 2.90E-2 |
| | 3d-CG | 5.60E-1 | 3.74E-1 | 7.05E-1 | **1.02E-1** | 9.47E-1 | 5.76E-1 | 3.63E-1 | 4.85E-1 | 7.38E-1 | 5.79E-1 | 5.02E-1 | 7.39E-1 |
| | 2d-MS | 6.30E-1 | 7.60E-1 | 1.45E+0 | 7.94E-1 | 7.48E-1 | 6.44E-1 | **5.90E-1** | 6.16E-1 | 9.72E-1 | 5.93E-1 | 9.31E-1 | 1.04E+0 |
| Heat | 2d-VC | 1.01E+0 | 2.35E-1 | 2.32E-1 | **2.12E-1** | 2.14E-1 | 9.66E-1 | 4.75E-1 | 2.12E+0 | 9.40E-1 | 6.42E-1 | 8.49E-1 | 9.52E-1 |
| | 2d-MS | 6.21E-2 | 2.42E-1 | **1.73E-2** | 8.79E-2 | 4.40E-2 | 7.49E-2 | 2.18E-1 | 1.13E-1 | 9.30E-1 | 7.40E-2 | 9.85E-1 | 8.20E-2 |
| | 2d-CG | 3.64E-2 | 1.45E-1 | 8.57E-1 | 1.25E-1 | 1.16E-1 | 2.72E-2 | 7.12E-2 | 9.38E-2 | – | **2.39E-2** | 4.61E-1 | 9.16E-2 |
| | 2d-LT | 9.99E-1 | 9.99E-1 | 1.00E+0 | 9.99E-1 | 1.00E+0 | 9.99E-1 | 1.00E+0 | 1.00E+0 | 1.00E+0 | 9.99E-1 | 9.99E-1 | 1.01E+0 |
| NS | 2d-C | 4.70E-2 | 1.45E-1 | 2.14E-1 | NaN | 1.98E-1 | 4.69E-1 | 7.27E-1 | 7.70E-2 | 2.92E-1 | **3.60E-2** | 3.79E-2 | 8.45E-2 |
| | 2d-CG | 1.19E-1 | 3.26E-1 | NaN | 3.32E-1 | 2.93E-1 | 3.34E-1 | 4.31E-1 | 1.54E-1 | 9.94E-1 | **8.24E-2** | 1.74E-1 | 8.27E+0 |
| | 2d-LT | 9.96E-1 | 1.00E+0 | 9.70E-1 | 1.00E+0 | 9.99E-1 | 1.00E+0 | 1.00E+0 | 9.95E-1 | 1.73E+0 | 9.98E-1 | 9.99E-1 | 1.00E+0 |
| Wave | 1d-C | 5.88E-1 | 2.85E-1 | NaN | 3.61E-1 | **9.79E-2** | 5.39E-1 | 1.21E-1 | 5.56E-1 | 8.39E-1 | 4.54E-1 | 6.77E-1 | 5.91E-1 |
| | 2d-CG | 1.84E+0 | 1.66E+0 | 1.33E+0 | 1.48E+0 | 2.16E+0 | 1.15E+0 | 1.09E+0 | 8.14E-1 | 7.99E-1 | 8.19E-1 | **7.94E-1** | 1.06E+0 |
| | 2d-MS | 1.34E+0 | 1.02E+0 | 1.37E+0 | 1.02E+0 | 1.04E+0 | 1.35E+0 | 1.01E+0 | 1.02E+0 | 9.82E-1 | 1.06E+0 | 1.06E+0 | 1.03E+0 |
| Chaotic | GS | 3.19E-1 | 1.58E-1 | NaN | 9.37E-2 | 2.16E-1 | 9.46E-2 | 9.37E-2 | 2.48E-1 | 1.16E+0 | 9.47E-2 | 9.46E-2 | **7.99E-2** |
| | KS | 1.01E+0 | 9.86E-1 | NaN | 9.57E-1 | 9.64E-1 | 1.01E+0 | 9.61E-1 | 9.94E-1 | 9.72E-1 | 1.01E+0 | 1.00E+0 | 1.02E+0 |
| High dim | PNd | 3.04E-3 | 2.58E-3 | 4.67E-4 | **4.58E-4** | 4.64E-3 | 3.59E-3 | 3.98E-3 | 5.05E-3 | – | 4.14E-3 | 7.75E-3 | – |
| | HNd | 3.61E-1 | 4.59E-1 | **1.19E-4** | 3.94E-1 | 3.97E-1 | 3.57E-1 | 3.02E-1 | 3.17E-1 | – | 5.22E-1 | 5.21E-1 | – |
| Inverse | PInv | 9.42E-2 | 1.66E-1 | NaN | 1.54E-1 | 1.93E-1 | 9.35E-2 | 1.30E-1 | 8.03E-2 | **1.96E-2** | 1.30E-1 | 2.54E-1 | 8.44E-1 |
| | HInv | 1.57E+0 | 5.26E-2 | NaN | 5.09E-2 | 7.52E-2 | 1.52E+0 | 8.04E-2 | 4.84E+0 | **1.19E-2** | 5.59E-1 | 2.12E-1 | 9.27E-1 |

Table 3: Mean L2RE of different PINN variants on our benchmark. Best results are highlighted in **blue** and second-places in lightblue. We do not bold any result if errors of all methods are about 100%. "NaN" means the method does not converge and "–" means the method is not suitable for the problem.

# 4 EXPERIMENTS

## 4.1 MAIN RESULTS

We now present experimental results. Except for the ablation study in Sec 4.3 and Appendix E.2, we use a learning rate of 0.001 and train all models with 20,000 epochs. We repeat all experiments three times and record the mean and std. Table 3 presents the main results for all methods on our tasks and shows their average $\ell_2$ relative errors (with standard deviation results available in Appendix E.1).

**PINN.** We use PINN-w to denote training PINNs with larger boundary weights. The vanilla PINNs struggle to accurately solve complex physics systems, indicating substantial room for improvement. Using an $\ell_2$ relative error (L2RE) of 10% as a threshold for a successful solution, we find that vanilla PINN only solves 10 out of 22 tasks, most of which involve simpler equations (e.g., 1.45% on Burgers-1d-C). They encounter significant difficulties when faced with physics systems characterized by complex geometries, multi-scale phenomena, nonlinearity, and longer time spans. This shows that directly optimizing an average of the PDE losses and initial/boundary condition losses leads to critical issues such as loss imbalance, suboptimal convergence, and limited expressiveness.

**PINN variants.** PINN variants offer approaches to addressing some of these challenges to varying degrees. Methods involving loss reweighting and resampling have shown improved performance in some cases involving complex geometries and multi-scale phenomena (e.g., 1.43% on Poisson-2d-CG). This is due to the configuration of loss weights and sampled collocation points, which adaptively place more weight on more challenging domains during the training process. However, these methods still struggle with Wave equations, Navier-Stokes equations, and other cases with higher dimensions or longer time spans. MultiAdam, a representative of novel optimizers, solves several simple cases and the chaotic GS equation (9.37%), but does not significantly outperform other methods. The new loss term of variational form demonstrates significant superiority in solving inverse problems (*e.g.,* 1.19% on HInv for vPINN), but no clear improvement in fitting error over standard PINN in forward cases. Changes in architecture can enhance expressiveness and flexibility for cases with complex geometries and multi-scale systems. For example, FBPINN achieves the smallest error on the chaotic GS equation (7.99%), while LAAF delivers the best fitting result on Heat-2d-CG (2.39%).

| L2RE | Name | Vanilla | Loss Reweighting/Sampling | | | Optimizer | Loss functions | | Architecture | | |
|------|------|---------|------|------|------|-----------|--------|--------|------|------|--------|
| – | – | PINN | LRA | NTK | RAR | MultiAdam | gPINN | vPINN | LAAF | GAAF | FBPINN |
| Burgers-P | 2d-C | 4.74E-01 | 4.36E-01 | **4.13E-01** | 4.71E-01 | 4.93E-01 | 4.91E-01 | 2.82E+0 | 4.37E-01 | 4.34E-01 | - |
| Poisson-P | 2d-C | 2.24E-01 | 7.07E-02 | **1.66E-02** | 2.33E-01 | 8.24E-02 | 4.03E-01 | 5.51E-1 | 1.84E-01 | 2.97E-01 | 2.87E-2 |
| Heat-P | 2d-MS | 1.73E-01 | 1.23E-01 | 1.50E-01 | 1.53E-01 | 4.00E-01 | 4.59E-01 | 5.12E-1 | **6.27E-02** | 1.89E-01 | 2.46E-1 |
| NS-P | 2d-C | 3.89E-01 | - | 4.52E-01 | 3.91E-01 | 9.33E-01 | 7.19E-01 | 3.76E-1 | **3.63E-01** | 4.85E-01 | 3.99E-1 |
| Wave-P | 1d-C | 5.22E-01 | 3.44E-01 | **2.69E-01** | 5.05E-01 | 6.89E-01 | 7.66E-01 | 3.58E-1 | 4.03E-01 | 9.00E-01 | 1.15E+0 |
| High dim-P | HNd | 7.66E-03 | 6.53E-03 | 9.04E-03 | 8.07E-03 | **2.22E-03** | 7.87E-03 | - | 6.97E-03 | 1.94E-01 | - |

Table 4: Results of different PINN variants on parametric PDEs. We report average L2RE on all examples within a class of PDE. We **bold** the best results across all methods.

**Discussion.** For challenges related to complex geometries and multi-scale phenomena, some methods can mitigate these issues by implementing mechanisms like loss reweighting, novel optimizers, and better capacity through adaptive activation. This holds true for the 2D cases of Heat and Poisson equations, which are classic linear equations. However, when systems have higher dimensions (Poisson3d-CG) or longer time spans (Heat2d-LT), all methods fail to solve, highlighting the difficulties associated with complex geometries and multi-scale systems.

In contrast, nonlinear, long-time PDEs like 2D Burgers, NS, and KS pose challenges for most methods. These equations are sensitive to initial conditions, resulting in complicated solution spaces and more local minima for PINNs (Steger et al., 2022). The Wave equation, featuring a second-order time derivative and periodic behavior, is particularly hard for PINNs, which often become unstable and may violate conservation laws (Jagtap et al., 2020c; Wang et al., 2022a). Although all methods perform well on Poisson-Nd, only PINN with LBFGS solves Heat-Nd, indicating the potential of a second-order optimizer for solving high dimensional PDEs(Tang et al., 2021).

## 4.2 PARAMETERIZED PDE EXPERIMENTS

The performance of PINNs is also highly influenced by parameters in PDEs (Krishnapriyan et al., 2021). To investigate whether PINNs could handle a class of PDEs, we design this experiment to solve the same PDEs with different parameters. We choose 6 PDEs, i.e., Burgers2d-C, Poisson2d-C, Heat2d-MS, NS-C, Wave1d-C, and Heat-Nd (HNd), with each case containing five parameterized examples. Details of the parametrized PDEs are shown in Appendix B. Here we report the average L2RE metric on these parameterized PDEs for every case, and results are shown in the following Table 4. First, we see that compared with the corresponding cases in Table E.1, the mean L2RE of parameterized PDEs is usually higher. We suggest that this is because there are some difficult cases under certain parameters for these PDEs with very high errors. Secondly, we find that PINN-NTK works well on parameterized PDE tasks which achieve three best results among all six experiments. We speculate that solving PDEs with different parameters requires different weights for loss terms, and PINN-NTK is a powerful method for automatically balancing these weights.

## 4.3 HYPERPARAMETER ANALYSIS

The performance of PINNs is strongly affected by hyperparameters, with each variant potentially introducing its own unique set. Here we aim to investigate the impact of several shared and method-specific hyperparameters via ablation studies. We consider the batch size and the number of training epochs. The corresponding results are shown in Figure 2. We focus on a set of problems, i.e., Burgers1d, GS, Heat2d-CG, and Poisson2d-C. Detailed numerical results and additional findings are in Appendix E.2.

**Batch Size.** The left figure of Figure 18 shows the testing L2RE of PINNs for varying batch sizes. It suggests that larger batch sizes provide superior results, possibly due to improved gradient estimation accuracy. Despite the GS and Poisson2d-C witnessing saturation beyond a batch size of 2048, amplifying the batch sizes generally enhances results for the remaining problems.

**Training Epochs.** The right figure of Figure 18 illustrates the L2RE of PINNs across varying training epochs. Mirroring the trend observed with batch size, we find that an increase in epochs leads to a decrease in overall error. However, the impact of the number of epochs also has a saturation point. Notably, beyond 20k or 80k epochs, the error does not decrease significantly, suggesting convergence of PINNs around 20~ 80 epochs.

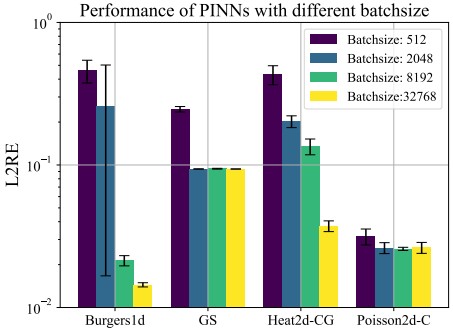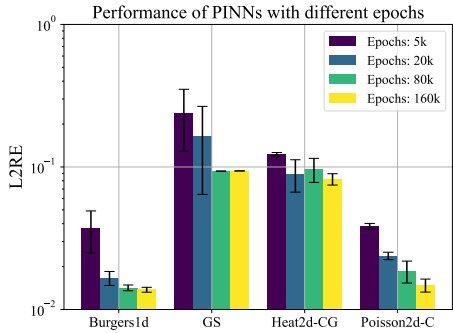

Figure 2: Performance of vanilla PINNs under different batch sizes (number of collocation points), which is shown in the left figure; and number of training epochs, which is shown in the right figure.

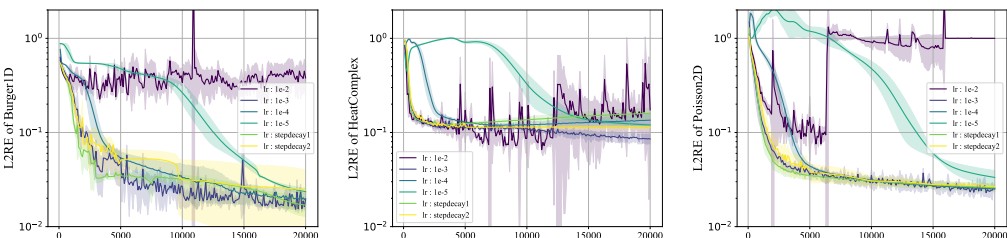

Figure 3: Convergence curve of PINNs with different learning rate schedules on Burgers1d, Heat2d-CG, and Poisson2d-C.

**Learning Rates.** The performance of standard PINNs under various learning rates and learning rate schedules is shown in Figure 3. We observe that the influence of the learning rate on performance is intricate, with optimal learning rates varying across problems. Furthermore, PINN training tends to be unstable. High learning rates, such as $10^{-2}$, often lead to error spikes, while low learning rates, like $10^{-5}$, result in slow convergence. Our findings suggest that a moderate learning rate, such as $10^{-3}$ or $10^{-4}$, or a step decay learning rate schedule, tends to yield more stable performance.

## 5 CONCLUSION AND DISCUSSION

In this work, we introduced PINNacle, a comprehensive benchmark offering a user-friendly toolbox that encompasses over 10 PINN methods. We evaluated these methods against more than 20 challenging PDE problems, conducting extensive experiments and ablation studies for hyperparameters. Looking forward, we plan to expand the benchmark by integrating additional state-of-the-art methods and incorporating more practical problem scenarios.

Our analysis of the experimental results yields several key insights. First, domain decomposition is beneficial for addressing problems characterized by complex geometries, and PINN-NTK is a strong method for balancing loss weights as experiments show. Second, the choice of hyperparameters is crucial to the performance of PINNs. Selecting a larger batch size and appropriately weighting losses or betas in Adam may significantly reduce the error. However, the best hyperparameters usually vary with PDEs. Third, we identify high-dimensional and nonlinear problems as a pressing challenge. The overall performance of PINNs is not yet on par with traditional numerical methods (Grossmann et al., 2023). Fourth, from a theoretical standpoint, the exploration of PINNs' loss landscape during the training of high-dimensional or nonlinear problems remains largely unexplored. Finally, from an algorithmic perspective, integrating the strengths of neural networks with numerical methods like preconditioning, weak formulation, and multigrid may present a promising avenue toward overcoming the challenges identified herein (Markidis, 2021).

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
