# PINNacle: A Comprehensive Benchmark of Physics-Informed Neural Networks for Solving PDEs

**Zhongkai Hao**[1][*] **Jiachen Yao**[1][*]**, Chang Su**[1][*]**, Hang Su**[1]**, Ziao Wang**[1]**,**

**Fanzhi Lu**[1]**, Zeyu Xia**[1]**, Yichi Zhang**[1]**, Songming Liu**[1]**, Lu Lu**[2]**, Jun Zhu**[1]

[1]Dept. of Comp. Sci. and Tech., Institute for AI, BNRist Center, THBI Lab,
Tsinghua-Bosch Joint ML Center, Tsinghua University
[2]Department of Chemical and Biomolecular Engineering, University of Pennsylvania
`hzj21@mails.tsinghua.edu.cn`

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

## A    OVERVIEW OF APPENDICES

We provide supplementary details about problems and experiments for the main text in the Appendix. In Appendix B, we provide mathematical descriptions and visualization for all PDEs in this paper. In Appendix C, we list the detailed hyperparameters and training/testing settings. In Appendix D, we provide a high-level overview of the codebase of the toolbox. In Appendix E, the results for the main experiments, i.e., the performance of L2RE, L1RE, MSE, and runtime for all methods on all PDEs are displayed. In Appendix F, we show the visualization results for several methods on some problems.

## B    DETAILS OF PDES AND METHODS

Here provide details of PDE tasks used for evaluating different variants of PINNs. Denote $u$ to be the function to solve and $x, t$ to be spatial and temporal variables.

### B.1    DEFINITIONS FOR PDES IN MAIN EXPERIMENTS

**1. One-dimensional Burgers Equation (Burgers1d)**

The Burgers 1D equation is given by

$$u_t + uu_x = \nu u_{xx}. \tag{6}$$

The domain is defined as

$$(x, t) \in \Omega = [-1, 1] \times [0, 1]. \tag{7}$$

The initial and boundary conditions are

$$u(x, 0) = -\sin \pi x, \tag{8}$$
$$u(-1, t) = u(1, t) = 0. \tag{9}$$

The parameter is

$$\nu = \frac{0.01}{\pi}. \tag{10}$$

**2. 2D Coupled Burgers equation (Burgers 2d)**

The 2D Coupled Burgers equation is given by

$$\boldsymbol{u}_t + \boldsymbol{u} \cdot \nabla \boldsymbol{u} - \nu \Delta \boldsymbol{u} = 0, \tag{11}$$
$$\boldsymbol{u}(0, y, t) = \boldsymbol{u}(L, y, t), \quad \boldsymbol{u}(x, 0, t) = \boldsymbol{u}(x, L, t), \tag{12}$$
$$\{x, y\} \in [0, L], \quad t \in [0, T], \tag{13}$$

The domain is defined as

$$(x, y, t) \in \Omega = [0, L]^2 \times [0, 1]. \tag{14}$$

The initial conditions are given by

$$\boldsymbol{w}(x, y) = \sum_{i=-L}^{L} \sum_{j=-L}^{L} \boldsymbol{a}_{ij} \sin(2\pi(ix + jy)) + \boldsymbol{b}_{ij} \cos(2\pi(ix + jy)), \tag{15}$$
$$\boldsymbol{u}(x, y, 0) = 2\boldsymbol{w}(x, y) + \boldsymbol{c} \tag{16}$$

where $a, b, c \sim N(0, 1)$. The parameters are

$$L = 4, \quad T = 1, \quad \nu = 0.001. \tag{17}$$

**3. Poisson 2D Classic (Poisson2d-C)**

The Poisson 2D equation is given by

$$-\Delta u = 0. \tag{18}$$

The domain is a rectangle minus four circles $\Omega = \Omega_{rec} \setminus R_i$ where $\Omega_{rec} = [-0.5, 0.5]^2$ is the rectangle and $R_i$ denotes four circle areas:

$$
\begin{align}
R_1 &= \{(x, y) : (x - 0.3)^2 + (y - 0.3)^2 \le 0.1^2\}, \tag{19} \\
R_2 &= \{(x, y) : (x + 0.3)^2 + (y - 0.3)^2 \le 0.1^2\}, \tag{20} \\
R_3 &= \{(x, y) : (x - 0.3)^2 + (y + 0.3)^2 \le 0.1^2\}, \tag{21} \\
R_4 &= \{(x, y) : (x + 0.3)^2 + (y + 0.3)^2 \le 0.1^2\}. \tag{22}
\end{align}
$$

The boundary condition is

$$
\begin{align}
u &= 0, x \in \partial R_i, \tag{23} \\
u &= 1, x \in \partial \Omega_{rec}. \tag{24}
\end{align}
$$

## 4. Poisson-Boltzmann (Helmholtz) 2D Irregular Geometry (Poisson2d-CG)

The Poisson-Boltzmann (Helmholtz) 2D equation is given by

$$-\Delta u + k^2 u = f(x, y). \tag{25}$$

The function $f(x)$ is defined as

$$f(x) = A \cdot \left( \sum_i \mu_i^2 + x_i^2 \right) \sin(\mu_1 \pi x_1) \sin(\mu_2 \pi x_2). \tag{26}$$

The domain is $[-1, 1]^2$ and the boundary conditions are

$$
\begin{align}
u &= 0.2, \quad x \in \partial \Omega_{\text{rec}}, \tag{27} \\
u &= 1, \quad x \in \partial \Omega_{\text{circle}}. \tag{28}
\end{align}
$$

Parameter references are

$$\mu_1 = 1, \quad \mu_2 = 4, \quad k = 8, \quad A = 10. \tag{29}$$

The domain is $[-1, 1]^2$ with several circles removed. The circles $\Omega_{circle} = \cup_{i=1}^4 R_i$ are

$$
\begin{align}
R_1 &= \{(x, y) : (x - 0.5)^2 + (y - 0.5)^2 \le 0.2^2\} \tag{30} \\
R_2 &= \{(x, y) : (x - 0.4)^2 + (y + 0.4)^2 \le 0.4^2\} \tag{31} \\
R_3 &= \{(x, y) : (x + 0.2)^2 + (y + 0.7)^2 \le 0.1^2\} \tag{32} \\
R_4 &= \{(x, y) : (x + 0.6)^2 + (y - 0.5)^2 \le 0.3^2\} \tag{33}
\end{align}
$$

## 5. Poisson 3D Complex Geometry with Two Domains (Poisson3d-CG)

The Poisson 3D equation with two domains is given by

$$-\mu_i \Delta u + k_i^2 u = f(x, y, z), \quad i = 1, 2. \tag{34}$$

The function $f(x, y, z)$ is defined as

$$
\begin{aligned}
f(x, y, z) =& A_1 \frac{\exp(\sin m_1 \pi x + \sin m_2 \pi y + \sin m_3 \pi z)}{x^2 + y^2 + z^2 + 1} (x^2 + y^2 + z^2 - 1) \\
& + A_2 \sin(m_1 \pi x) \sin(m_2 \pi y) \sin(m_3 \pi z).
\end{aligned} \tag{35}
$$

The coefficients are defined as $\begin{cases} \mu = \mu_1, & k = k_1, \quad x \in \Omega_1, \\ \mu = \mu_2, & k = k_2, \quad x \in \Omega_2. \end{cases}$

The boundary condition is

$$\frac{\partial u}{\partial n} = 0, \quad x \in \partial \Omega. \tag{36}$$

The domains and other parameters are defined as follows:

$$\Omega_1 = [0,1] \times [0,1] \times [0,0.5]/\cup_{i=1}^4 R_i, \tag{37}$$

$$\Omega_2 = [0,1] \times [0,1] \times [0.5,1]/\cup_{i=1}^4 R_i. \tag{38}$$

The circular regions $R_i$ are

$$R_1 = \{(x,y,z) : (x-0.4)^2 + (y-0.3)^2 + (z-0.6)^2 \le 0.2^2\} \tag{39}$$

$$R_1 = \{(x,y,z) : (x-0.6)^2 + (y-0.7)^2 + (z-0.6)^2 \le 0.2^2\} \tag{40}$$

$$R_1 = \{(x,y,z) : (x-0.2)^2 + (y-0.8)^2 + (z-0.7)^2 \le 0.1^2\} \tag{41}$$

$$R_1 = \{(x,y,z) : (x-0.6)^2 + (y-0.2)^2 + (z-0.3)^2 \le 0.1^2\} \tag{42}$$

$$\tag{43}$$

Other parameters are

$$m_1 = 1, m_2 = 10, m_3 = 5, \mu_1 = 1, \mu_2 = 1, k_1 = 8, k_2 = 10, A_1 = 20, A_2 = 100. \tag{44}$$

## 6. 2D Poisson equation with many subdomains (Poisson2d-MS)

The PDE and boundary condition is given by

$$-\nabla(a(x)\nabla u) = f(x,y), x in \Omega \tag{45}$$

$$\frac{\partial u}{\partial n} + u = 0, x \in \partial\Omega. \tag{46}$$

Here the domain is $(x,y) \in \Omega = [-10,10]^2$. We divide the whole domain into many small squares, and $a(x)$ is a piecewise linear function in each square. We store the $a(x)$ in a file in practical implementation.

## 7. 2D Heat with Varying Coefficients (Heat2d-VC)

The 2D heat equation with a varying source is given by

$$\frac{\partial u}{\partial t} - \nabla(a(x)\nabla u) = f(x,t). \tag{47}$$

The domain is $\Omega \times T = [0,1]^2 \times [0,5]$. The function $a(x)$ is chosen similarly to Darcy flow but with an exponential GRF. The function $f(x,t)$ is defined as

$$f(x,t) = A \sin(m_1 \pi x) \sin(m_2 \pi y) \sin(m_3 \pi t). \tag{48}$$

with $A = 200, m_1 = 1, m_2 = 5, m_3 = 1$. The initial and boundary conditions are

$$u(x,y,0) = 0, x \in \Omega \tag{49}$$

$$u(x,y,t) = 0, x \in \partial\Omega. \tag{50}$$

## 8. 2D Heat Multi-Scale (Heat2d-MS)

The 2D heat multi-scale equation is given by

$$\frac{\partial u}{\partial t} - \frac{1}{(500\pi)^2} u_{xx} - \frac{1}{\pi^2} u_{yy} = 0, \tag{51}$$

with domain $\Omega \times T = [0,1]^2 \times [0,5]$.

The initial and boundary conditions are

$$u(x,y,0) = \sin(20\pi x)\sin(\pi y), \quad x \in \Omega, \tag{52}$$

$$u(x,y,t) = 0, \quad x \in \partial\Omega. \tag{53}$$

## 9. 2D Heat Complex Geometry (Heat Exchanger, Heat2d-CG)

The 2D heat equation for a complex geometry is given by

$$\frac{\partial u}{\partial t} - \Delta u = 0. \tag{54}$$

The domain is defined as $\Omega \times T = ([-8, 8] \times [-12, 12] \setminus \cup_i R_i) \times [0, 3]$.

The boundary condition is

$$-n \cdot (-c\nabla u) = g - qu. \tag{55}$$

Here we choose $c = 1$. The positions of large circles are

$$(\pm 4, \pm 3), \quad (\pm 4, \pm 9), \quad (0, 0), \quad (0, \pm 6), \quad r = 1 \tag{56}$$

with $g = 5$ and $q = 1$. The positions of small circles are

$$(\pm 3.2, \pm 6), \quad (\pm 3.2, 0), \quad r = 0.4 \tag{57}$$

with $g = 1$ and $q = 1$. For the rectangular boundary conditions, $g = 0.1$ and $q = 1$.

## 10. 2D Heat Long Time (Heat2d-LT)

The governing PDE is

$$\frac{\partial u}{\partial t} = 0.001\Delta u + 5\sin(ku^2)\left(1 + 2\sin\left(\frac{\pi t}{4}\right)\right)\sin(m_1\pi x)\sin(m_2\pi y) \tag{58}$$

with domain $\Omega \times T = [0, 1]^2 \times [0, 100]$, $m_1 = 4$, $m_2 = 2$, and $k = 1$.

The initial and boundary conditions are given by

$$\begin{aligned}
u(x, y, 0) &= \sin(4\pi x)\sin(3\pi y), x \in \Omega \tag{59}\\
u(x, y, t) &= 0, \quad x \in \partial\Omega. \tag{60}
\end{aligned}$$

## 11. 2D NS lid-driven flow (NS2d-C).

The PDE is given by

$$\begin{aligned}
\boldsymbol{u} \cdot \nabla\boldsymbol{u} + \nabla p - \frac{1}{Re}\Delta\boldsymbol{u} &= 0, x \in \Omega \tag{61}\\
\nabla \cdot \boldsymbol{u} &= 0, x \in \Omega \tag{62}
\end{aligned}$$

The domain is $\Omega = [0, 1]^2$, the top boundary is $\Gamma_1$, the left, right and bottom boundary is $\Gamma_2$.

The boundary conditions are

$$\begin{aligned}
\boldsymbol{u}(\boldsymbol{x}) &= (4x(1 - x), 0), x \in \Gamma_1 \tag{63}\\
\boldsymbol{u}(\boldsymbol{x}) &= (0, 0), x \in \Gamma_2 \tag{64}\\
p &= 0, x = (0, 0). \tag{65}
\end{aligned}$$

The Reynolds number $Re = 100$.

## 12. 2D Back Step Flow (NS-CG)

The equations and boundary conditions are given by

$$\begin{aligned}
u \cdot \nabla u + \nabla p - \frac{1}{Re}\Delta u &= 0, \tag{66}\\
\nabla \cdot u &= 0. \tag{67}
\end{aligned}$$

The domain is defined as $\Omega = [0, 4] \times [0, 2] \setminus ([0, 2] \times [1, 2] \bigcup R_i)$ (excluding the top-left quarter).

The inlet velocity is given by $u_{in} = 4y(1 - y)$, the outlet pressure is $p = 0$, and the boundary condition is no-slip: $\mathbf{u} = 0$. The Reynolds number of $Re = 100$.

## 13. 2D NS Long Time (NS2d-LT)

The PDE of this case is given by

$$\frac{\partial u}{\partial t} + u \cdot \nabla u + \nabla p - \frac{1}{\text{Re}} \Delta u = f(x, y, t), \tag{68}$$

$$\nabla \cdot u = 0. \tag{69}$$

The domain is $\Omega \times T = ([0, 2] \times [0, 1]) \times [0, 5]$, and the forcing term $f(x, y, t)$ can be given as

$$f(x, y, t) = (0, -\sin(\pi x)\sin(\pi y)\sin(\pi t)). \tag{70}$$

The boundary conditions are similar to case 12, and the left inlet initial condition can be given as an oscillatory form:

$$u(0, y, t) = \sin(\pi y)(A_1 \sin(\pi t) + A_2 \sin(3\pi t) + A_3 \sin(5\pi t)). \tag{71}$$

where $A_1 = 1, A_2 = 1, A_3 = 1$.

The initial condition in the domain is

$$u(x, y, 0) = 0. \tag{72}$$

## 14. Basic 1D Wave Equation (Wave1d-C)

The governing PDE is

$$u_{tt} - 4u_{xx} = 0 \tag{73}$$

The domain is $\Omega \times T = [0, 1] \times [0, 1]$. The boundary conditions are

$$u(0, t) = u(1, t) = 0 \tag{74}$$

The initial condition:

$$u(x, 0) = \sin(\pi x) + \frac{1}{2}\sin(4\pi x) \tag{75}$$

$$u_t(x, 0) = 0 \tag{76}$$

The analytical solution of this problem is

$$u(x, t) = \sin(\pi x)\cos(2\pi t) + \frac{1}{2}\sin(4\pi x)\cos(8\pi t). \tag{77}$$

## 15. 2D Wave Equation in Heterogeneous Medium (Wave2d-CG)

The governing PDE is given by

$$\left[\nabla^2 - \frac{1}{c(x)}\frac{\partial^2}{\partial t^2}\right] u(x, t) = 0 \tag{78}$$

The Domain is $\Omega = [-1, 1] \times [-1, 1]$ and the initial condition is

$$u(x, 0) = \exp\left(-\frac{\|x - \mu\|^2}{2\sigma^2}\right), x \in \Omega \tag{79}$$

$$\frac{\partial u}{\partial t}(x, 0) = 0, x \in \Omega \tag{80}$$

The boundary conditions are

$$\frac{\partial u}{\partial n} = 0, x \in \partial\Omega \tag{81}$$

The parameters are

$$\mu = (-0.5, 0), \sigma = 0.3, \tag{82}$$

and $c(x)$ are generated by a Gaussian random field.

## 16. 2D Multi-Scale Long Time Wave Equation (Wave2d-MS)

The governing PDE is

$$u_{tt} - (u_{xx} + a^2 u_{yy}) = 0 \tag{83}$$

The domain is defined as $\Omega = [0,1]^2 \times [0,100]$ and the boundary and initial conditions are

$$u(x,y,t) = c_1 \sinh(m_1 \pi x) \sinh(n_1 \pi y) \cos(p_1 \pi t), (x,y) \in \partial\Omega. \tag{84}$$

$$\frac{\partial u}{\partial t}(x,y,0) = 0 \tag{85}$$

The exact solution to this problem is

$$u(x,y,t) = c_1 \sinh(m_1 \pi x) \sinh(n_1 \pi y) \cos(p_1 \pi t), \tag{86}$$

where $a = \sqrt{2}, m_1 = 1, n_1 = 1, p_1 = \sqrt{3}$ and $c_1 = 1$.

### 17. 2D Diffusion-Reaction Gray-Scott Model (GS)

The governing PDE is

$$\begin{align}
u_t &= \varepsilon_1 \Delta u + b(1-u) - uv^2 \tag{87} \\
v_t &= \varepsilon_2 \Delta v - dv + uv^2 \tag{88}
\end{align}$$

The domain is $\Omega \times T = [-1,1]^2 \times [0,200]$ and parameters are

$$b = 0.04, d = 0.1, \varepsilon_1 = 1 \times 10^{-5}, \varepsilon_2 = 5 \times 10^{-6} \tag{89}$$

The initial conditions are

$$\begin{align}
u(x,y,0) &= 1 - \exp(-80((x+0.05)^2 + (y+0.02)^2)) \tag{90} \\
v(x,y,0) &= \exp(-80((x-0.05)^2 + (y-0.02)^2)) \tag{91}
\end{align}$$

The visualization of the reference solution of this case is in Figure 16.

### 18. Kuramoto-Sivashinsky Equation (KS)

The governing PDE is

$$u_t + \alpha u u_x + \beta u_{xx} + \gamma u_{xxxx} = 0 \tag{92}$$

The domain is $\Omega \times T = [0, 2\pi] \times [0,1]$. (Note: Error may increase rapidly in chaotic problems.)

$$\alpha = \frac{100}{16}, \beta = \frac{100}{16^2}, \gamma = \frac{100}{16^4} \tag{93}$$

The initial condition is

$$u(x,0) = \cos(x)(1 + \sin(x)) \tag{94}$$

The reference solution of KS equation is shown in Figure B.1.

### 19. N-Dimensional Poisson equation (PNd)

The governing PDE is

$$-\Delta u = \frac{\pi^2}{4} \sum_{i=1}^{n} \sin\left(\frac{\pi}{2}x_i\right) \tag{95}$$

The domain is defined by $\Omega = [0,1]^n$. The exact solution is

$$u = \sum_{i=1}^{n} \sin\left(\frac{\pi}{2}x_i\right) \tag{96}$$

We choose $n = 5$ in our code.

### 20. N-Dimensional Heat Equation (HNd)

The governing PDE is

$$\begin{align}
\frac{\partial u}{\partial t} &= k\Delta u + f(x,t), x \in \Omega \times [0,1] \tag{97} \\
\boldsymbol{n} \cdot \nabla u &= g(x,t), x \in \partial\Omega \times [0,1] \tag{98} \\
u(x,0) &= g(x,0), x \in \Omega \tag{99}
\end{align}$$

The geometric domain $\Omega = \{x : |x|_2 \leqslant 1\}$ is a unit sphere in $d$-dimensional space. We choose dimension $d = 5$.

$$k = \frac{1}{d} \tag{100}$$

The two functions are

$$f(x,t) = -\frac{1}{d}|x|_2^2 \exp\left(\frac{1}{2}|x|_2^2 + t\right) \tag{101}$$

$$g(x,t) = \exp\left(\frac{1}{2}|x|_2^2 + t\right) \tag{102}$$

We can see that the exact solution of the equation is $g(x,t)$.

## 21. Poisson inverse problem (PInv)

The governing PDE is

$$-\nabla(a\nabla u) = f \tag{103}$$

The geometric domain is $\Omega = [0,1]^2$, and

$$u = \sin \pi x \sin \pi y. \tag{104}$$

The source term $f$ is

$$f = \frac{2\pi^2 \sin \pi x \sin \pi y}{1 + x^2 + y^2 + (x-1)^2 + (y-1)^2} + \frac{2\pi((2x-1)\cos \pi x \sin \pi y + (2y-1)\sin \pi x \cos \pi y)}{(1 + x^2 + y^2 + (x-1)^2 + (y-1)^2)^2}. \tag{105}$$

To ensure the uniqueness of the solution, we impose a boundary condition of $a(x,y)$, i.e.,

$$a(x,y) = \frac{1}{1 + x^2 + y^2 + (x-1)^2 + (y-1)^2}, x \in \partial\Omega \tag{106}$$

We sample data of $u(x,y)$ with 2500 uniformly distributed $50 \times 50$ points and add Gaussian noise $\mathcal{N}(0, 0.1)$ to it. The goal is to reconstruct the diffusion coefficients. We see that the ground truth of $a(x,y)$ is

$$a(x,y) = \frac{1}{1 + x^2 + y^2 + (x-1)^2 + (y-1)^2}, x \in \Omega. \tag{107}$$

## 22. Heat (Diffusion) inverse problem (HInv)

The governing PDE of this inverse problem is

$$u_t - \nabla(a\nabla u) = f \tag{108}$$

The geometric domain is $\Omega \times T = [-1,1]^2 \times [0,1]$, and

$$u = e^{-t} \sin \pi x \sin \pi y \tag{109}$$

Similarly, we impose a boundary condition for the diffusion coefficient field:

$$a(x,y) = 2, \partial x \in \Omega. \tag{110}$$

Then the source function $f$ is

$$f = ((4\pi^2 - 1)\sin \pi x \sin \pi y + \pi^2(2\sin^2 \pi x \sin^2 \pi y - \cos^2 \pi x \sin^2 \pi y - \sin^2 \pi x \cos^2 \pi y))e^{-t} \tag{111}$$

We sample data of $u(x,y,t)$ randomly with 2500 points from the temporal domain $\Omega \times T$ and add Gaussian noise $\mathcal{N}(0, 0.1)$ to it. The goal is to reconstruct the diffusion coefficients. We see that the ground truth is

$$a(x,y) = 2 + \sin \pi x \sin \pi y, x \in \Omega. \tag{112}$$

## B.2   DEFINITIONS AND DESIGN CHOICES FOR PARAMETRIC PDEs

We design a set of parametric PDEs and evaluate the average performance of PINN variants on cases with different parameters. We choose Burgers2d-C, Poisson2d-C, Heat2d-MS, NS2d-C, Wave2d-C, and Heat-Nd to design these parametric cases.

### 1. 2D Coupled Burgers equation (Burgers2d-C) with different initial values.

The initial values of this case are shown in Eq 16 where $a$ and $b$ are sampled from Gaussian Random Field. Here the initial values are used as parameters and we sample 5 different $a$ and $b$ from GRF and test the performance of PINN variants on all 5 cases. Each parametrized PDE is solved using COMSOL. In PDEBench, the authors similarly tested the average effect of PDEs sampled multiple times from the GRF with the same equation. Since the GRF has not changed, there is not much variation in the magnitude and frequency of the initial flow velocity, but there may be significant differences in their spatial distribution. This can also lead to differences in difficulty when solving with the PINN method. From the 4, we see that the error of the best method increased from 26% to 41%, indicating a significant influence of the flow distribution on the solution.

### 2. Poisson 2d Classic (Poisson2d-C)

This PDE is defined on $\Omega = [-L, L]^2$. We parametrize this case by using different domain scales $L$ from $\{1, 2, 4, 8, 16\}$. Since this PDE is linear, we could compute the ground truth solution by linearly scaling the original PDE where $L = 0.5$. Some papers (Yao et al., 2023) pointed out that the effect of PINN is influenced by the size of the domain. This is because scaling the domain directly to $[0, 1]^d$ may be suboptimal and can lead to an imbalanced ratio of PDE loss to boundary loss. This is because PINNs are sensitive to initialization, so different domain scales might lead to different results. Here the real solution of this linear PDE can be obtained through a linear transformation from a solution of another domain scale $L$. The condition number does not differ when we change $L$, making it suitable to study the influence of domain scale on PINN's performance. We observed from the results that some methods (PINN-NTK, MultiAdam, FBPINN) are relatively robust to domain scale.

### 3. 2D Heat Multi-Scale (Heat2d-MS)

We parameterize this case using different initial conditions in Eq 53,

$$u(x, y, 0) = \sin(a\pi x)\sin(\pi y). \tag{113}$$

Here we choose $(a, b)$ from $\{(20, 1), (1, 20), (10, 2), (2, 10), (5, 4)\}$. The reference solutions for different parameters are solved using COMSOL. Changes in the frequency of the initial condition will lead to changes in the frequency of the solution, which allows us to study the influence of the initial condition frequency on PINN. Comparing the results of several experiments, we found that the loss reweighting strategy of PINN-NTK and the adaptive activation function of LAAF perform well for multi-scale problems overall. However, when the frequency variation range is more significant, both their performances decline, suggesting room for improvement.

### 4. 2D NS lid-driven flow (NS2d-C)

We parametrize NS2d-C by setting different speeds at the top boundary in Eq 65,

$$\boldsymbol{u}(\boldsymbol{x}) = (ax(1 - x), 0), x \in \Gamma_1, \tag{114}$$

where $a$ is chosen from $\{2, 4, 8, 16, 32\}$. The reference solutions for different parameters are solved using COMSOL. Different flow rates imply different Reynolds numbers, thus altering the difficulty of solving the equation. As the Reynolds number increases, the condition number of the equation will also increase. Generally, the higher the Reynolds number, the more likely turbulence or some small-scale complex flow states will occur. Testing different Reynolds numbers is a natural idea. Specifically, we chose a velocity $u = ax(1 - x)$, where $a$ ranges between 2 and 32. Compared to the main experiment with $a = 4$, the Reynolds number increased eightfold when $a = 32$.

### 5. 1D Wave Equation

We parametrize this case with different initial conditions in Eq 76,

$$u(x, 0) = \sin(\pi x) + \frac{1}{2}\sin(a\pi x), \tag{115}$$

where $a$ is chosen from $\{2, 4, 6, 8, 10\}$. The ground truth solution is given by,

$$u(x, t) = \sin(\pi x)\cos(2\pi t) + \frac{1}{2}\sin(\pi a x)\cos(2a\pi t). \tag{116}$$

### 6. N-Dimensional Heat Equation

We parametrize this case by choosing a different number of dimensions $n$ from $\{4, 5, 6, 8, 10\}$. The solutions are given by Eq 102. Although neural networks are theoretically universal function approximators, the ability to fit the solution of high-dimensional PDEs still needs to be studied. So, we chose heat equations of different dimensions to compare the effects of various PINN methods. We observed that for high-dimensional heat equations, the improved optimizer MultiAdam is very helpful in solving high-dimensional problems.

### B.3 RELATIONSHIP WITH PDEBENCH AND PDEARENA

Here we compare the PDEs we used with PDEs in PDEBench Takamoto et al. (2022) and PDEArena Gupta & Brandstetter (2022). The selection of PDEs for our study was carefully curated to align with the objectives of comparing PINN methods, which differs from the approach taken in PDEBench or PDEArena. While PDEBench and PDEArena are oriented towards time-dependent PDEs, such as the compressible Naiver-Stokes and Diffusion Reaction equations, and provide extensive datasets for neural operator research, our focus was distinct. We chose a range of PDEs specifically for their relevance to PINN research, where datasets are not typically provided, emphasizing the direct application of PINNs to the PDEs themselves. We select a diverse range of PDE types and complexities from existing PINN literature. Among these, we included widely applicable and representative PDEs like the incompressible Naiver-Stokes equation and the Poisson equation (Darcy flow), which are fundamental to a multitude of disciplines. Our choice thus facilitates a more targeted and appropriate comparison of PINN methodologies, underscoring the unique aspects of our research approach.

### B.4 OVERVIEW OF METHODS

The baselines we selected could be roughly divided into several categories, i.e., loss reweighting/re-sampling, novel optimizer, novel loss functions, and novel activation/architectures. As shown in Eq 117, the general formulation of PINNs is to optimize a mixture of PDE residual loss, boundary loss, and available data loss,

$$\mathcal{L}(\theta) = \frac{w_c}{N_c} \sum_{i=1}^{N_c} ||\mathcal{F}(u_\theta(x_c^i); x_c^i)||^2 + \frac{w_b}{N_b} \sum_{i=1}^{N_b} ||\mathcal{B}(u_\theta(x_b^i); x_b^i)||^2 + \frac{w_d}{N_d} \sum_{i=1}^{N_d} ||u_\theta(x_d^i) - u(x_d^i)||^2. \tag{117}$$

Under this formulation, we could explain different variants of PINNs.

- Loss reweighting methods dynamically modify the weights $w_c, w_b, w_d$ to enable a better convergence rate. Resampling methods allocate new collocation points $x_c, x_b$ or adjust their sampling probability. These methods alleviate the imbalance between PINN optimization. Results show that they achieve remarkable results on many cases of Poisson, Heat, and Wave equations.

- Novel loss functions. It modifies the form of $\mathcal{L}(\theta)$ or adds new regularization terms for higher convergence accuracy. Results show that vPINNs are excellent at solving inverse problems.

- Novel optimizer. An example of novel optimizer is Multi-Adam which is more suitable for dealing with multiple conflict loss terms especially when they have a different scale. Results show that it works for several problems with multi-scale problems.

- Novel activations/architectures. It modifies the form of surrogate neural networks $u_\theta$ for better model capacity. We see that these modifications are effective for some problems with complex geometries and nonlinear NS equations.

## C MODEL CONFIGURATION AND HYPERPARAMETERS

### C.1 MODEL ARCHITECTURE

Our research employs a specific model structure: a Multilayer Perceptron (MLP) with 5 layers, each of which has a width of 100 neurons.

The model was trained for a total of 20,000 iterations or epochs. This number of training rounds was found to be sufficient for the model to learn the underlying patterns in the data, while also avoiding potential overfitting that might occur with too many epochs.

As for the number of collocation points, for 2-dimensional problems, we used 8192 points. These collocation points provide dense coverage of the problem space while it does not consume too much GPU memory. In addition to these, we utilized 2048 boundary/initial points.

For 3-dimensional problems, the number of collocation points and boundary/initial points were increased to 32768 and 8192, respectively. This increase corresponds to the added complexity of 3-dimensional problems, requiring a more comprehensive representation of the problem space to achieve reliable and accurate results.

## C.2 OPTIMIZATION HYPERPARAMETERS

In our primary experiment, we use Adam optimizer with momentum $(0.9, 0.999)$. We set the learning rate at 1e-3. This learning rate was selected after carefully considering the trade-off between the speed of convergence and the stability of learning, which we discussed previously. We found that this learning rate provides a good balance, enabling robust learning without the issues associated with excessively high or low rates. For vanilla PINNs, the loss weights are set to 1.

In summary, our model structure and parameters were carefully selected to balance the need for accuracy and computational efficiency, providing a fair and effective comparison in our study. Detailed ablation studies about these hyperparameters are reported in Appendix E.

## C.3 OTHER METHOD-SPECIFIC HYPERPARAMETERS

Here we present the hyperparameters of the methods we tested.

- **PINN.** There are no special hyperparameters for the baseline PINN. Please refer to the section above for the network structure and optimization hyperparameters.

- **PINN-w.** We assign larger weights to boundary conditions for PINN-w. Specifically, the weight for PDE loss is set at 1, while those for initial and boundary conditions are increased to 100. These losses are then aggregated as the target loss.

- **PINN-LRA.** We set $\alpha = 0.1$ for updating loss weights, which is the recommended value in the original paper.

- **PINN-NTK.** No special hyperparameter is needed for this method.

- **RAR.** For residual-based adaptive refinement, we add new points where the residual is greatest into the training set every 2000 epochs.

- **MultiAdam.** Although there is no manual weighting for MultiAdam, the loss grouping criteria can affect its performance. Due to time constraints, we only tuned the grouping criteria for the Wave1d-C case, where losses were divided into Dirichlet boundary losses and non-Dirichlet losses and trained for 10,000 epochs. For all other cases, we simply categorize the losses into PDE and boundary losses.

- **gPINN.** For simplicity, we assign a weight of $0.01$ to the gradient terms and a weight of $1$ to all others. However, these weights are delicate and require further fine-tuning.

## D  HIGH-LEVEL STRUCTURE OF TOOLBOX

In Figure 18, we provide a high-level overview of the usage and modules of the benchmark. We provide several encapsulated classes upon DeepXDE. Specifically, we have a PDE class for building PDE problems conveniently. Then we warp the model class by passing neural network architecture, optimizer, and custom callbacks. After that, the model is compiled by DeepXDE. Finally, we invoke the multi-GPU parallel training and evaluation framework to allocate the training tasks to different GPUs. We support convenient one-button parallel training and testing on all PDE cases using all methods. An example code snippet is shown here.

```python
import deepxde as dde
from trainer import Trainer
from src.pde import PDE1, ..., PDEn
from src.utils.callbacks import TesterCallback

trainer = Trainer('experiment-name',device)
for pde_class in [PDE1, ..., PDEn]:
    def get_model():
        pde = pde_class()
        net = dde.nn.FNN([pde.input_dim] + n_layers * [n_hidden] + [pde.output_dim])
        opt = torch.optim.Adam(net.parameters(), lr=learning_rate)
        model = pde.create_model(net)
        model.compile(opt)
        return model

    trainer.add_task(
        get_model, {'iterations': num_iterations,'callbacks': [TesterCallback()]}
    )
trainer.train_all_parallel()
```

# E  DETAILED EXPERIMENTAL RESULTS

## E.1  DETAILED RESULTS OF MAIN EXPERIMENTS.

The detailed results of the main experiments in listed in the subsection. In Table 7, we provide the mean and std of L2RE for all baselines on all PDEs. In Table 6, we provide the mean and std of L1RE for all baselines on all PDEs. In Table 9, Table 10, and Table 11, we provide the low-frequency, medium-frequency, and high-frequency Fourier errors, respectively. In Table 8, we provide the mean and std of MSE for all baselines on all PDEs. In Table 12, we provide the average runtime (seconds) for all baselines trained with 20000 epochs on all PDEs averaged by three runs. In Table E.2, we show the results of all baselines on parametric PDEs. We run all experiments on a Linux server with 20 Intel(R) Xeon(R) Silver 4210 CPUs @ 2.20GHz and eight NVIDIA GeForce RTX 2080 Ti each with 12 GB GPU memory.

Here we provide an analysis of these results. Since the results of the main experiments have been described in the main text, we won't go over them again. For different metrics of the same PDE, the best-performing methods often differ. This is because different errors reflect different mismatches between the predicted solution and the true solution.

- From the results, we can see that for most cases, methods that perform well in L2RE error also perform well in L1RE. This shows that L1RE and L2RE are generally similar. Although the absolute values differ, they can mostly be used interchangeably, or one can be chosen for calculation.

- Max error measures the worst-case error, significantly different from the average loss measured by L1RE/L2RE. From the results, we can see that hp-VPINN performs very well on this metric, followed by the adaptive activation function LAAF. PINN-LRA and PINN-NTK are optimal for some equations, but their effects are not as stable.

- Fourier error allows for the convergence of different frequency components, so it's an essential reference indicator. Since functions defined in irregular geometric areas are not suitable for calculating Fourier error, we ignored these equations. Looking at **Table 9, Table 10, and Table 11** comprehensively, for mid-low frequency functions, FBPINN is the best performing in most instances. Loss reweighting methods like PINN-LRA and ordinary PINN are better for low and high-frequency components, respectively. We speculate that reweighting the loss to some extent changes the convergence order of different function components.

- Regarding the runtime metric, hp-VPINN is the fastest in most problems. This might be due to the optimization inherent in hp-VPINN's implementation and its fewer required differentiations than vanilla PINN. All other methods introduced varying degrees of additional computational overhead compared to vanilla PINN, with some methods like gPINN even requiring about twice the computational time. We list all training and inference Flops in Table 13 and Table 16. The flops metric also shows that vanilla PINNs and hp-VPINNs are the most efficient PINN variants.

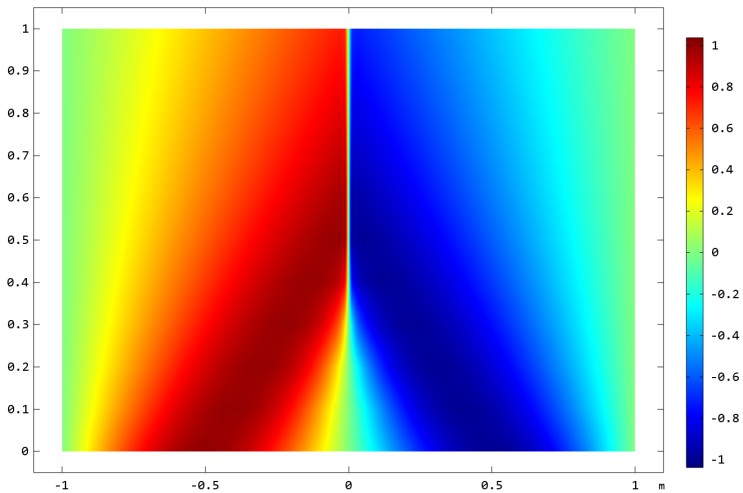

Figure 4: Reference solution of Burgers1d using FEM solver.

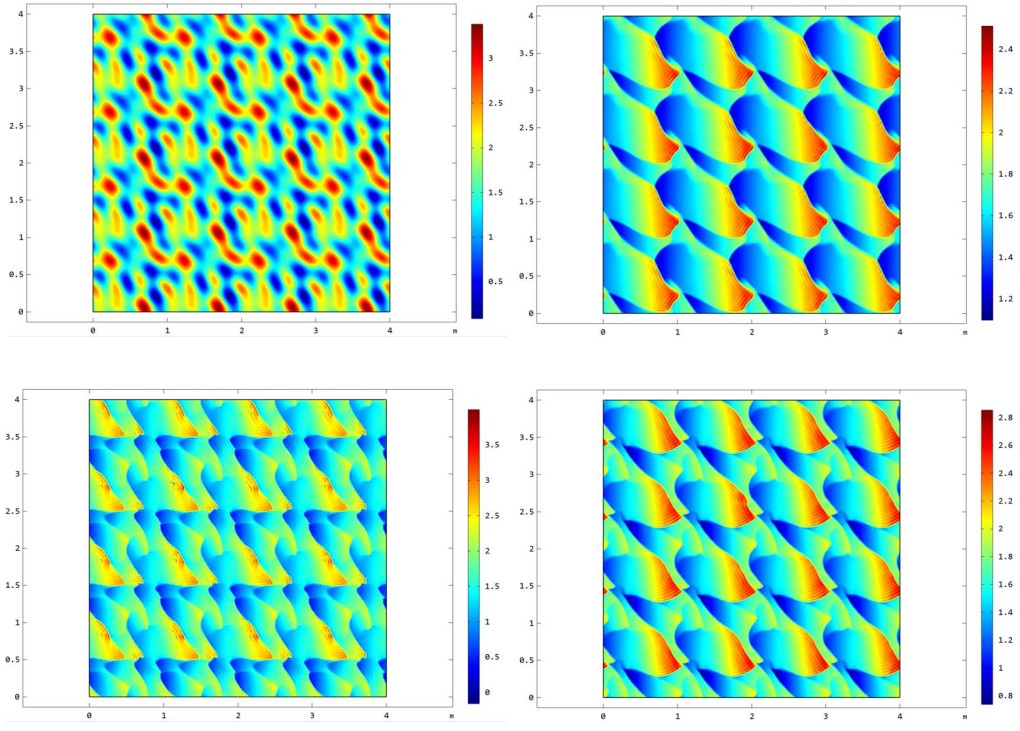

Figure 5: Reference solution of Burgers2d at timesteps $t = 0, 0.2, 0.4, 1.0$ using FEM solver.

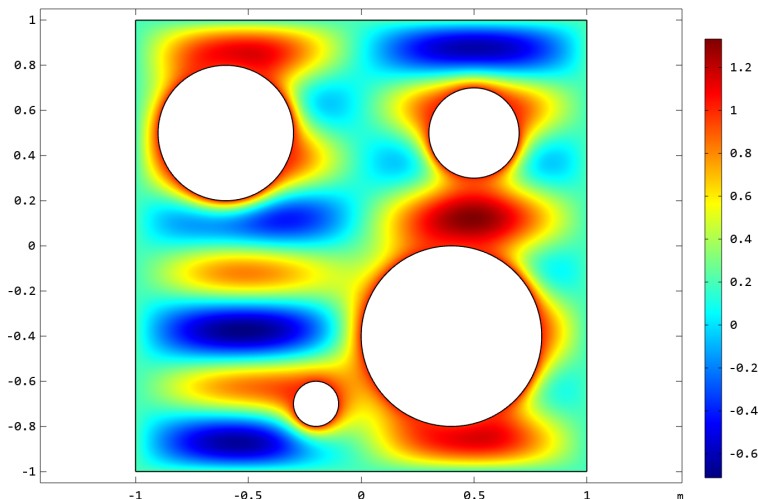

Figure 6: Reference solution of Poisson2d-CG by FEM solver.

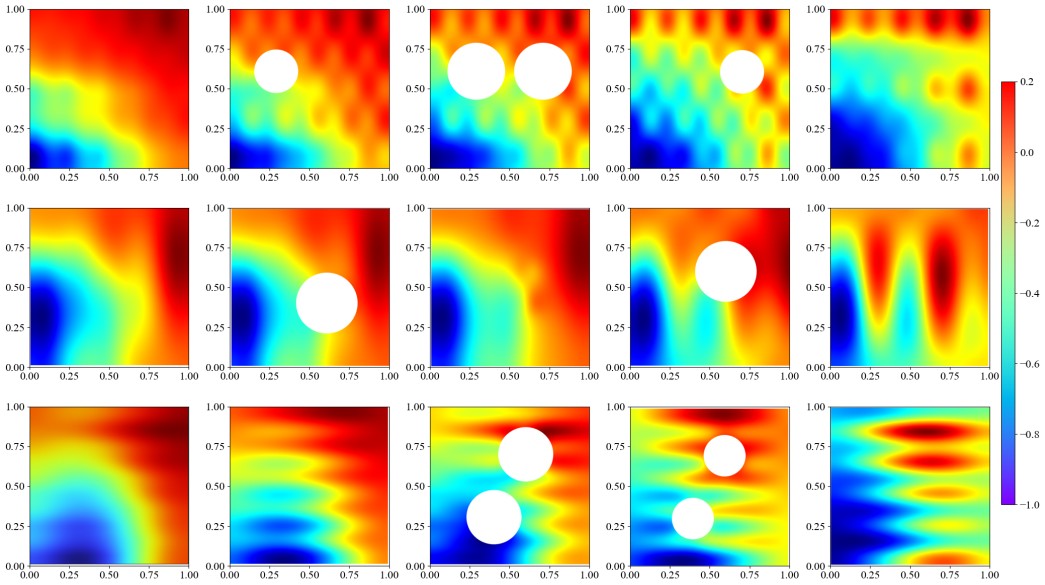

Figure 7: Reference solution of Poisson3d-CG by FEM solver. The top row displays the solution at 5 YZ planes with $x = 0, 0.25, 0.5, 0.75, 1.0$. The medium row displays it at XZ planes with $y = 0.0, 0.25, 0.5, 0.75, 1.0$. The bottom row displays it at XY planes with $z = 0.0, 0.25, 0.5, 0.75, 1.0$.

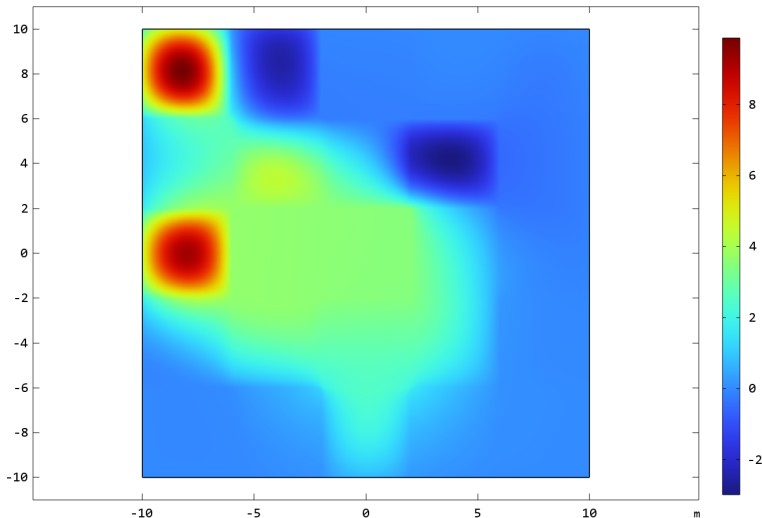

Figure 8: Reference solution of Poisson2d-MS by FEM solver.

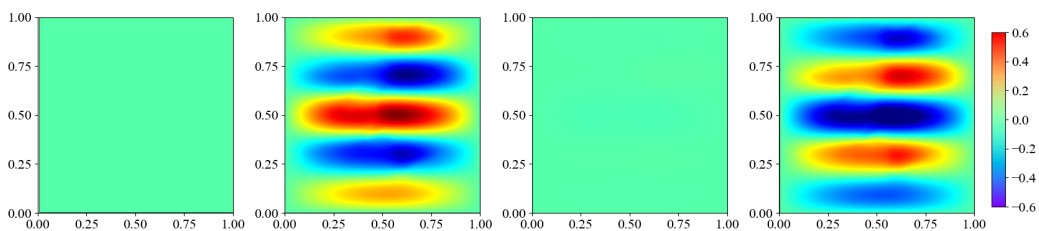

Figure 9: Reference solution of Heat2d-VC by FEM solver at timesteps $t = 0, 0.5, 2.0, 3.5$.

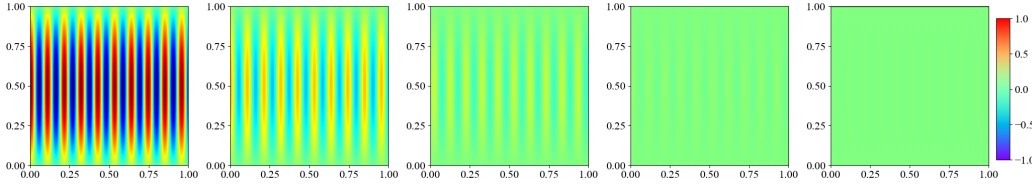

Figure 10: Reference solution of Heat2d-MS by FEM solver.

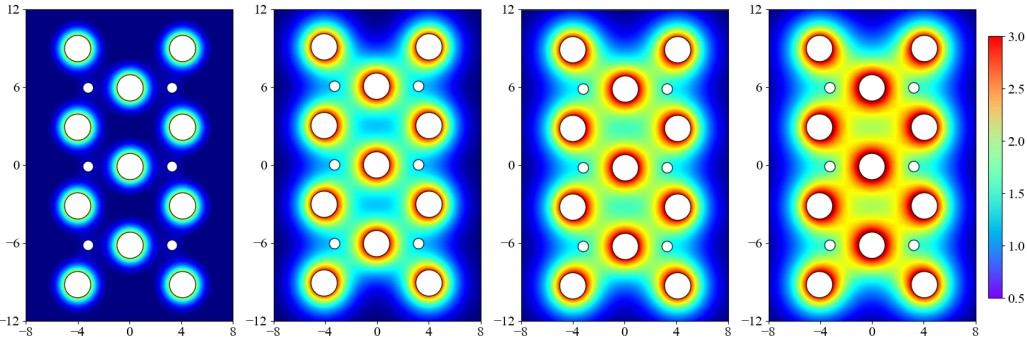

Figure 11: reference solution of Heat2d-CG by FEM solver at timesteps $t = 0.5, 2.0, 2, 5, 3.0$.

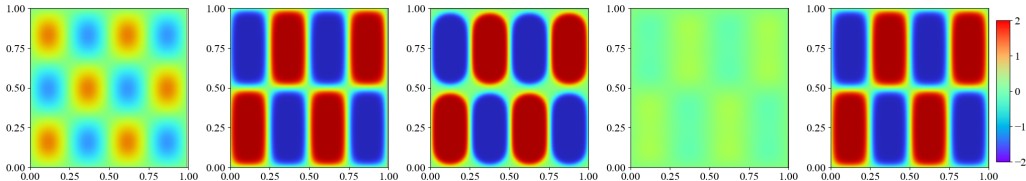

Figure 12: reference solution of Heat2d-LT by FEM solver at timesteps $t = 0, 20, 50, 80, 100$.

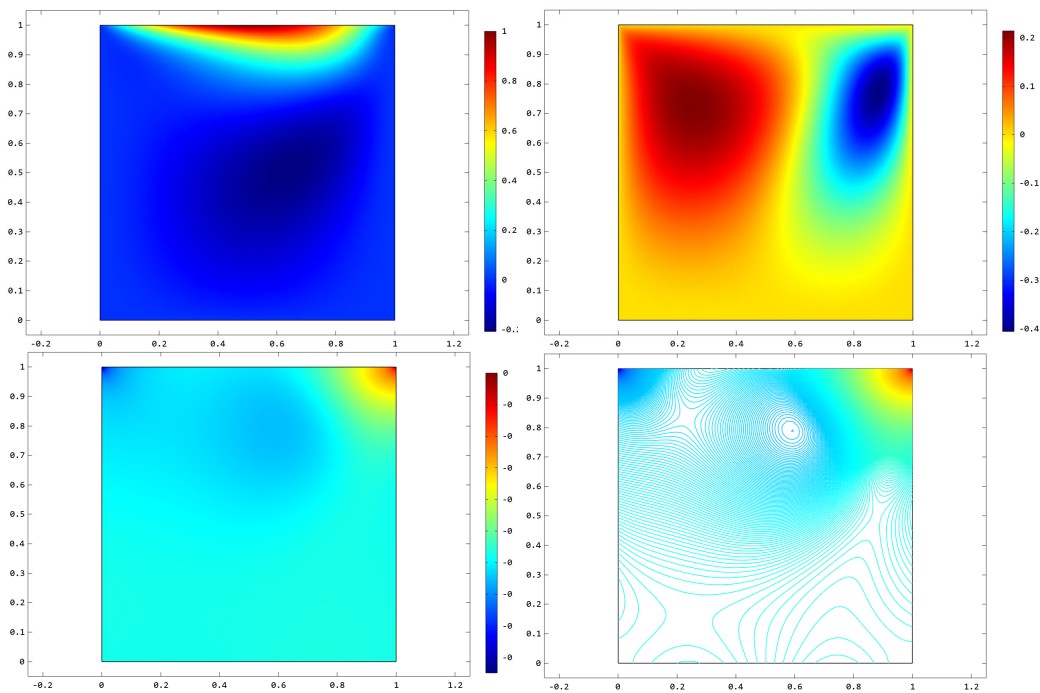

Figure 13: Reference solution of NS2d-Ld by FEM solver.

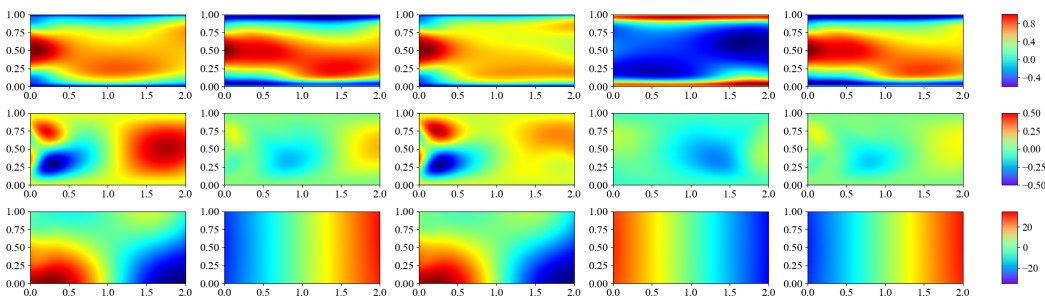

Figure 14: Reference fields $u, v, p$ from top to bottom of NS2d-LT by FEM solver at timesteps $t = 0.5, 1.0, 2.5, 4.0, 5.0$.

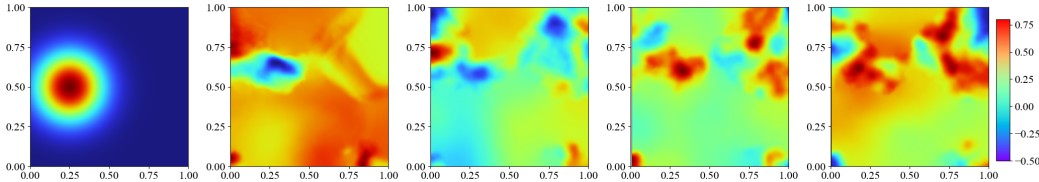

Figure 15: Reference solution of Wave2d-CG by FEM solver at timesteps $t = 0, 0.5, 2.0, 4.0, 5.0$.

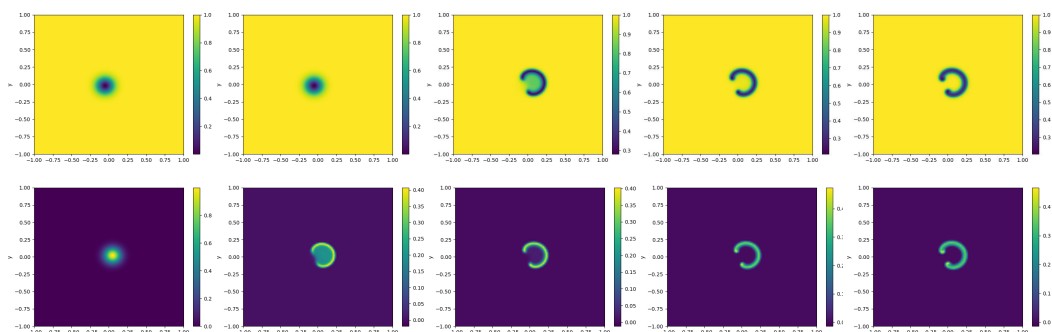

Figure 16: Reference solution of GS equation at timestep $t = 0.0, 2.5, 5.0, 7.5, 10.0$.

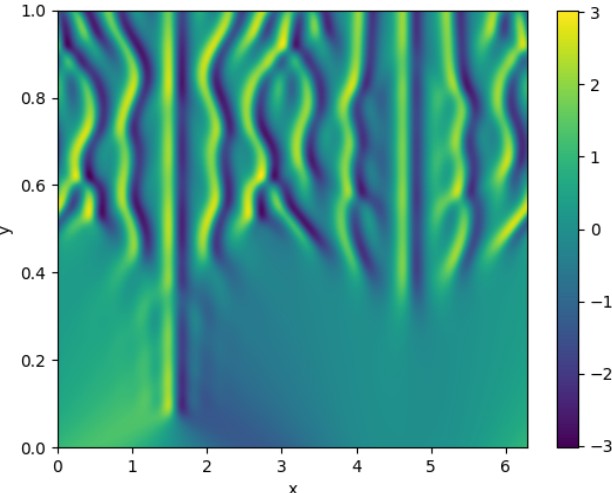

Figure 17: Reference solution of KS equation.

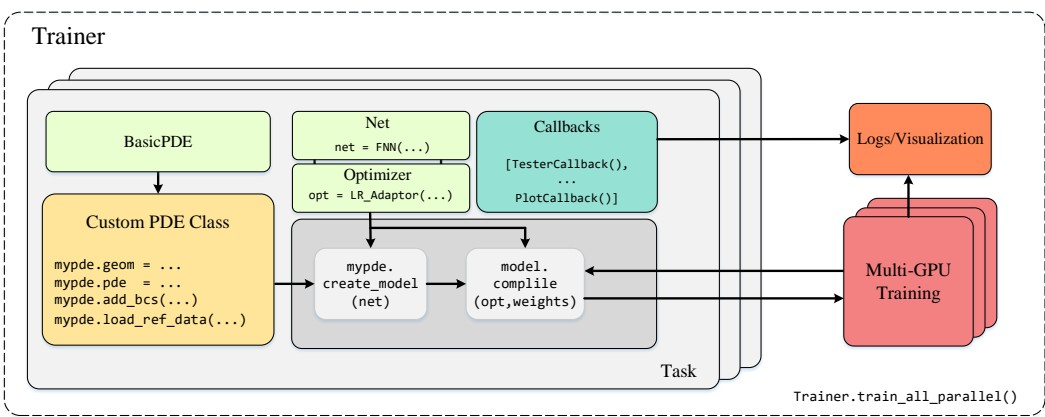

Figure 18: A high-level illustration of PINNacle code structure.

| L2RE | Name | Vanilla | | Loss Reweighting/Sampling | | | Optimizer | Loss functions | | Architecture | | |
|---|---|---|---|---|---|---|---|---|---|---|---|---|
| – | | PINN | PINN-w | LRA | NTK | RAR | MultiAdam | gPINN | vPINN | LAAF | GAAF | FBPINN |
| Burgers | 1d-C | 1.45E-2(1.59E-3) | 2.63E-2(4.68E-3) | 2.61E-2(1.18E-2) | 1.84E-2(3.66E-3) | 3.32E-2(2.14E-2) | 4.85E-2(1.61E-2) | 2.16E-1(3.34E-2) | 3.47E-1(3.49E-2) | **1.43E-2(1.44E-3)** | 5.20E-2(2.08E-2) | 2.32E-1(9.14E-2) |
| | 2d-C | 3.24E-1(7.54E-4) | 2.70E-1(3.93E-3) | **2.60E-1(5.78E-3)** | 2.75E-1(4.78E-3) | 3.45E-1(4.56E-5) | 3.33E-1(8.65E-3) | 3.27E-1(1.25E-4) | 6.38E-1(1.47E-2) | 2.77E-1(1.39E-2) | 2.95E-1(1.17E-2) | – |
| Poisson | 2d-C | 6.94E-1(8.78E-3) | 3.49E-2(6.91E-3) | 1.17E-1(1.26E-1) | **1.23E-2(7.37E-3)** | 6.99E-1(7.46E-3) | 2.63E-2(6.57E-3) | 6.87E-1(1.87E-2) | 4.91E-1(1.55E-2) | 7.68E-1(4.70E-2) | 6.04E-1(7.52E-2) | 4.49E-2(7.91E-3) |
| | 2d-CG | 6.36E-1(2.57E-3) | 6.08E-2(4.88E-3) | 4.34E-2(7.95E-3) | **1.43E-2(4.31E-3)** | 6.48E-1(7.87E-3) | 2.76E-1(1.03E-1) | 7.92E-1(4.56E-3) | 2.86E-1(2.00E-3) | 4.80E-1(1.43E-2) | 8.71E-1(2.67E-1) | 2.90E-2(3.92E-3) |
| | 3d-CG | 5.60E-1(2.84E-2) | 3.74E-1(3.23E-2) | **1.02E-1(3.16E-2)** | 9.47E-1(4.94E-4) | 5.76E-1(5.40E-2) | 3.63E-1(7.81E-2) | 4.85E-1(5.70E-2) | 7.38E-1(6.47E-4) | 5.79E-1(2.65E-2) | 5.02E-1(7.47E-2) | 7.39E-1(7.24E-2) |
| | 2d-MS | 6.30E-1(1.07E-2) | 7.60E-1(6.96E-3) | 7.94E-1(6.51E-2) | 7.48E-1(9.94E-3) | 6.44E-1(2.13E-2) | **5.90E-1(4.06E-2)** | 6.16E-1(1.74E-2) | 9.72E-1(2.23E-2) | 5.93E-1(1.18E-1) | 9.31E-1(7.12E-2) | 1.04E+0(6.13E-5) |
| Heat | 2d-VC | 1.01E+0(6.34E-2) | 2.35E-1(1.70E-2) | **2.12E-1(8.61E-4)** | 2.14E-1(5.82E-3) | 9.66E-1(1.86E-2) | 4.75E-2(8.44E-2) | 2.12E+0(5.51E-1) | 9.40E-1(1.73E-1) | 6.42E-1(6.32E-2) | 8.49E-1(1.06E-1) | 9.52E-1(2.29E-3) |
| | 2d-MS | 6.21E-2(1.38E-2) | 2.42E-1(2.67E-2) | 8.79E-2(2.56E-2) | **4.40E-2(4.81E-3)** | 7.49E-2(1.05E-2) | 2.18E-1(9.26E-2) | 1.13E-1(3.08E-3) | 9.30E-2(2.06E-2) | 7.40E-2(1.92E-2) | 9.85E-1(1.04E-1) | 8.20E-2(4.87E-3) |
| | 2d-CG | 3.64E-2(8.82E-3) | 1.45E-1(4.77E-3) | 1.25E-1(4.30E-3) | 1.16E-1(1.21E-2) | 2.72E-2(3.22E-3) | 7.12E-2(1.30E-2) | 9.38E-2(1.45E-2) | 1.67E+0(3.62E-3) | **2.39E-2(1.39E-3)** | 4.61E-1(2.63E-1) | 9.16E-2(3.29E-2) |
| | 2d-LT | 9.99E-1(1.05E-5) | 9.99E-1(8.01E-5) | 9.99E-1(7.37E-5) | 1.00E+0(2.82E-4) | 9.99E-1(1.56E-4) | 1.00E+0(3.85E-5) | 1.00E+0(9.82E-5) | 1.00E+0(0.00E+0) | 9.99E-1(4.49E-4) | 9.99E-1(2.20E-4) | 1.01E+0(1.23E-4) |
| NS | 2d-C | 4.70E-2(1.12E-3) | 1.45E-1(1.21E-2) | NaN(NaN) | 1.98E-1(2.60E-2) | 4.69E-1(1.16E-2) | 7.27E-1(1.95E-1) | 7.70E-2(2.99E-3) | 2.92E-1(8.24E-2) | **3.60E-2(3.87E-3)** | 3.79E-2(4.32E-3) | 8.45E-2(2.26E-2) |
| | 2d-CG | 1.19E-1(5.46E-3) | 3.26E-1(7.69E-3) | 3.32E-1(7.60E-3) | 2.93E-2(2.02E-2) | 3.34E-1(6.52E-4) | 4.31E-1(6.95E-2) | 1.54E-1(5.89E-3) | 9.94E-1(3.80E-3) | **8.24E-2(8.21E-3)** | 1.74E-1(7.00E-2) | 8.27E+0(3.68E-5) |
| | 2d-LT | 9.96E-1(1.19E-3) | 1.00E+0(3.34E-4) | 1.00E+0(4.05E-4) | 9.99E-1(6.04E-4) | 1.00E+0(3.35E-4) | 1.00E+0(2.19E-4) | 9.95E-1(7.19E-4) | 1.73E+0(1.00E-5) | 9.98E-1(3.42E-3) | 9.99E-1(1.10E-3) | 1.00E+0(2.07E-3) |
| Wave | 1d-C | 5.88E-2(9.63E-2) | 2.85E-1(8.97E-3) | 3.61E-1(1.95E-2) | **9.79E-2(7.72E-3)** | 5.39E-1(1.77E-2) | 1.21E-1(1.76E-2) | 5.56E-1(1.67E-2) | 8.39E-1(5.94E-2) | 4.54E-1(1.08E-2) | 6.77E-1(1.05E-1) | 5.91E-1(4.74E-2) |
| | 2d-CG | 1.84E+0(3.40E-1) | 1.66E+0(7.39E-2) | 1.48E+0(1.03E-1) | 2.16E+0(1.01E-1) | 1.15E+0(1.06E-1) | 1.09E+0(1.24E-1) | 8.14E-1(1.18E-2) | 7.99E-1(4.31E-2) | 8.19E-1(2.67E-2) | **7.94E-1(9.33E-3)** | 1.06E+0(7.54E-2) |
| | 2d-MS | 1.34E+0(2.34E-1) | 1.02E+0(1.16E-2) | 1.02E+0(1.36E-2) | 1.04E+0(3.11E-2) | 1.35E+0(2.43E-1) | 1.01E+0(5.64E-3) | 1.02E+0(4.00E-3) | 9.82E-1(1.23E-3) | 1.06E+0(1.71E-2) | 1.06E+0(5.35E-2) | 1.03E+0(6.68E-3) |
| Chaotic | GS | 3.19E-1(3.18E-1) | 1.58E-1(9.10E-2) | 9.37E-2(4.42E-5) | 2.16E-1(7.73E-2) | 9.46E-2(9.46E-4) | 9.37E-2(1.21E-5) | 2.48E-1(1.10E-1) | 1.16E+0(1.43E-1) | 9.47E-2(7.07E-5) | 9.46E-2(1.15E-4) | **7.99E-2(1.69E-2)** |
| | KS | 1.01E+0(1.28E-3) | 9.86E-1(2.24E-2) | 9.57E-1(2.85E-3) | 9.64E-1(4.94E-3) | 1.01E+0(8.63E-4) | 9.61E-1(4.77E-3) | 9.94E-1(3.83E-3) | 9.72E-1(5.80E-4) | 1.01E+0(2.12E-3) | 1.00E+0(1.24E-2) | 1.02E+0(2.31E-2) |
| High dim | PNd | 3.04E-3(5.62E-4) | 2.58E-3(1.31E-3) | **4.58E-4(1.89E-5)** | 4.64E-3(4.36E-3) | 3.59E-3(1.25E-3) | 3.98E-3(1.11E-3) | 5.05E-3(6.07E-4) | – | 4.14E-3(5.59E-4) | 7.75E-3(1.41E-3) | – |
| | HNd | 3.61E-1(4.40E-3) | 4.59E-1(4.34E-3) | 3.94E-1(1.28E-2) | 3.97E-1(1.26E-2) | 3.57E-1(3.69E-3) | **3.02E-1(4.07E-2)** | 3.17E-1(6.66E-3) | – | 5.22E-1(3.12E-3) | 5.21E-1(7.79E-4) | – |
| Inverse | PInv | 9.42E-2(1.58E-3) | 1.66E-1(5.45E-3) | 1.54E-1(3.32E-3) | 1.93E-1(1.39E-2) | 9.35E-2(1.12E-2) | 1.30E-1(1.55E-2) | 8.03E-2(2.79E-3) | **2.45E-2(1.03E-2)** | 1.30E-1(1.07E-2) | 2.54E-1(1.53E-1) | 8.44E-1(1.37E-1) |
| | HInv | 1.57E+0(7.21E-2) | 5.26E-2(3.31E-3) | **5.09E-2(4.34E-3)** | 7.52E-2(5.42E-3) | 1.52E+0(6.46E-2) | 8.04E-2(1.20E-2) | 4.84E+0(2.07E+0) | 4.56E-1(1.30E-2) | 5.59E-1(5.24E-1) | 2.12E-1(4.89E-2) | 9.27E-1(1.20E-1) |

Table 5: Mean (Std) of L2RE for main experiments.

| L1RE | Name | Vanilla | | Loss Reweighting/Sampling | | | Optimizer | Loss functions | | Architecture | | |
|---|---|---|---|---|---|---|---|---|---|---|---|---|
| – | | PINN | PINN-w | LRA | NTK | RAR | MultiAdam | gPINN | vPINN | LAAF | GAAF | FBPINN |
| Burgers | 1d-C | **9.55E-3(6.42E-4)** | 1.88E-2(4.05E-3) | 1.35E-2(2.57E-3) | 1.30E-2(1.73E-3) | 1.35E-2(4.66E-3) | 2.64E-2(5.69E-3) | 1.42E-1(1.98E-2) | 4.02E-2(6.41E-3) | 1.40E-2(3.68E-3) | 1.95E-2(8.30E-3) | 3.75E-2(9.70E-3) |
| | 2d-C | 2.96E-1(7.40E-4) | 2.43E-1(2.98E-3) | 2.31E-1(7.16E-3) | **2.48E-1(5.33E-3)** | 3.27E-1(3.73E-5) | 3.12E-1(1.15E-2) | 3.01E-1(3.55E-4) | 6.56E-1(3.01E-2) | 2.57E-1(2.06E-2) | 2.67E-1(1.22E-2) | – |
| Poisson | 2d-C | 7.40E-1(5.49E-3) | 3.08E-2(5.13E-3) | 7.82E-2(7.47E-2) | **1.30E-2(8.23E-3)** | 7.48E-1(1.01E-2) | 2.47E-2(6.38E-3) | 7.35E-1(2.08E-2) | 4.60E-2(1.39E-2) | 7.67E-1(1.36E-2) | 6.57E-1(3.99E-2) | 5.01E-2(4.71E-3) |
| | 2d-CG | 5.45E-1(4.71E-3) | 4.54E-2(6.42E-3) | 2.63E-2(5.50E-3) | **1.33E-2(4.96E-3)** | 5.60E-1(8.19E-3) | 2.46E-1(1.07E-1) | 7.31E-1(2.77E-3) | 2.45E-1(5.14E-3) | 4.04E-1(1.03E-2) | 7.09E-1(2.12E-1) | 3.21E-2(6.23E-3) |
| | 3d-CG | 4.51E-1(3.35E-2) | 3.33E-1(2.64E-2) | **7.76E-2(1.63E-2)** | 9.93E-1(2.91E-4) | 4.61E-1(4.46E-2) | 3.55E-1(7.75E-2) | 4.57E-1(5.07E-2)) | 7.96E-1(3.57E-4) | 4.60E-1(1.13E-2) | 3.82E-1(4.89E-2) | 6.91E-1(7.52E-2) |
| | 2d-MS | 7.60E-1(1.06E-2) | 7.49E-1(1.12E-2) | 7.93E-1(7.62E-2) | 7.26E-1(1.46E-2) | 7.84E-1(2.42E-2) | 6.94E-1(5.61E-2) | 7.41E-1(2.01E-2) | 9.61E-1(5.67E-2) | **6.31E-1(5.42E-2)** | 9.04E-1(1.01E-1) | 9.94E-1(9.67E-5) |
| Heat | 2d-VC | 1.12E+0(5.79E-2) | 2.41E-1(1.73E-2) | 2.07E-1(1.04E-3) | **2.03E-1(1.12E-2)** | 1.06E+0(5.13E-2) | 5.45E-1(1.07E-1) | 2.41E+0(5.27E-1) | 8.79E-1(2.57E-1) | 7.49E-1(8.54E-2) | 9.91E-1(1.37E-1) | 9.44E-1(1.75E-3) |
| | 2d-MS | 9.30E-2(2.27E-2) | 2.90E-1(2.43E-2) | 1.13E-1(3.57E-2) | 6.69E-2(8.24E-3) | 1.19E-1(2.16E-2) | 3.00E-1(1.14E-1) | 1.80E-1(1.12E-2) | 9.25E-1(3.90E-2) | 1.14E-1(4.98E-2) | 1.08E+0(2.02E-1) | **5.33E-2(3.92E-3)** |
| | 2d-CG | 3.05E-2(8.47E-3) | 1.37E-1(7.70E-3) | 1.12E-1(2.57E-3) | 1.07E-1(1.44E-2) | 2.21E-2(3.42E-3) | 5.88E-2(1.02E-2) | 8.20E-2(1.32E-2) | 3.09E+0(1.86E-2) | **1.94E-2(1.98E-3)** | 3.77E-1(2.17E-1) | 6.77E-1(3.93E-2) |
| | 2d-LT | 9.98E-1(6.00E-5) | 9.98E-1(1.42E-4) | 9.98E-1(1.47E-4) | 9.99E-1(1.01E-3) | 9.98E-1(2.28E-4) | 9.99E-1(5.69E-5) | 9.98E-1(8.62E-4) | 9.98E-1(0.00E+0) | 9.98E-1(1.27E-4) | 9.98E-1(8.58E-5) | 1.01E+0(7.75E-4) |
| NS | 2d-C | 5.08E-1(3.06E-3) | 1.84E-1(1.52E-2) | NaN | 2.44E-1(3.05E-3) | 5.54E-1(1.24E-2) | 9.86E-1(3.16E-1) | 9.43E-2(3.24E-3) | 1.98E-1(7.81E-2) | 4.42E-2(7.38E-3) | **3.78E-2(8.71E-3)** | 1.18E-1(3.10E-2) |
| | 2d-CG | 1.77E-1(1.00E-2) | 4.22E-1(8.72E-3) | 4.12E-1(6.93E-3) | 3.69E-1(2.46E-2) | 4.65E-1(4.44E-3) | 6.23E-2(8.86E-2) | 2.36E-1(1.15E-2) | 9.95E-1(3.50E-4) | **1.25E-1(1.42E-2)** | 2.40E-1(8.01E-2) | 5.92E+0(5.65E-4) |
| | 2d-LT | 9.88E-1(1.86E-3) | 9.98E-1(4.68E-4) | 9.97E-1(3.64E-4) | 9.95E-1(6.66E-4) | 1.00E+0(2.46E-4) | 9.99E-1(9.27E-4) | 9.90E-1(3.60E-4) | 1.00E+0(1.40E-4) | 9.90E-1(3.78E-3) | 9.96E-1(2.68E-3) | 1.00E+0(1.38E-3) |
| Wave | 1d-C | 5.87E-1(9.20E-2) | 2.78E-1(8.86E-3) | 3.49E-1(2.02E-2) | **9.42E-2(9.13E-3)** | 5.40E-1(1.74E-2) | 1.15E-1(1.91E-2) | 5.60E-1(1.69E-2) | 1.41E+0(1.30E-1) | 4.38E-1(1.40E-2) | 6.82E-1(1.08E-1) | 6.55E-1(4.86E-2) |
| | 2d-CG | 1.96E+0(3.83E-1) | 1.78E+0(8.89E-2) | 1.58E+0(1.15E-1) | 2.34E+0(1.14E-1) | 1.16E+0(1.16E-1) | 1.09E+0(1.54E-1) | 7.22E-1(1.63E-2) | 1.08E+0(1.25E-1) | 7.45E-1(2.15E-2) | **7.08E-1(9.13E-3)** | 1.15E+0(1.03E-1) |
| | 2d-MS | 2.04E+0(7.38E-1) | 1.10E+0(4.25E-2) | 1.08E+0(6.01E-2) | 1.13E+0(4.91E-2) | 2.08E+0(7.45E-1) | 1.07E+0(1.40E-2) | 1.11E+0(1.91E-2) | 1.05E+0(1.00E-2) | 1.17E+0(4.66E-2) | 1.12E+0(8.62E-2) | 1.29E+0(2.81E-2) |
| Chaotic | GS | 3.45E-1(4.57E-1) | 1.29E-1(1.54E-1) | 2.01E-2(5.99E-5) | 1.11E-1(4.79E-2) | 2.98E-2(6.44E-3) | **2.00E-2(6.12E-5)** | 2.72E-1(1.79E-1) | 1.04E+0(3.04E-1) | 2.07E-2(9.19E-4) | 1.16E-1(1.31E-1) | 5.06E-2(1.87E-2) |
| | KS | 9.44E-1(8.57E-4) | 8.95E-1(2.99E-2) | **8.60E-1(3.48E-3)** | 8.64E-1(3.31E-3) | 9.42E-1(8.75E-4) | 8.73E-1(8.40E-3) | 9.36E-1(6.12E-3) | 8.88E-1(9.92E-3) | 9.39E-1(3.25E-3) | 9.44E-1(9.86E-3) | 9.85E-1(3.35E-2) |
| High dim | PNd | 2.40E-3(3.44E-4) | 2.34E-3(1.27E-3) | **3.17E-4(9.16E-6)** | 4.58E-3(4.56E-3) | 2.98E-3(1.24E-3) | 3.40E-3(8.71E-4) | 4.43E-3(8.45E-4) | – | 4.33E-3(1.88E-3) | 5.72E-3(1.57E-3) | – |
| | HNd | 2.25E-1(3.87E-3) | 3.27E-1(5.13E-3) | 2.63E-1(1.30E-2) | 2.64E-1(1.59E-2) | 2.24E-1(2.56E-2) | **1.58E-1(2.71E-2)** | 1.83E-1(5.99E-3) | – | 3.42E-1(3.32E-3) | 3.40E-1(5.24E-3) | – |
| Inverse | PInv | 8.30E-2(6.88E-4) | 1.14E-1(3.56E-3) | 1.14E-1(6.95E-3) | 1.33E-1(1.01E-2) | 8.35E-2(9.53E-3) | 1.13E-1(1.64E-2) | 7.33E-2(2.49E-3) | **1.96E-2(7.75E-3)** | 8.12E-1(1.01E+0) | 2.18E-1(1.20E-1) | 8.39E-1(1.39E-1) |
| | HInv | 1.06E+0(5.39E-2) | 4.16E-2(3.18E-3) | **3.94E-2(1.52E-3)** | 5.96E-2(2.54E-3) | 1.01E+0(5.68E-2) | 6.29E-2(8.58E-3) | 3.51E+0(1.59E+0) | 4.59E-1(1.22E-3) | 3.93E-1(3.32E-1) | 1.89E-1(6.30E-2) | 8.46E-1(7.18E-2) |

Table 6: Mean (Std) of L1RE for main experiments.

| mERR | Name | Vanilla | Loss Reweighting/Sampling | | | Optimizer | Loss functions | | Architecture | | |
|---|---|---|---|---|---|---|---|---|---|---|---|
| – | | PINN | LRA | NTK | RAR | MultiAdam | gPINN | vPINN | LAAF | GAAF | FBPINN |
| Burgers | 1d-C | 9.03E-02(6.76E-03) | 1.53E-01(5.64E-02) | **1.33E-01(7.85E-02)** | 3.38E-01(2.82E-01) | 4.02E-01(2.04E-01) | 9.47E-01(1.88E-02) | 1.84E+00(1.53E-02) | 2.58E-01(2.96E-01) | 1.88E-01(6.02E-02) | 1.34E+00(4.98E-1) |
| | 2d-C | 4.32E+00(8.01E-02) | 3.84E+00(1.96E-01) | 4.07E+00(6.99E-02) | 4.47E+00(1.32E-01) | 3.99E+00(1.33E-01) | **3.83E+00(6.03E-03)** | 9.12E+00(3.83E+00) | 4.11E+00(1.99E-01) | 4.14E+00(1.11E-01) | - |
| Poisson | 2d-C | 9.41E-01(9.40E-02) | 5.93E-01(3.87E-01) | **2.94E-02(1.37E-02)** | 9.27E-01(9.67E-02) | 3.99E-01(3.67E-01) | 7.99E-01(2.80E-02) | 2.09E-01(1.30E-02) | 8.26E-01(2.78E-02) | 5.02E-01(2.37E-03) | 1.71E-1(1.52E-2) |
| | 2d-CG | 1.63E+00(1.76E-02) | 3.49E-01(2.49E-01) | **6.81E-02(3.06E-02)** | 1.65E+00(1.32E-02) | 5.84E-01(7.86E-02) | 1.67E+00(5.90E-03) | 1.62E+00(3.37E-03) | 1.61E+00(2.51E-02) | 1.49E+00(1.45E-01) | 1.98E-1(3.14E-2) |
| | 3d-CG | 1.04E+00(5.37E-02) | 3.91E-01(1.40E-01) | 1.12E+00(7.37E-04) | 1.09E+00(1.11E-01) | 6.88E-01(1.51E-01) | 7.87E-01(1.34E-01) | **1.59E-1(7.13E-5)** | 1.14E+00(3.77E-02) | 1.21E+00(2.49E-01) | 1.07E+00(2.63E-02) |
| | 2d-MS | 4.87E+00(2.10E-01) | 9.58E+00(2.05E-01) | 9.66E+00(1.86E-02) | 4.96E+00(2.90E-01) | 5.88E+00(6.69E-01) | 5.01E+00(2.61E-01) | 9.87E+00(1.20E-03) | **4.40E+00(4.58E-01)** | 8.77E+00(2.15E+00) | 9.87E+00(5.44E-4) |
| Heat | 2d-VC | 9.93E-01(7.20E-02) | **2.63E-01(8.90E-03)** | 2.67E-01(1.74E-02) | 1.03E+00(7.73E-02) | 4.73E-01(1.07E-01) | 4.46E+00(1.05E+00) | 8.83E-01(3.35E-01) | 7.79E-01(8.19E-02) | 7.85E-01(2.12E-01) | 7.78E-1(1.11E-3) |
| | 2d-MS | 9.10E-02(3.20E-02) | 1.60E-01(5.65E-02) | 6.65E-02(2.50E-02) | **4.36E-02(1.28E-02)** | 1.58E-01(8.85E-02) | 7.10E-01(3.05E-01) | 3.69E-01(1.00E-03) | 5.53E-02(1.20E-02) | 8.35E-02(4.70E-02) | 1.81E-1(4.94E-3) |
| | 2d-CG | 9.40E-01(7.48E-02) | **6.40E-01(3.70E-02)** | 1.14E+00(1.21E-01) | 9.00E-01(1.47E-01) | 1.39E+00(2.32E-01) | 2.20E+00(2.95E-01) | 4.38E+00(3.48E-01) | 9.59E-01(5.39E-02) | 3.18E+00(4.99E-01) | 2.83E+00(3.63E-1) |
| | 2d-LT | 2.18E+00(6.95E-01) | 1.82E+00(1.60E-02) | 1.83E+00(1.40E-02) | 1.85E+00(1.16E-02) | 1.82E+00(2.40E-02) | 5.46E+00(6.13E+00) | 3.09E+00(3.46E-01) | 1.84E+00(1.51E-02) | **1.81E+00(7.94E-03)** | 3.32E+00(6.15E-2) |
| NS | 2d-C | 2.26E-01(6.33E-03) | nan(nan) | 2.65E-01(3.05E-02) | 2.22E-01(1.42E-02) | 5.67E-01(6.28E-02) | 4.73E-01(3.17E-02) | **1.80E-01(1.64E-02)** | 1.84E-01(5.41E-03) | 1.99E-01(9.09E-03) | 2.00E-1(4.73E-2) |
| | 2d-CG | 2.06E-01(6.69E-03) | 4.97E-01(9.10E-02) | 3.33E-01(3.92E-02) | 2.11E-01(4.38E-03) | 6.23E-01(1.87E-01) | 2.94E-01(9.84E-03) | 4.31E+00(1.47E-02) | **1.68E-01(2.34E-03)** | 1.80E-01(7.47E-03) | 8.00E+00(0.00E+00) |
| | 2d-LT | 1.17E+02(5.00E-01) | 1.21E+02(2.00E-01) | 1.21E+02(6.51E-01) | 1.18E+02(7.69E-01) | 1.21E+02(2.40E-01) | 1.21E+02(5.69E-01) | 1.23E+02(5.54E-01) | **1.18E+02(6.76E-01)** | 1.19E+02(5.28E-01) | 1.24E+02(7.76E-1) |
| Wave | 1d-C | 9.34E-01(1.16E-01) | 5.17E-01(6.11E-02) | **2.75E-01(2.22E-02)** | 8.16E-01(6.80E-02) | 1.26E+00(1.89E-01) | 1.28E+00(6.21E-02) | 6.17E-01(5.41E-02) | 7.40E-01(7.71E-02) | 1.18E+00(3.23E-01) | 8.51E-1(1.11E-1) |
| | 2d-CG | 2.00E+00(9.89E-02) | 1.95E+00(1.26E-01) | 2.00E+00(1.80E-02) | 1.93E+00(8.80E-02) | 1.71E+00(5.74E-02) | 1.73E+00(2.81E-03) | 1.66E+00(2.19E-02) | 1.93E+00(1.48E-01) | 1.88E+00(1.13E-01) | **1.65E+00(2.44E-2)** |
| | 2d-MS | 1.44E+03(2.92E+02) | 1.95E+03(3.91E+02) | 1.74E+03(2.15E+02) | 1.30E+03(2.72E+02) | 1.05E+03(4.29E+01) | 1.09E+03(4.19E+01) | 4.43E+02(4.24E+00) | 1.80E+03(8.80E+01) | 1.45E+03(4.66E+02) | 5.59E+03(1.55E+02) |
| Chaotic | GS | 3.66E+00(1.00E-01) | 3.48E+00(8.97E-02) | 3.61E+00(6.38E-02) | 3.60E+00(6.85E-02) | 3.41E+00(1.27E-01) | 3.41E+00(3.54E-02) | 8.93E-01(6.51E-02) | 3.76E+00(5.27E-02) | 3.41E+00(1.28E-01) | **8.36E-1(8.16E-2)** |
| | KS | 9.84E-01(1.64E-03) | 9.83E-01(3.76E-04) | **8.76E-01(1.72E-01)** | 9.83E-01(7.11E-04) | 9.82E-01(1.42E-04) | 9.84E-01(4.09E-03) | 3.33E+00(7.80E-02) | 9.83E-01(6.72E-04) | 9.83E-01(3.76E-04) | 3.30E+00(4.74E-2) |
| High dim | PNd | 2.96E-02(1.57E-02) | **4.05E-03(9.49E-04)** | 4.99E-03(4.48E-03) | 2.72E-02(1.17E-02) | 3.96E-02(2.29E-02) | 3.16E-02(1.21E-02) | – | 5.90E-02(4.88E-02) | 1.76E+00(8.43E-01) | – |
| | HNd | 5.18E-02(2.21E-02) | 1.29E-01(1.94E-01) | 6.32E-02(3.49E-02) | 4.64E-02(1.59E-02) | 7.92E-03(3.01E-03) | 5.02E-02(5.95E-03) | – | **2.04E-02(1.22E-02)** | 1.27E+00(1.45E+00) | – |

Table 7: Mean (Std) of max error for main experiments.

| MSE | Name | Vanilla | | Loss Reweighting/Sampling | | | Optimizer | Loss functions | | Architecture | | |
|---|---|---|---|---|---|---|---|---|---|---|---|---|
| – | | PINN | PINN-w | LRA | NTK | RAR | MultiAdam | gPINN | vPINN | LAAF | GAAF | FBPINN |
| Burgers | 1d-C | **7.90E-5(1.78E-5)** | 2.64E-4(8.69E-5) | 3.03E-4(2.62E-4) | 1.30E-4(5.19E-5) | 5.78E-4(6.31E-4) | 9.68E-4(5.51E-4) | 1.77E-2(5.58E-3) | 5.13E-3(1.90E-3) | 1.80E-4(1.35E-4) | 3.00E-4(1.56E-4) | 1.53E-2(1.03E-2) |
| | 2d-C | 1.69E-1(7.86E-4) | 1.17E-1(3.41E-3) | **1.09E-1(4.84E-3)** | 1.22E-1(4.22E-3) | 1.92E-1(5.07E-5) | 1.79E-1(9.36E-3) | 1.72E-1(1.31E-4) | 7.08E-1(5.16E-2) | 1.26E-1(1.54E-2) | 1.41E-1(1.12E-2) | – |
| Poisson | 2d-C | 1.17E-1(2.98E-3) | 3.09E-4(1.25E-4) | 7.24E-3(9.95E-3) | **5.00E-5(5.33E-5)** | 1.19E-1(2.55E-3) | 1.79E-4(8.84E-5) | 1.15E-1(6.22E-3) | 4.86E-2(4.43E-3) | 1.39E-1(5.67E-3) | 9.38E-2(1.91E-2) | 7.89E-4(2.17E-4) |
| | 2d-CG | 1.28E-1(1.03E-3) | 1.17E-1(1.83E-4) | 6.13E-4(2.31E-4) | **6.99E-5(3.50E-5)** | 1.32E-1(3.23E-3) | 2.73E-2(1.92E-2) | 1.98E-1(2.28E-3) | 2.50E-2(3.80E-4) | 7.67E-2(2.73E-3) | 1.77E-1(8.70E-2) | 4.84E-4(9.87E-5) |
| | 3d-CG | 2.64E-2(2.67E-3) | 1.18E-2(1.97E-3) | **9.51E-4(6.51E-4)** | 7.54E-2(7.86E-5) | 2.81E-2(5.15E-3) | 1.16E-2(4.42E-3) | 2.01E-2(4.93E-3) | 4.58E-2(8.04E-5) | 2.82E-2(2.62E-3) | 2.16E-2(5.87E-3) | 4.63E-2(9.28E-3) |
| | 2d-MS | 2.67E+0(9.04E-2) | 3.90E+0(7.16E-2) | 4.28E+0(6.83E-1) | 3.77E+0(9.98E-2) | 2.80E+0(1.87E-1) | 2.36E+0(3.15E-1) | 2.56E+0(1.43E-1) | 6.09E+0(5.46E-1) | **1.83E+0(3.00E-1)** | 5.87E+0(8.72E-1) | 6.68E+0(8.23E-4) |
| Heat | 2d-VC | 4.00E-2(4.94E-3) | 2.19E-3(3.21E-4) | **1.76E-3(1.43E-5)** | 1.79E-3(9.80E-5) | 3.67E-2(1.42E-3) | 9.14E-3(3.13E-3) | 1.89E-1(9.44E-2) | 3.23E-2(2.26E-2) | 1.74E-2(4.35E-3) | 2.93E-2(7.12E-3) | 3.56E-3(1.71E-4) |
| | 2d-MS | 1.09E-4(4.94E-5) | 1.60E-3(3.35E-4) | 2.25E-4(1.22E-4) | **5.27E-5(1.18E-5)** | 1.54E-4(4.17E-5) | 1.51E-3(1.25E-3) | 3.43E-4(1.87E-5) | 2.57E-2(2.22E-3) | 1.57E-4(8.06E-5) | 3.10E-2(1.15E-2) | 2.17E-4(2.47E-5) |
| | 2d-CG | 2.09E-3(9.69E-4) | 3.15E-2(2.08E-3) | 2.32E-2(1.59E-3) | 2.02E-2(4.15E-3) | 1.12E-3(2.65E-4) | 7.79E-3(2.63E-3) | 1.34E-2(4.13E-3) | 1.16E+1(9.04E-2) | **8.53E-4(9.74E-5)** | 3.94E-1(2.71E-1) | 5.61E-1(5.96E-2) |
| | 2d-LT | 1.14E+0(2.38E-5) | **1.13E+0(1.82E-4)** | 1.14E+0(1.67E-4) | 1.14E+0(6.41E-4) | 1.14E+0(3.55E-4) | 1.14E+0(8.74E-5) | 1.14E+0(2.23E-4) | 1.14E+0(0.00E+0) | 1.14E+0(2.20E-4) | 1.14E+0(3.27E-4) | 1.16E+0(2.83E-4) |
| NS | 2d-C | 4.19E-5(2.00E-6) | 4.03E-4(6.45E-5) | NaN | 7.56E-4(1.90E-4) | 4.18E-3(2.05E-4) | 1.07E-2(5.67E-3) | 1.13E-4(8.77E-6) | 5.30E-4(3.50E-4) | **2.33E-5(4.71E-6)** | 2.67E-5(4.71E-6) | 1.37E-4(7.24E-5) |
| | 2d-CG | 6.94E-4(6.45E-5) | 5.19E-3(2.43E-4) | 5.40E-3(2.49E-4) | 4.22E-3(5.82E-4) | 5.45E-3(2.13E-5) | 9.32E-3(3.09E-3) | 1.16E-3(8.97E-5) | 1.06E+0(1.61E-2) | **3.37E-4(6.60E-5)** | 1.72E-3(1.33E-3) | 3.34E+0(2.97E-5) |
| | 2d-LT | 5.06E+2(1.21E+0) | 5.10E+2(3.40E-1) | 5.10E+2(4.13E-1) | 5.09E+2(6.15E-1) | 5.10E+2(3.42E-1) | 5.10E+2(2.23E-1) | **5.05E+2(7.30E-1)** | 5.11E+2(1.76E-2) | 5.06E+2(1.82E+0) | 5.11E+2(2.99E+0) | 5.15E+2(1.77E+0) |
| Wave | 1d-C | 1.11E-1(3.66E-2) | 2.54E-2(1.61E-3) | 4.08E-2(4.31E-3) | **3.01E-3(4.82E-4)** | 9.07E-2(6.02E-3) | 4.68E-3(1.28E-3) | 9.66E-2(5.85E-3) | 6.17E-1(1.19E-1) | 6.03E-2(2.87E-3) | 1.48E-1(4.44E-2) | 1.39E-1(1.97E-2) |
| | 2d-CG | 1.64E-1(6.13E-2) | 1.28E-1(1.13E-2) | 1.03E-1(1.46E-2) | 2.17E-1(2.05E-2) | 6.25E-2(1.17E-2) | 5.59E-2(1.29E-2) | 3.09E-2(8.98E-4) | 5.24E-2(9.01E-3) | 3.49E-2(3.38E-3) | **2.99E-2(4.68E-4)** | 5.78E-2(7.99E-3) |
| | 2d-MS | 1.30E+5(4.25E+4) | 7.35E+4(1.68E+3) | 7.34E+4(1.97E+3) | 7.69E+4(4.55E+3) | 1.33E+5(4.47E+4) | 7.15E+4(8.04E+2) | 7.27E+4(5.47E+2) | 1.13E+2(1.46E+2) | 7.91E+4(2.55E+3) | 7.98E+4(8.00E+3) | 8.95E+5(1.15E+4) |
| Chaotic | GS | 1.00E-1(1.35E-1) | 1.64E-2(1.70E-2) | **4.32E-3(4.07E-6)** | 2.59E-2(1.44E-2) | 4.40E-3(8.83E-5) | 4.32E-3(1.11E-6) | 3.62E-2(2.28E-2) | 4.00E-1(2.33E-1) | 4.32E-3(4.71E-6) | 1.69E-2(1.79E-2) | 5.16E-3(1.64E-3) |
| | KS | 1.16E+0(2.95E-3) | 1.11E+0(5.07E-2) | **1.04E+0(6.20E-3)** | 1.06E+0(1.09E-2) | 1.16E+0(1.98E-3) | 1.05E+0(1.04E-2) | 1.12E+0(8.67E-3) | 1.05E+0(2.50E-3) | 1.16E+0(4.50E-3) | 1.14E+0(2.33E-2) | 1.16E+0(5.28E-2) |
| High dim | PNd | 9.47E-5(3.47E-5) | 8.30E-5(5.53E-5) | **2.09E-6(1.69E-7)** | 4.02E-4(5.23E-4) | 1.43E-4(9.92E-5) | 1.70E-4(9.61E-5) | 2.57E-4(6.31E-5) | – | 3.03E-4(2.25E-4) | 4.80E-4(2.81E-4) | – |
| | HNd | 1.19E+1(2.92E-1) | 1.93E+1(3.65E-1) | 1.42E+1(9.23E-1) | 1.44E+1(9.14E-1) | 1.17E+1(2.41E-1) | **8.52E+0(2.34E+0)** | 9.21E+0(3.90E-1) | – | 2.49E+1(2.99E-1) | 2.50E+1(2.76E-1) | – |
| Inverse | PInv | 1.89E-3(6.31E-5) | 5.89E-3(3.88E-4) | 5.08E-3(2.18E-4) | 7.94E-3(1.16E-3) | 1.89E-3(4.49E-4) | 3.64E-3(8.28E-4) | 1.37E-3(9.45E-5) | **1.23E-4(9.50E-5)** | 6.25E-1(8.80E-1) | 1.87E-2(1.98E-2) | 3.98E+0(1.33E+0) |
| | HInv | 5.36E+0(4.86E-1) | 6.02E-3(7.71E-4) | **5.66E-3(9.88E-4)** | 1.23E-2(1.75E-3) | 5.01E+0(4.22E-1) | 1.43E-2(4.35E-3) | 6.01E+1(3.72E+1) | 8.83E-1(6.52E-2) | 1.27E+0(1.69E+0) | 1.03E-1(4.73E-2) | 2.23E+2(5.54E+1) |

Table 8: Mean (Std) of MSE for main experiments.

| fMSE-L | Name | Vanilla | Loss Reweighting/Sampling | | | Optimizer | Loss functions | | Architecture | | |
|---|---|---|---|---|---|---|---|---|---|---|---|
| – | | PINN | LRA | NTK | RAR | MultiAdam | gPINN | vPINN | LAAF | GAAF | FBPINN |
| Burgers | 1d-C | 2.21E-02(1.02E-02) | **1.46E-02(1.77E-02)** | 1.75E-01(2.76E-01) | 5.02E-01(6.21E-01) | 1.19E-01(2.00E-01) | 1.79E+00(1.99E+00) | 1.40E+01(1.06E+00) | 1.32E-01(2.50E-01) | 9.38E-02(1.47E-01) | 2.10E+00(1.50E+00) |
| | 2d-C | 4.85E+01(9.27E+00) | 8.18E+01(1.24E+01) | 8.36E+01(1.07E+01) | **4.77E+01(6.85E+00)** | 1.39E+02(5.63E+01) | 8.89E+01(3.98E+00) | 4.54E+03(5.57E+03) | 8.34E+01(7.63E+00) | 9.27E+01(7.53E+00) | – |
| Poisson | 2d-C | – | – | – | – | – | – | – | – | – | – |
| | 2d-CG | – | – | – | – | – | – | – | – | – | – |
| | 3d-CG | – | – | – | – | – | – | – | – | – | – |
| | 2d-MS | **1.74E+03(6.29E+01)** | 8.62E+03(1.10E+03) | 8.62E+03(6.08E+02) | 2.99E+03(2.59E+02) | 3.46E+03(1.93E+03) | 7.41E+03(5.99E+02) | 1.13E+04(4.70E+01) | 2.61E+03(5.60E+02) | 1.24E+04(5.71E+03) | 5.90E+03(6.03E+00) |
| Heat | 2d-VC | 4.78E+00(5.53E-01) | **3.66E-02(8.92E-03)** | 3.58E-01(2.81E-01) | 2.00E+00(1.49E+00) | 2.78E+00(3.95E+00) | 2.91E+03(1.84E+03) | 1.74E+00(1.04E+00) | 1.43E+00(1.87E+00) | 1.28E+01(2.08E+01) | 4.34E+00(2.13E-2) |
| | 2d-MS | 1.56E-01(2.33E-01) | **1.12E-01(1.76E-01)** | 1.46E+00(1.60E+00) | 3.55E-01(3.96E-01) | 3.48E-01(3.43E-01) | 1.37E+01(1.38E+01) | – | 3.50E-01(2.74E-01) | 1.11E+00(1.01E+00) | – |
| | 2d-CG | – | – | – | – | – | – | – | – | – | – |
| | 2d-LT | 3.90E+02(6.18E+02) | 2.71E+01(3.42E-02) | 2.75E+01(5.86E-01) | 2.70E+01(1.36E-01) | **2.70E+01(9.84E-02)** | 3.34E+05(6.48E+05) | 2.63E+01(7.51E-01) | 2.70E+01(1.56E-01) | 2.71E+01(3.78E-02) | 8.47E+01(7.99E-1) |
| NS | 2d-C | 4.29E-02(3.76E-02) | – | 3.74E-01(1.26E-01) | 2.35E-02(1.24E-02) | 2.20E+01(2.96E+01) | 6.97E-01(4.72E-01) | 5.38E-01(1.29E-01) | **1.34E-02(1.03E-02)** | 1.73E-02(8.42E-03) | 2.47E-2(2.70E-3) |
| | 2d-CG | – | – | – | – | – | – | – | – | – | – |
| | 2d-LT | 2.07E+05(9.61E+02) | 2.05E+05(2.37E+02) | 2.07E+05(5.52E+02) | 2.06E+05(4.53E+02) | **2.05E+05(2.78E+02)** | 2.05E+05(2.96E+02) | 4.87E+04(1.64E+02) | 2.07E+05(7.29E+02) | 2.06E+05(7.01E+02) | – |
| Wave | 1d-C | 5.38E+01(1.52E+01) | 4.81E-01(5.90E-01) | **4.65E-01(4.37E-01)** | 1.10E+02(7.79E+01) | 3.57E+02(1.97E+02) | 3.00E+02(8.28E+01) | 2.85E+01(8.07E+00) | 1.98E+01(1.48E+01) | 3.89E+02(3.79E+02) | 6.01E+01(1.46E+01) |
| | 2d-CG | 2.42E+01(1.08E+01) | 3.47E+02(2.09E+02) | 5.26E+02(5.40E+01) | 1.75E+02(1.90E+02) | 1.25E+02(9.42E+01) | 8.10E+01(6.42E+00) | 1.25E+01(4.98E+00) | 4.36E+02(4.69E+02) | 7.19E+01(6.05E+01) | **1.49E+01(3.22E+00)** |
| | 2d-MS | 3.72E+08(3.03E+08) | 8.91E+05(1.18E+06) | 7.01E+05(3.95E+05) | 3.93E+08(3.18E+08) | 2.39E+06(2.83E+06) | 1.85E+06(1.89E+06) | **1.13E+02(9.57E-01)** | 3.08E+06(2.04E+06) | 4.33E+06(8.23E+06) | 1.10E+07(1.93E+06) |
| Chaotic | GS | 1.45E+02(4.99E+00) | 1.44E+01(1.09E+01) | 7.79E+00(4.79E+00) | 2.96E+02(7.67E+01) | 6.51E+00(9.23E+00) | 5.26E+01(2.87E+01) | 2.79E+01(2.09E+01) | 2.69E+02(1.34E+01) | 4.89E+01(5.30E+01) | **3.49E+00(2.45E+00)** |
| | KS | 1.65E+01(3.09E+01) | 1.06E+00(5.77E-03) | 3.81E+02(2.09E+02) | **1.03E+00(6.13E-02)** | 1.07E+00(6.48E-03) | 1.38E+02(6.24E+00) | 1.24E+02(1.76E+01) | 1.08E+00(1.41E-02) | 1.04E+00(2.96E-02) | – |
| High dim | PNd | – | – | – | – | – | – | – | – | – | – |
| | HNd | – | – | – | – | – | – | – | – | – | – |

Table 9: Mean (Std) of low-frequency Fourier error for main experiments.

| fMSE-M | Name | Vanilla | Loss Reweighting/Sampling | | | Optimizer | Loss functions | | Architecture | | |
|---|---|---|---|---|---|---|---|---|---|---|---|
| – | | PINN | LRA | NTK | RAR | MultiAdam | gPINN | vPINN | LAAF | GAAF | FBPINN |
| Burgers | 1d-C | 1.43E-03(1.93E-04) | 2.94E-05(2.03E-05) | 9.34E-05(7.38E-05) | 7.51E-05(6.01E-05) | 6.18E-04(5.93E-04) | 6.69E-03(1.40E-03) | 3.00E+00(2.22E-01) | 3.53E-05(5.77E-05) | **1.72E-05(1.72E-05)** | 4.88E-1(3.65E-1) |
| | 2d-C | 3.28E-01(4.31E-03) | 2.23E-01(3.59E-02) | **2.11E-01(2.04E-02)** | 3.32E-01(4.83E-03) | 3.25E-01(1.15E-03) | 3.03E+00(3.63E+00) | 3.23E-01(2.30E-04) | 3.25E-01(2.89E-02) | 2.95E-01(1.71E-02) | – |
| Poisson | 2d-C | – | – | – | – | – | – | – | – | – | – |
| | 2d-CG | – | – | – | – | – | – | – | – | – | – |
| | 3d-CG | – | – | – | – | – | – | – | – | – | – |
| | 2d-MS | 6.57E+00(3.70E-02) | 1.70E+01(4.14E+00) | 1.47E+01(1.54E+00) | 1.18E+01(2.32E+00) | **5.61E+00(3.18E+00)** | 8.26E+00(5.99E-01) | 2.88E+01(2.37E-01) | 1.26E+01(3.53E+00) | 9.87E+00(2.25E+00) | 2.03E+01(2.50E-3) |
| Heat | 2d-VC | 2.75E-02(2.11E-03) | **1.46E-03(4.26E-04)** | 7.99E-03(1.17E-02) | 1.01E-01(7.00E-02) | 1.54E-02(1.23E-02) | 6.91E+01(1.07E+02) | 9.26E-03(3.06E-03) | 3.58E-02(2.86E-02) | 8.90E-01(1.27E+00) | 1.56E-2(2.62E-4) |
| | 2d-MS | 7.35E-04(7.70E-04) | 4.55E-05(5.24E-05) | 1.26E-04(3.07E-05) | **2.07E-05(1.15E-05)** | 3.13E-03(3.56E-03) | 2.65E-03(1.55E-03) | 7.57E-02(2.40E-03) | 6.19E-05(1.89E-05) | 1.05E-04(1.25E-04) | Nan |
| | 2d-CG | – | – | – | – | – | – | – | – | – | – |
| | 2d-LT | 2.11E+00(3.00E+00) | 1.59E+02(1.37E-01) | 1.59E+02(1.97E-01) | 1.59E+02(2.12E-01) | 1.59E+02(3.85E-02) | 1.59E+02(1.81E-01) | **4.57E-01(2.67E-02)** | 1.59E+02(2.20E-01) | 1.59E+02(1.36E-01) | 8.91E-1(4.84E-2) |
| NS | 2d-C | 1.72E-04(9.81E-05) | – | 3.96E-03(3.45E-03) | 1.48E-04(6.28E-05) | 1.48E-01(2.31E-01) | 5.84E-03(9.82E-04) | 2.86E-03(1.58E-03) | 1.05E-04(8.46E-05) | 4.46E-05(3.38E-05) | **2.18E-5(2.47E-6)** |
| | 2d-CG | – | – | – | – | – | – | – | – | – | – |
| | 2d-LT | **1.00E-02(9.51E-04)** | 2.63E-02(2.55E-03) | 1.49E-02(3.02E-03) | 1.06E-02(9.30E-04) | 2.52E-02(4.51E-03) | 2.12E-02(2.08E-03) | – | 1.05E-02(1.37E-03) | 1.26E-02(2.12E-03) | 4.53E+00(1.83E-2) |
| Wave | 1d-C | 1.61E-01(3.55E-02) | **7.58E-03(8.03E-03)** | 3.62E-02(8.80E-03) | 8.55E-01(3.57E-01) | 2.63E+00(2.04E+00) | 2.50E+00(9.12E-01) | 5.93E-01(6.18E-02) | 1.96E-01(1.46E-01) | 1.48E+00(9.12E-01) | 8.48E-2(2.36E-2) |
| | 2d-CG | 8.29E-02(6.50E-03) | 1.06E-03(1.01E-03) | **8.18E-04(2.57E-04)** | 8.27E-04(3.83E-04) | 3.12E-03(2.43E-03) | 1.36E-03(3.20E-04) | 4.73E-02(3.85E-03) | 4.73E-02(3.85E-03) | 1.53E-03(5.15E-04) | 1.49E-03(5.58E-04) |
| | 2d-MS | 1.47E+04(1.73E+04) | 1.31E+05(1.66E+05) | 1.78E+05(9.50E+04) | 2.39E+04(4.63E+04) | 2.39E+05(1.80E+05) | 1.82E+04(2.13E+04) | **4.75E+01(2.01E+00)** | 1.62E+05(1.72E+05) | 3.15E+05(4.71E+05) | 6.18E+04(3.16E+04) |
| Chaotic | GS | 5.39E+01(2.19E-01) | 7.94E-02(5.38E-02) | 3.37E-02(9.80E-03) | 6.27E-02(1.76E-02) | 8.88E-02(8.20E-02) | 1.72E-01(3.91E-02) | **2.36E-02(1.40E-02)** | 4.62E-02(6.99E-03) | 1.35E-01(1.17E-01) | 5.08E-2(3.92E-2) |
| | KS | 5.54E-01(1.46E-02) | **5.45E-01(2.42E-03)** | 5.60E-01(2.11E-02) | 5.47E-01(3.82E-03) | 5.46E-01(6.98E-05) | 5.48E-01(1.06E-02) | – | 5.46E-01(3.93E-04) | 5.46E-01(1.23E-03) | – |
| High dim | PNd | – | – | – | – | – | – | – | – | – | – |
| | HNd | – | – | – | – | – | – | – | – | – | – |

Table 10: Mean (Std) of medium-frequency Fourier error for main experiments.

| fMSE-H | Name | Vanilla | Loss Reweighting/Sampling | | | Optimizer | Loss functions | | Architecture | | |
|---|---|---|---|---|---|---|---|---|---|---|---|
| – | | PINN | LRA | NTK | RAR | MultiAdam | gPINN | vPINN | LAAF | GAAF | FBPINN |
| Burgers | 1d-C | **2.96E-05(9.65E-06)** | 1.85E-04(8.34E-05) | 1.48E-04(1.36E-04) | 2.39E-03(2.55E-03) | 6.16E-04(4.21E-04) | 1.09E-01(8.12E-03) | 1.40E-02(1.26E-03) | 7.15E-04(1.24E-03) | 1.93E-04(7.30E-05) | 6.87E-3(4.12E-3) |
| | 2d-C | 6.78E-02(1.23E-03) | **5.33E-02(1.37E-03)** | 5.43E-02(1.56E-03) | 6.78E-02(1.08E-03) | 7.07E-02(7.45E-04) | 6.80E-02(2.95E-04) | 2.31E-01(2.46E-01) | 5.84E-02(1.19E-03) | 5.98E-02(1.01E-03) | – |
| Poisson | 2d-C | – | – | – | – | – | – | – | – | – | – |
| | 2d-CG | – | – | – | – | – | – | – | – | – | – |
| | 3d-CG | – | – | – | – | – | – | – | – | – | – |
| | 2d-MS | **1.68E-02(4.05E-04)** | 2.07E+00(3.81E-01) | 2.23E+00(1.14E-01) | 1.89E+00(2.11E-01) | 2.30E+00(5.97E-01) | 8.62E-01(3.40E-02) | 5.44E-02(4.98E-03) | 1.27E+00(2.34E-01) | 3.07E+00(1.14E+00) | 7.17E-2(8.16E-6) |
| Heat | 2d-VC | 4.22E-04(2.39E-04) | 1.88E-03(1.07E-04) | 1.90E-03(7.60E-05) | 3.02E-02(3.49E-03) | 1.15E-02(8.98E-03) | 1.99E+00(9.14E-01) | 5.11E-04(3.47E-04) | 2.43E-02(2.66E-03) | 2.63E-02(1.52E-02) | **6.39E-5(3.77E-6)** |
| | 2d-MS | **6.81E-06(5.66E-06)** | 7.12E-05(4.00E-05) | 9.24E-05(6.51E-05) | 9.91E-05(1.35E-04) | 5.69E-04(3.34E-04) | 1.03E-02(3.68E-03) | 1.99E-03(1.45E-04) | 8.63E-05(3.67E-05) | 1.62E-04(1.36E-04) | – |
| | 2d-CG | – | – | – | – | – | – | – | – | – | – |
| | 2d-LT | 2.10E-01(1.45E-02) | 7.73E-01(2.31E-04) | 7.72E-01(2.41E-04) | 7.72E-01(9.35E-05) | 7.73E-01(2.03E-04) | 7.95E-01(4.27E-02) | 2.70E-01(2.79E-02) | 7.73E-01(1.59E-04) | 7.72E-01(8.87E-05) | **2.05E-1(4.46E-4)** |
| NS | 2d-C | 4.89E-06(1.01E-06) | – | 2.05E-04(4.41E-05) | 3.80E-06(3.71E-07) | 2.16E-03(5.85E-04) | 1.32E-03(3.12E-04) | 6.98E-06(4.41E-06) | 1.18E-06(2.56E-07) | 2.05E-06(7.23E-07) | **6.48E-8(1.75E-8)** |
| | 2d-CG | – | – | – | – | – | – | – | – | – | – |
| | 2d-LT | 1.09E+02(1.92E-01) | 1.11E+02(3.51E-02) | 1.10E+02(1.61E-01) | 1.09E+02(1.16E-01) | 1.10E+02(2.72E-01) | 1.10E+02(1.00E-01) | 4.51E+02(3.00E+00) | 1.09E+02(1.30E-01) | 1.09E+02(4.04E-01) | – |
| Wave | 1d-C | **1.25E-03(3.43E-04)** | 3.43E-02(9.35E-03) | 4.28E-03(6.00E-04) | 7.06E-02(7.51E-03) | 8.96E-02(1.06E-02) | 8.64E-02(5.08E-03) | 6.15E-04(7.67E-05) | 5.34E-02(2.92E-03) | 8.45E-02(1.85E-02) | – |
| | 2d-CG | 6.39E-03(1.09E-03) | 3.93E-02(1.01E-02) | 4.80E-02(3.17E-03) | 3.31E-02(9.96E-03) | 2.70E-02(4.96E-03) | 3.09E-02(5.50E-04) | 3.03E-03(2.29E-04) | 4.91E-02(3.19E-02) | 2.54E-02(2.53E-03) | **5.11E-3(1.86E-4)** |
| | 2d-MS | 7.61E+04(4.06E+03) | 7.51E+04(1.67E+03) | 7.90E+04(4.65E+03) | 7.62E+04(5.81E+03) | **7.35E+04(3.05E+02)** | 7.49E+04(5.94E+02) | – | 8.09E+04(2.45E+03) | 8.09E+04(6.69E+03) | – |
| Chaotic | GS | 5.30E-01(1.48E-03) | 1.04E+00(7.44E-03) | 1.05E+00(4.11E-03) | 1.12E+00(2.99E-03) | 1.04E+00(5.34E-03) | 1.10E+00(2.14E-03) | **1.70E-03(1.01E-03)** | 1.12E+00(2.14E-03) | 1.09E+00(7.71E-03) | – |
| | KS | 1.27E-03(2.94E-04) | 1.11E-03(1.32E-06) | 2.17E-03(2.09E-03) | 1.12E-03(2.36E-05) | **1.11E-03(7.08E-09)** | 7.28E-03(1.16E-03) | 4.48E-01(1.76E-03) | 1.11E-03(1.45E-07) | 1.11E-03(7.82E-07) | – |
| High dim | PNd | – | – | – | – | – | – | – | – | – | – |
| | HNd | – | – | – | – | – | – | – | – | – | – |

Table 11: Mean (Std) of high-frequency Fourier error for main experiments.

| Avg Runtime | Name | Vanilla | | Loss Reweighting/Sampling | | Optimizer | Loss functions | | Architecture | | |
|---|---|---|---|---|---|---|---|---|---|---|---|
| – | | PINN | PINN-w | LRA | NTK | MultiAdam | gPINN | vPINN | LAAF | GAAF | FBPINN |
| Burgers | 1d-C | **2.84E+2** | 2.78E+2 | 7.64E+2 | 6.70E+2 | 5.06E+2 | 6.28E+2 | 2.85E+2 | 3.61E+2 | 3.56E+2 | 1.11E+3 |
| | 2d-C | 3.11E+3 | 3.11E+3 | 1.84E+4 | 4.35E+3 | 2.72E+3 | 4.03E+3 | **8.95E+2** | 4.08E+3 | 4.07E+3 | – |
| Poisson | 2d-C | 3.39E+2 | 3.33E+2 | 9.01E+2 | 8.09E+2 | 6.13E+2 | 7.66E+2 | **3.29E+2** | 5.72E+2 | 4.20E+2 | 4.12E+3 |
| | 2d-CG | 3.69E+2 | 3.59E+2 | 9.36E+2 | 8.80E+2 | 6.57E+2 | 8.06E+2 | **3.55E+2** | 6.05E+2 | 4.34E+2 | 4.17E+3 |
| | 3d-CG | **1.45E+3** | 2.32E+3 | 4.06E+3 | 4.40E+3 | 2.41E+3 | 5.01E+3 | 1.94E+3 | 2.01E+3 | 1.68E+3 | 2.18E+3 |
| | 2d-MS | 3.83E+2 | **3.74E+2** | 7.47E+2 | 8.74E+2 | 6.67E+2 | 7.92E+2 | 1.81E+3 | 6.62E+2 | 4.57E+2 | 4.22E+3 |
| Heat | 2d-VC | **1.16E+3** | **1.16E+3** | 3.52E+3 | 1.69E+3 | 1.91E+3 | 1.34E+3 | 3.03E+3 | 1.52E+3 | 1.52E+3 | 3.92E+3 |
| | 2d-MS | **1.13E+3** | 1.14E+3 | 3.48E+3 | 1.61E+3 | 1.89E+3 | 1.30E+3 | 1.69E+3 | 1.51E+3 | 1.50E+3 | 5.84E+3 |
| | 2d-CG | **1.16E+3** | 1.17E+3 | 5.14E+3 | 1.64E+3 | 1.90E+3 | 1.31E+3 | 3.05E+3 | 1.52E+3 | 1.51E+3 | 5.28E+3 |
| | 2d-LT | **1.15E+3** | 1.18E+3 | 3.52E+3 | 1.65E+3 | 1.90E+3 | 1.32E+3 | 2.12E+3 | 1.51E+3 | 1.50E+3 | 3.93E+3 |
| NS | 2d-C | 7.52E+2 | 7.64E+2 | 2.24E+3 | 1.84E+3 | 1.25E+3 | 2.03E+3 | **5.68E+2** | 9.49E+2 | 9.43E+2 | 7.16E+3 |
| | 2d-CG | 7.56E+2 | 7.58E+2 | 3.26E+3 | 1.84E+3 | 1.22E+3 | 1.97E+3 | **6.79E+2** | 9.35E+2 | 9.31E+2 | 5.48E+3 |
| | 2d-LT | 3.05E+3 | 3.05E+3 | 2.25E+4 | 4.29E+3 | 3.73E+3 | 4.42E+3 | **1.38E+3** | 3.99E+3 | 3.99E+3 | 4.10E+3 |
| Wave | 1d-C | 3.50E+2 | 3.52E+2 | 1.12E+3 | 8.40E+2 | 2.72E+2 | 7.75E+2 | **2.22E+2** | 6.01E+2 | 4.36E+2 | 3.09E+3 |
| | 2d-CG | 1.21E+3 | 1.24E+3 | 4.50E+3 | 1.77E+3 | 2.01E+3 | 1.27E+3 | **5.99E+2** | 2.35E+3 | 1.57E+3 | 3.01E+3 |
| | 2d-MS | **2.19E+3** | **2.19E+3** | 6.76E+3 | 5.02E+3 | 4.12E+3 | 6.18E+3 | **2.11E+3** | 2.63E+3 | 2.25E+3 | 3.67E+3 |
| Chaotic | GS | 2.55E+3 | 2.55E+3 | 7.57E+3 | 3.17E+3 | 4.22E+3 | 2.59E+3 | **6.12E+2** | 3.23E+3 | 3.22E+3 | 5.47E+3 |
| | KS | 1.40E+3 | 1.40E+3 | 3.17E+3 | 3.59E+3 | 2.29E+3 | 3.83E+3 | **7.14E+2** | 1.62E+3 | 1.63E+3 | 8.83E+3 |
| High dim | PNd | **1.78E+3** | 1.83E+3 | 4.30E+3 | 4.75E+3 | 3.02E+3 | 1.91E+3 | – | 3.50E+3 | 2.33E+3 | – |
| | HNd | **2.35E+3** | 2.45E+3 | 7.42E+3 | 6.28E+3 | 4.00E+3 | 2.74E+3 | – | 3.09E+3 | 3.08E+3 | – |
| Inverse | PInv | 4.53E+2 | 4.88E+2 | 1.25E+3 | 1.71E+3 | 7.46E+2 | 1.50E+3 | **4.90E+2** | 5.75E+2 | 5.88E+2 | 3.63E+3 |
| | HInv | **1.09E+3** | 1.12E+3 | 3.39E+3 | 1.68E+3 | 1.77E+3 | 1.56E+3 | 1.86E+3 | 1.44E+3 | 1.44E+3 | 3.93E+3 |

Table 12: Average running time (seconds) for main experiments, we run all methods three times with 20000 epochs.

| Training Flops | Name | Vanilla | | Loss Reweighting/Sampling | | Optimizer | Loss functions | | Architecture | | |
|---|---|---|---|---|---|---|---|---|---|---|---|
| – | | PINN | PINN-w | LRA | NTK | MultiAdam | gPINN | vPINN | LAAF | GAAF | FBPINN |
| Burgers | 1d-C | **1.87E+11** | 1.87E+11 | 5.12E+11 | 4.29E+11 | 3.39E+11 | 4.11E+11 | 1.81E+11 | 2.22E+11 | 2.29E+11 | 7.34E+11 |
| | 2d-C | 2.72E+12 | 2.72E+12 | 1.23E+13 | 2.61E+12 | 1.82E+12 | 2.79E+12 | **6.23E+11** | 2.53E+12 | 2.73E+12 | – |
| Poisson | 2d-C | 2.55E+11 | 2.55E+11 | 6.04E+11 | 5.32E+11 | 4.15E+11 | 5.03E+11 | **2.21E+11** | 3.83E+11 | 2.81E+11 | 2.46E+12 |
| | 2d-CG | 2.37E+11 | 2.37E+11 | 6.17E+11 | 5.82E+11 | 4.4E+11 | 5.29E+11 | **2.38E+11** | 4.05E+11 | 2.91E+11 | 2.79E+12 |
| | 3d-CG | **9.03E+11** | 9.03E+11 | 2.72E+12 | 2.95E+12 | 1.61E+12 | 3.36E+12 | 1.3E+12 | 1.35E+12 | 1.13E+12 | 1.46E+12 |
| | 2d-MS | **2.75E+11** | 2.75E+11 | 5.02E+11 | 5.66E+11 | 4.47E+11 | 5.31E+11 | 1.21E+12 | 4.44E+11 | 3.06E+11 | 2.83E+12 |
| Heat | 2d-VC | **7.10E+11** | 7.10E+11 | 2.26E+12 | 1.03E+12 | 1.28E+12 | 8.98E+11 | 2.03E+12 | 1.02E+12 | 1.02E+12 | 2.63E+12 |
| | 2d-MS | **7.15E+11** | 7.15E+11 | 2.23E+12 | 1.01E+12 | 1.27E+12 | 8.71E+11 | 1.13E+12 | 1.05E+12 | 1.01E+12 | 3.71E+12 |
| | 2d-CG | **6.91E+11** | 6.91E+11 | 3.34E+12 | 1.07E+12 | 1.27E+12 | 8.78E+11 | 2.04E+12 | 1.08E+12 | 1.01E+12 | 3.54E+12 |
| | 2d-LT | **7.62E+11** | 7.62E+11 | 2.26E+12 | 1.11E+12 | 1.27E+12 | 8.84E+11 | 1.42E+12 | 1.01E+12 | 1.01E+12 | 2.63E+12 |
| NS | 2d-C | 5.05E+11 | 5.05E+11 | 1.39E+12 | 1.23E+12 | 8.38E+11 | 1.36E+12 | **3.81E+11** | 6.36E+11 | 6.32E+11 | 4.85E+12 |
| | 2d-CG | 4.85E+11 | 4.85E+11 | 2.14E+12 | 1.23E+12 | 8.17E+11 | 1.32E+12 | **4.55E+11** | 6.26E+11 | 6.24E+11 | 3.67E+12 |
| | 2d-LT | 1.87E+12 | 1.87E+12 | 1.51E+13 | 2.87E+12 | 2.5E+12 | 2.96E+12 | **9.25E+11** | 2.77E+12 | 2.67E+12 | 2.75E+12 |
| Wave | 1d-C | 2.15E+11 | 2.15E+11 | 7.51E+11 | 5.63E+11 | 1.82E+11 | 5.19E+11 | **1.49E+11** | 4.13E+11 | 2.92E+11 | 2.07E+12 |
| | 2d-CG | 7.11E+11 | 7.11E+11 | 3.02E+12 | 1.19E+12 | 1.35E+12 | 8.51E+11 | **4.01E+11** | 1.75E+12 | 1.08E+12 | 2.02E+12 |
| | 2d-MS | 1.47E+12 | 1.47E+12 | 4.53E+12 | 3.36E+12 | 2.76E+12 | 4.14E+12 | **1.41E+12** | 1.74E+12 | 1.51E+12 | 2.46E+12 |
| Chaotic | GS | 1.68E+12 | 1.68E+12 | 5.07E+12 | 2.12E+12 | 2.83E+12 | 1.74E+12 | **4.07E+11** | 2.16E+12 | 2.16E+12 | 3.66E+12 |
| | KS | 9.12E+11 | 9.12E+11 | 2.12E+12 | 2.41E+12 | 1.53E+12 | 2.57E+12 | **4.78E+11** | 1.09E+12 | 1.09E+12 | 5.92E+12 |
| High dim | PNd | **1.19E+12** | 1.19E+12 | 2.88E+12 | 3.18E+12 | 2.02E+12 | 1.28E+12 | – | 2.35E+12 | 1.56E+12 | – |
| | HNd | **1.57E+12** | 1.57E+12 | 4.97E+12 | 4.21E+12 | 2.68E+12 | 1.84E+12 | – | 2.07E+12 | 2.06E+12 | – |
| Inverse | PInv | **3.04E+11** | 3.27E+11 | 8.38E+11 | 1.15E+12 | 5.24E+11 | 1.01E+12 | 3.28E+11 | 3.85E+11 | 3.94E+11 | 2.43E+12 |
| | HInv | **7.34E+11** | 7.34E+11 | 2.27E+12 | 1.13E+12 | 1.19E+12 | 1.05E+12 | 1.25E+12 | 9.65E+11 | 9.65E+11 | 2.63E+12 |

Table 13: Average Flops every epoch for main experiments, we run all methods three times.

| L2RE | | Burgers1d | GS | Heat2d-CG | Poisson2d-C |
|------|------|-----------|-----|-----------|-------------|
| PINN | 1e-5 | 2.35E-2(1.90E-3) | 9.39E-2(3.60E-4) | 1.20E-1(2.40E-3) | 1.08E+0(1.08E-1) |
| | 1e-4 | 1.99E-2(4.30E-3) | 1.79E-1(1.20E-1) | 1.35E-1(2.00E-2) | 2.81E-2(2.44E-3) |
| | 1e-3 | 1.93E-2(4.00E-3) | **9.35E-2(2.30E-4)** | **8.51E-2(8.90E-3)** | **2.32E-2(1.52E-3)** |
| | 1e-2 | 3.79E-1(1.40E-1) | 1.91E-1(1.30E-1) | 1.73E-1(7.10E-2) | 3.26E-2(1.51E-3) |
| | decay | **1.69E-2(4.10E-3)** | 1.81E-1(1.20E-1) | 1.59E-1(2.00E-2) | 2.41E-2(9.33E-4) |
| PINN-LRA | 1e-5 | 3.44E-2(1.40E-2) | 1.79E-1(1.20E-1) | 1.18E-1(7.60E-4) | 2.91E-2(3.19E-3) |
| | 1e-4 | 2.12E-2(5.30E-3) | **9.36E-2(4.50E-4)** | 1.37E-1(8.50E-3) | 2.49E-2(3.88E-3) |
| | 1e-3 | 1.49E-2(9.60E-4) | 9.37E-2(3.63E-5) | 1.31E-1(9.60E-3) | **2.26E-2(1.93E-3)** |
| | 1e-2 | 6.23E-1(7.40E-2) | 1.29E-1(5.10E-2) | **8.99E-2(7.00E-3)** | 1.00E+0(5.62E-7) |
| | decay | **1.37E-2(5.00E-4)** | 1.81E-1(1.20E-1) | 1.19E-1(1.30E-2) | 2.61E-2(7.64E-4) |
| PINN-NTK | 1e-5 | 1.08E-1(2.70E-2) | 4.09E-1(1.20E-3) | **1.21E-1(2.80E-3)** | 1.86E-3(1.26E-4) |
| | 1e-4 | 4.72E-2(8.70E-3) | **1.96E-1(1.40E-1)** | 1.27E-1(5.30E-3) | 2.30E-3(9.48E-4) |
| | 1e-3 | 2.91E-2(7.40E-3) | 2.99E-1(1.50E-1) | 1.21E-1(9.50E-3) | 5.34E-3(1.22E-4) |
| | 1e-2 | NaN | 1.90E+0(1.63E+0) | NaN | 2.39E-1(2.00E-1) |
| | decay | **1.74E-2(2.30E-3)** | 3.06E-1(1.50E-1) | 1.48E-1(9.60E-3) | **8.24E-4(1.32E-4)** |

Table 14: Results of PINN, PINN-NTK, PINN-LRA under different learning rates or learning rate schedules.

| L2RE | | Burgers1d | GS | Heat2d-CG | Poisson2d-C |
|------|------|-----------|-----|-----------|-------------|
| PINN | 512 | 4.59E-1(8.36E-2) | 2.46E-1(1.09E-1) | 4.31E-1(6.57E-2) | 3.15E-2(4.04E-3) |
| | 2048 | 2.60E-1(2.43E-1) | 9.37E-2(2.60E-4) | 2.02E-1(1.92E-2) | 2.62E-2(2.31E-3) |
| | 8192 | 2.14E-2(1.76E-3) | 9.41E-2(6.05E-4) | 1.35E-1(1.71E-2) | **2.58E-2(6.51E-4)** |
| | 32768 | **1.44E-2(4.91E-4)** | **9.37E-2(3.89E-5)** | **3.73E-2(3.23E-3)** | 2.63E-2(2.32E-3) |
| PINN-LRA | 512 | 2.80E-1(2.02E-1) | 9.39E-2(1.66E-4) | 3.66E-1(3.86E-2) | 3.00E-2(3.16E-3) |
| | 2048 | 1.82E-1(1.85E-1) | 1.33E-1(5.57E-2) | 2.07E-1(4.96E-3) | 2.57E-2(1.78E-3) |
| | 8192 | 1.88E-2(9.45E-4) | **9.36E-2(2.14E-4)** | 1.01E-1(2.18E-2) | 2.82E-2(8.12E-4) |
| | 32768 | **1.49E-2(1.51E-3)** | 1.17E-1(3.25E-2) | **4.44E-2(1.05E-2)** | **2.49E-2(6.32E-4)** |

Table 15: Comparison of PINN and PINN-LRA's performance under different batch sizes (number of collocation points).

## E.2 ABLATION EXPERIMENTS

**Influence of learning rates.**    To understand the impact of learning rates We selected three methods, i.e., vanilla Physics-Informed Neural Networks (PINN), PINN-NTK, and PINN-LRA. We conduct experiments on four PDE problems, i.e., Burgers1d-C, GS, Heat2d-CG, and Poisson2d-C. The comparative analysis involved evaluating the performance of these methods using learning rates of 1e-5, 1e-4, 1e-3, and 1e-2, along with a step learning rate decay strategy implemented every 1000 epochs with a decay factor of 0.75. The results are shown in Table E.2. As stated in the main text, a moderate learning rate like 1e-3, 1e-4, or using a decay strategy is a good choice.

**Influence of batch size (Collocation points).**    To further understand the impact of the number of collocation points on our model's performance, we conducted an ablation study. We used four different numbers of collocation points, specifically 512, 2048, 8192, and 32768. The cases tested in this study were burgers1d, GS, Heat2d-CG, and Poisson2d, which is the same as the ablation study on learning rates. We utilized two variants of Physics-Informed Neural Networks: the vanilla PINN and the PINN-LRA. We found that using more batch size leads to a continual improvement in performance. For some cases, 8192 is a enough large batch size and the performance saturates. The conclusions and plots of this experiment are shown in the main text.

**Influence of training epochs.**    In this ablation study, we examine the impact of varying the number of training epochs on our model's performance. We selected four different values, specifically 5k, 20k, 80k, and 160k epochs. Similar to the previous study, the cases chosen for testing were burgers1d, GS, Heat2d-CG, and Poisson2d. The trend is that training more epochs leads to better performance. However, it is easier to saturate than a larger batch size.

**Influence of Adam hyperparameters.**    Here we examine the impact of varying the momentum hyperparameters in the Adam optimizer. Despite the learning rate, Adam contains two momentum

| Name | | Inference Flops |
|---|---|---|
| Burgers | 1d-C | 5.03E+4 |
| | 2d-C | 5.05E+4 |
| Poisson | 2d-C | 5.03E+4 |
| | 2d-CG | 5.03E+4 |
| | 3d-CG | 5.03E+4 |
| | 2d-MS | 5.04E+4 |
| Heat | 2d-VC | 5.03E+4 |
| | 2d-MS | 5.04E+4 |
| | 2d-CG | 5.04E+4 |
| | 2d-LT | 5.04E+4 |
| NS | 2d-C | 5.04E+4 |
| | 2d-CG | 5.05E+4 |
| | 2d-LT | 5.05E+4 |
| Wave | 1d-C | 5.06E+4 |
| | 2d-CG | 5.04E+4 |
| | 2d-MS | 5.04E+4 |
| Chaotic | GS | 5.04E+4 |
| | KS | 5.02E+4 |
| High dim | PNd | 5.06E+4 |
| | HNd | 5.06E+4 |
| Inverse | PInv | 5.04E+4 |
| | HInv | 5.05E+4 |

Table 16: Inference flops on a single collocation point for PINNs using network parameters the same with main experiments.

| L2RE | | Burgers1d-C | GS | Heat2d-CG | Poisson2d-C |
|---|---|---|---|---|---|
| PINN | 5k | 3.71E-2(1.21E-2) | 2.40E-1(1.11E-1) | 1.23E-1(3.77E-3) | 3.84E-2(1.77E-3) |
| | 20k | 1.66E-2(1.87E-3) | 1.65E-1(1.01E-1) | 8.95E-2(2.29E-2) | 2.38E-2(1.43E-3) |
| | 80k | 1.42E-2(6.63E-4) | **9.36E-2(8.41E-5)** | 9.64E-2(1.85E-2) | 1.86E-2(3.26E-3) |
| | 160k | 1.**38E-2(5.45E-4)** | 9.38E-2(5.38E-5) | **8.21E-2(7.52E-3)** | **1.48E-2(1.55E-3)** |
| PINN-LRA | 5k | 3.60E-2(8.82E-3) | 1.64E-1(9.91E-2) | 1.18E-1(2.32E-3) | 3.87E-2(3.28E-3) |
| | 20k | 1.56E-2(8.87E-4) | 1.09E-1(2.19E-2) | 9.29E-2(1.97E-2) | 2.65E-2(1.92E-3) |
| | 80k | 1.42E-2(1.23E-3) | 9.38E-2(1.63E-4) | **1.05E-1(1.48E-2)** | 1.79E-2(4.19E-4) |
| | 160k | **1.35E-2(1.84E-4)** | **9.38E-2(5.48E-4)** | 1.19E-1(2.28E-2) | **1.66E-2(3.50E-3)** |

Table 17: Performance of PINNs and PINN-LRA with different numbers of training epochs on 4 cases.

hyperparameters, i.e., $(\beta_1, \beta_2)$ for storing the approximate first and second-order momentum. In experiments, we observe that the momentum parameters not only affect the convergence speed and stability but also influence the final error. Here we list the results in Table E.2. We observe that in average $(\beta_1, \beta_2) = (0.99, 0.99)$ achieves the best results compared with others.

**Other method-specific parameters**

We chose several different method-specific hyperparameters to study their influence.

| L2RE | | burgers | GS | HeatComplex | Poisson2d |
|---|---|---|---|---|---|
| PINN | (0.9,0.999) | 1.79E-2(2.20E-3) | 2.47E-1(1.09E-1) | 7.76E-2(8.27E-3) | 2.72E-2(2.40E-3) |
| | (0.9,0.99) | 1.52E-2(1.34E-4) | 9.38E-2(5.93E-5) | 5.10E-2(7.20E-3) | 3.00E-2(6.98E-3) |
| | (0.9,0.9) | 1.68E-2(2.45E-3) | 9.38E-2(1.98E-4) | 4.56E-2(2.55E-3) | 2.81E-2(3.95E-3) |
| | (0.99,0.99) | **1.35E-2(1.03E-4)** | **9.37E-2(1.38E-5)** | **2.98E-2(5.24E-3)** | **9.18E-3(4.90E-4)** |
| PINN-NTK | (0.9,0.999) | 1.60E-2(5.50E-4) | 1.79E-1(1.20E-1) | 7.37E-2(1.59E-2) | 1.40E-2(4.06E-3) |
| | (0.9,0.99) | 1.57E-2(1.34E-4) | 9.37E-2(5.93E-5) | 6.65E-2(7.20E-3) | 1.57E-2(3.03E-3) |
| | (0.9,0.9) | 1.74E-2(1.40E-3) | 9.37E-2(2.11E-4) | 8.12E-2(3.33E-2) | 2.45E-2(3.64E-3) |
| | (0.99,0.99) | **1.35E-2(2.27E-4)** | **9.37E-2(1.54E-5)** | **3.62E-2(1.60E-3)** | **2.85E-3(1.68E-4)** |

Table 18: Performance comparison of PINN and PINN-NTK under different momentum parameters of Adam optimizer.

| $\alpha$ | Burgers1d | GS | Heat2d-CG | Poisson2d-C |
|---|---|---|---|---|
| 0.01 | 2.45E-2(1.75E-3) | 9.37E-2(4.25E-5) | **1.18E-1(4.72E-3)** | **2.51E-2(8.40E-3)** |
| 0.05 | 5.20E-2(2.14E-2) | 9.37E-2(3.48E-5) | 1.25E-1(7.62E-3) | 2.63E-2(1.10E-2) |
| 0.1 | **1.99E-2(5.61E-3)** | **9.37E-2(1.70E-5)** | 1.28E-1(4.66E-3) | 2.62E-1(3.00E-1) |
| 0.2 | 2.04E-2(3.66E-3) | 9.37E-2(1.03E-5) | 1.55E-1(3.31E-2) | 4.69E-2(1.37E-2) |
| 0.4 | 3.53E-2(2.47E-2) | 1.75E-1(1.15E-1) | 1.35E-1(7.37E-3) | 1.14E-1(1.23E-1) |
| 0.7 | 2.00E-2(3.72E-3) | 9.37E-2(4.21E-5) | 1.90E-1(4.09E-2) | 3.50E-1(2.24E-1) |

Table 19: Performance comparison of PINN-LRA with different momentum parameters.

| weight $w$ | Burgers1d | GS | Heat2d-CG | Poisson2d-C |
|---|---|---|---|---|
| 0.001 | **6.12E-2(1.36E-2)** | 1.66E-1(1.01E-1) | **4.97E-2(7.10E-4)** | **6.74E-1(1.71E-2)** |
| 0.01 | 1.95E-1(2.47E-2) | 1.79E-1(1.21E-1) | 7.78E-2(1.47E-2) | 6.89E-1(2.47E-2) |
| 0.1 | 4.93E-1(1.59E-2) | 4.61E-1(1.99E-1) | 1.34E-1(1.37E-3) | 6.92E-1(7.72E-3) |
| 1 | 5.53E-1(7.49E-2) | **9.38E-2(1.79E-5)** | 2.19E-1(9.90E-2) | 6.96E-1(4.39E-3) |

Table 20: Performance comparison of gPINN with different weights.

**Influence of momentum parameters for loss reweighting.** Here we choose the momentum update $\alpha$ from $\{0.01, 0.05, 0.2, 0.4, 0.7\}$. We see that the optimal value of $\alpha$ is problem-dependent. However, we observe that relatively small $\alpha$ achieves better performance.

**Influence of weight for gPINNs.** Here we choose the weight of gPINNs $w$ from $\{0.001, 0.01, 0.1, 1\}$. We see that the optimal value of $w$ is also problem-dependent and the property is intriguing. We observe that the performance of gPINNs is bad on Poisson2d-C for all values of $w$. We suggest that adding higher-order PDE residuals might harm the training process in some situations.

**Influence of number of grids for hp-VPINNs.** The number of points to compute integral within a domain $Q$ and number of grids $N_{\text{grid}}$ are two critical hyperparameters for hp-VPINN. Here we choose $Q$ from $\{5, 10, 15, 20\}$ for 2-dimensional problems and $\{6, 8, 10, 12\}$ for 3-dimensional problems to investigate their influence. We also take $N_{\text{grid}}$ into consideration, which varies in $\{4, 8, 16, 32\}$ for 2-dimensional problems and 3-dimensional problems. Different parameter selection is applied due to the limit of the VRAM. We can observe a consistent trend that as the $Q$ value rises, the accuracy of the model's predictions also enhanced. This is attributed to the fact that the $Q$ value dictates the number of integration points; hence, a higher value leads to more precise integration. However, for certain scenarios where hp-VPINN might not be the best fit, a surge in the $Q$ value doesn't significantly bolster the prediction accuracy. On the other hand, the choice of $N_{\text{grid}}$ exhibits a complex influence on accuracy. Generally, as the value of $N_{\text{grid}}$ increases, precision tends to improve. However, in regions where the solution has large gradients or discontinuities, a denser grid might amplify these anomalies, leading to larger errors during model training.

**Influence of the number of subdomains and overlap factors for FBPINNs.** The number of subdomains for domain decomposition and the overlap ratio $\alpha$ are two important hyperparameters for FBPINNs. The overlap ratio is chosen from $\{0.2, 0.4, 0.6, 0.8\}$.

**Results on different domain scale** Here we study the influence of domain scales. While numerical methods are usually resistant to domain scales, PINN methods are not invariant to domain scale changes. Moreover, normalizing the domain to $[0, 1]$ might be suboptimal for PINNs. Here we take the domain scale $L$ of Poisson2d-C as an example to study the performance under different

| Q | Burgers1d | Q | GS | Q | Heat2d-CG | Q | Poisson2d-C |
|---|---|---|---|---|---|---|---|
| 5 | 3.19E-01(2.91E-02) | 6 | 3.88E-01(9.73E-02) | 6 | 7.14E-01(7.14E-01) | 5 | 2.46E-01(1.62E-01) |
| 10 | 2.88E-01(6.03E-03) | 8 | 4.25E-01(1.51E-01) | 8 | 7.19E-01(4.89E-02) | 10 | 2.43E-01(1.57E-01) |
| 15 | 1.85E-01(6.97E-02) | 10 | 3.68E-01(2.04E-01) | 10 | 7.19E-01(4.75E-02) | 15 | 2.45E-01(1.61E-01) |
| 20 | 1.85E-01(4.65E-02) | 12 | 3.58E-01(2.06E-01) | 12 | 7.21E-01(4.95E-02) | 20 | 2.46E-01(2.46E-01) |

Table 21: Performance comparison of hp-VPINN with different $Q$.

| $N_{\text{grid}}$ | Burgers1d | $N_{\text{grid}}$ | GS | $N_{\text{grid}}$ | Heat2d-CG | $N_{\text{grid}}$ | Poisson2d-C |
|---|---|---|---|---|---|---|---|
| 4 | 3.67E-01(1.28E-02) | 3 | 1.93E-01(2.06E-02) | 3 | 6.91E-01(2.44E-02) | 4 | 4.95E-01(8.46E-02) |
| 8 | 2.43E-01(2.39E-03) | 4 | 3.68E-01(2.04E-01) | 4 | 7.19E-01(4.75E-02) | 8 | 4.95E-01(8.63E-02) |
| 16 | 3.66E-01(3.67E-02) | 5 | 3.59E-01(1.34E-01) | 5 | 7.22E-01(5.14E-02) | 16 | 2.86E-01(1.94E-02) |
| 32 | 4.59E-01(1.34E-02) | 6 | 2.81E-01(1.96E-01) | 6 | 7.23E-01(5.19E-02) | 32 | 2.43E-01(1.57E-01) |

Table 22: Performance comparison of hp-VPINN with different number of grids $N_{\text{grid}}$.

|  | Burgers1d |  | GS |
|---|---|---|---|
| (1,1) | 2.12E-1(1.19E-1) | (1,1,1) | 7.98E-2(3.59E-3) |
| (2,1) | 1.75E-1(7.97E-2) | (1,1,3) | 8.15E-2(1.73E-3) |
| (3,1) | 1.61E-1(9.77E-2) | (1,1,5) | 7.90E-2(1.28E-3) |
| (1,2) | 1.98E-1(7.34E-2) | (2,2,1) | 8.15E-2(3.56E-3) |
|  | Heat2d-CG |  | Poisson2d-C |
| (1,1,1) | 3.30E-1(1.04E-1) | (1,1) | 5.01E-2(2.80E-3) |
| (1,1,3) | 6.80E-1(1.18E-1) | (1,2) | 3.51E-1(1.26E-1) |
| (1,1,5) | 7.48E-3(3.39E-2) | (2,1) | 4.38E-1(5.30E-2) |
| (2,2,1) | 2.89E-1(2.30E-2) | (2,2) | 5.54E-2(1.23E-3) |

Table 23: Performance (L2RE) comparison of FBPINN with different domain decomposition types.

settings. We see that Multi-Adam is the most stable under domain scale changes and achieves the best performance when $L$ is small.

**Comparison between MultiAdam and L-BFGS** . Here we compare the new MUltiAdam optimizer for PINNs with L-BFGS, which is a frequently used optimizer in PINN variants. The L2Re result is listed in the Table E.2. We see that L-BFGS does not converge in many cases as it is unstable while MultiAdam has a better convergence property. However, L-BFGS achieves better accuracy on some of the problems like high dimensional PDEs.

**Temporal error analysis** For time-dependent problems, an important metric is the generalization ability along the time dimension. We selected Heat2d-CG, Heat2d-MS, and Wave1d-C with two different parameters (domain scale is 2 and 8) to observe how the error evolves over time. We found that the error accumulation over time varies depending on the specific PDE problem. For instance, in the case of Heat2d-CG, its final state is a relatively easy steady state, which results in a gradual reduction of error over time. On the other hand, for Heat2d-MS, the solution continuously oscillates, leading to an increasing error as time progresses. In the case of Wave1d-C, due to the periodic nature of the wave equation and the presence of a ground truth solution that is entirely zero, we observed the L2 Relative Error (L2RE) also increases with fluctuations. In summary, error accumulation in time-dependent problems remains challenging for PINNs, necessitating deeper analysis and improved optimization methods in future research.

**Runtime analysis** The runtime results for different methods are shown in Table 12. We have analyzed the results in the previous section.

## F    OTHER VISUALIZATION RESULTS AND ANALYSIS

Here we list some visualization results of these experiments. We see that Burgers1d, Poisson2d-C, Poisson2d-CG, and NS2d-C could be solved with a relatively low error. Other problems are difficult to learn, even the approximate shape of the solution. Here we only visualize two-dimensional cases,

| $\alpha$ | Burgers1d | GS | Heat2d-CG | Poisson2d-C |
|---|---|---|---|---|
| 0.2 | 9.88E-2(1.75E-2) | 8.57E-2(3.14E-3) | 1.05E+0(1.68E-1) | 5.81E-1(1.01E-3) |
| 0.4 | 9.01E-2(1.43E-2) | 8.09E-2(7.63E-4) | 7.36E-1(7.23E-2) | 2.85E-1(9.30E-2) |
| 0.6 | 1.75E-1(7.97E-2) | 7.95E-2(6.30E-4) | 6.79E-1(1.17E-1) | 5.54E-2(1.23E-3) |
| 0.8 | 1.61E-1(1.08E-1) | 8.04E-2(1.03E-3) | 6.96E-1(1.50E-1) | 4.19E-2(4.71E-3) |

Table 24: Performance (L2RE) comparison of FBPINN with different overlap ratios $\alpha$.

| Scale $L$ | Adam | MultiAdam | LRA | GePinn |
|---|---|---|---|---|
| 0.5 | 6.94E-1(1.76E-2) | **5.71E-1(6.11E-2)** | 6.93E-1(1.48E-2) | 7.06E-1(2.94E-3) |
| 1 | 6.92E-1(1.79E-2) | **3.56E-2(1.25E-2)** | 3.88E-1(2.61E-1) | 6.89E-1(1.41E-2) |
| 2 | 4.41E-1(9.57E-2) | **3.81E-2(9.38E-3)** | 1.68E-1(6.78E-2) | 6.76E-1(3.86E-2) |
| 4 | **1.77E-2(4.66E-3)** | 3.38E-2(9.71E-3) | 1.11E-1(1.43E-1) | 3.13E-2(2.85E-3) |
| 8 | 2.39E-2(7.26E-3) | 4.40E-2(3.07E-2) | 1.41E-1(7.10E-2) | **1.95E-2(6.42E-3)** |
| 16 | 1.83E-2(8.19E-3) | 3.62E-2(1.10E-2) | 9.45E-2(2.05E-2) | **1.59E-2(6.03E-3)** |

Table 25: Performance comparison of vanilla PINNs, Multi-Adam, PINN-LRA, and gPINN on Poisson2d-C different domain scales.

| L2RE | | MultiAdam | L-BFGS |
|---|---|---|---|
| Burgers | 1d-C | 4.85E-2(1.61E-2) | **1.33E-2(5.30E-5)** |
| | 2d-C | **3.33E-1(8.65E-3)** | 4.65E-1(4.69E-3) |
| Poisson | 2d-C | **2.63E-2(6.57E-3)** | NaN |
| | 2d-CG | **2.76E-1(1.03E-1)** | 2.96E-1(4.77E-1) |
| | 3d-CG | 3.64E+0(2.74E-2) | 3.51E+0(9.33E-2) |
| | 2d-MS | **5.90E-1(4.06E-2)** | 1.45E+0(4.75E-3) |
| Heat | 2d-VC | 4.75E-1(8.44E-2) | **2.32E-1(5.29E-3)** |
| | 2d-MS | 2.18E-1(9.26E-2) | **1.73E-2(4.74E-3)** |
| | 2d-CG | **7.12E-2(1.30E-2)** | 8.57E-1(6.69E-4) |
| | 2d-LT | 1.00E+0(3.85E-5) | 1.00E+0(6.69E-5) |
| NS | 2d-C | 7.27E-1(1.95E-1) | **2.14E-1(1.07E-3)** |
| | 2d-CG | **4.31E-1(6.95E-2)** | NaN |
| | 2d-LT | 1.00E+0(2.19E-4) | 9.70E-1(3.66E-4) |
| Wave | 1d-C | **1.21E-1(1.76E-2)** | NaN |
| | 2d-CG | 1.09E+0(1.24E-1) | 1.33E+0(2.34E-1) |
| | 2d-MS | 9.33E-1(1.26E-2) | NaN |
| Chaotic | GS | **9.37E-2(1.21E-5)** | NaN |
| | KS | 9.61E-1(4.77E-3) | NaN |
| High dim | PNd | 3.98E-3(1.11E-3) | **4.67E-4(7.12E-5)** |
| | HNd | 3.02E-1(4.07E-2) | **1.19E-4(4.01E-6)** |

Table 26: Mean L2RE comparison between MultiAdam and L-BFGS.

| L2RE | – | Burgers-P | Poisson-P | Heat-P | NS-P | Wave-P | High dim-P |
|---|---|---|---|---|---|---|---|
| Name | – | 2d-C | 2d-C | 2d-MS | 2d-C | 1d-C | HNd |
| Vanilla | PINN | 4.74E-1(1.93E-1) | 1.73E-1(2.40E-1) | 7.66E-3(3.61E-3) | 3.89E-1(4.40E-1) | 2.24E-1(3.03E-1) | 5.22E-1(3.56E-2) |
| Reweighting | LRA | 4.36E-1(1.99E-1) | 1.23E-1(1.56E-1) | 6.53E-3(6.12E-3) | 0.00E+0(0.00E+0) | 7.07E-2(1.14E-1) | 3.44E-1(1.81E-1) |
| | NTK | **4.13E-1(1.82E-1)** | **1.50E-1(1.86E-1)** | 9.04E-3(6.52E-3) | 4.52E-1(3.01E-1) | **1.66E-2(4.52E-3)** | 2.69E-1(1.88E-1) |
| Sampling | RAR | 4.71E-1(1.98E-1) | 1.53E-1(2.11E-1) | 8.07E-3(1.75E-3) | 3.91E-1(4.46E-1) | 2.33E-1(3.10E-1) | 5.05E-1(6.10E-2) |
| Optimizer | MultiAdam | 4.93E-1(1.94E-1) | 4.00E-1(3.20E-1) | 2.22E-3(1.55E-3) | 9.33E-1(4.32E-1) | 8.24E-2(9.22E-2) | **6.89E-1(8.46E-2)** |
| Loss functions | gPINN | 4.91E-1(2.01E-1) | 4.59E-1(4.57E-1) | 7.87E-3(2.82E-3) | 7.19E-1(2.89E-1) | 4.03E-1(3.44E-1) | 7.66E-1(3.30E-1) |
| | vPINN | 2.82E+0(1.79E+0) | 5.12E-1(2.43E-1) | – | 3.76E-1(6.90E-2) | 5.51E-1(6.09E-1) | – |
| Architecture | LAAF | 4.37E-1(1.77E-1) | 6.27E-2(4.65E-2) | **6.97E-3(5.23E-3)** | **3.63E-1(4.38E-1)** | 1.84E-1(2.91E-1) | 4.03E-1(1.27E-1) |
| | GAAF | 4.34E-1(1.85E-1) | 1.89E-2(2.54E-1) | 1.94E-1(8.63E-2) | 4.85E-1(4.09E-1) | 2.97E-1(2.38E-1) | 9.00E-1(1.68E-1) |
| | FBPINN | – | 2.46E-1(4.50E-1) | – | 3.99E-1(2.97E-1) | 2.87E-2(2.81E-2) | 1.15E+0(1.06E+0) |

Table 27: L2RE (mean/std) of different methods on parametric experiments.

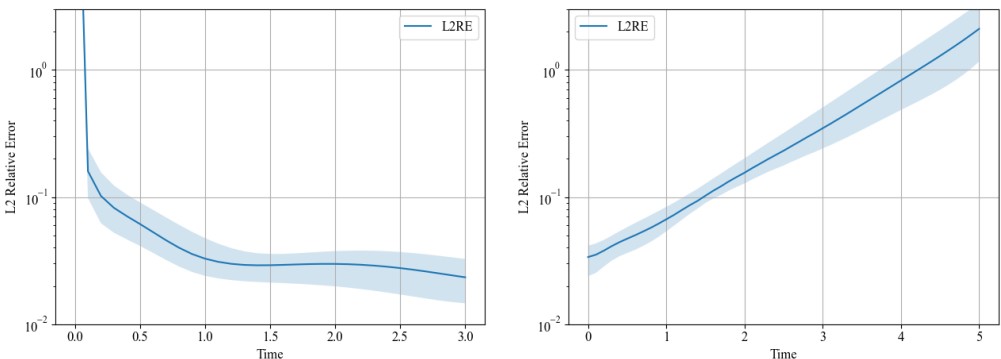

Figure 19: L2RE varying with time for PINNs on Heat2d-CG, Heat2d-MS.

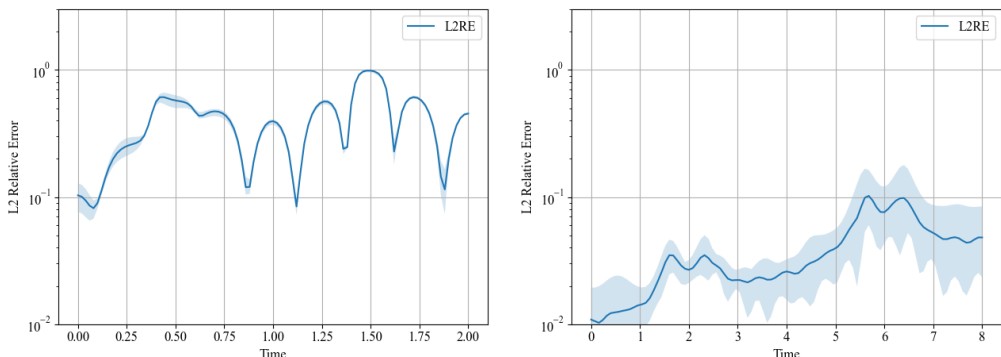

Figure 20: L2RE varying with time for PINNs on Wave1d-C-scale2 and Wave1d-C-Scale8.

which are easier to display in the paper. Note that we also support different forms of three-dimensional plot functionals in our code.

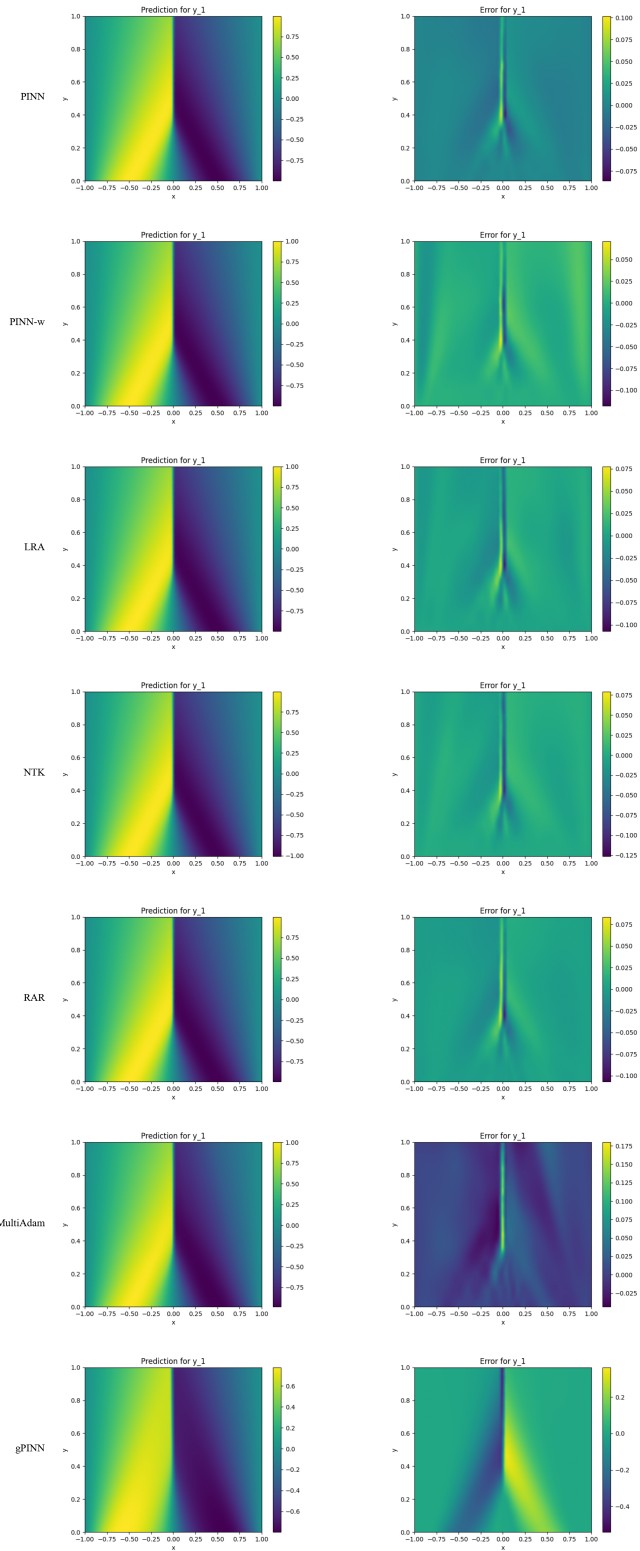

Figure 21: Visualization of Burgers1d. The left pictures are the prediction of PINN methods. The right pictures show the error between the prediction and the ground truth.

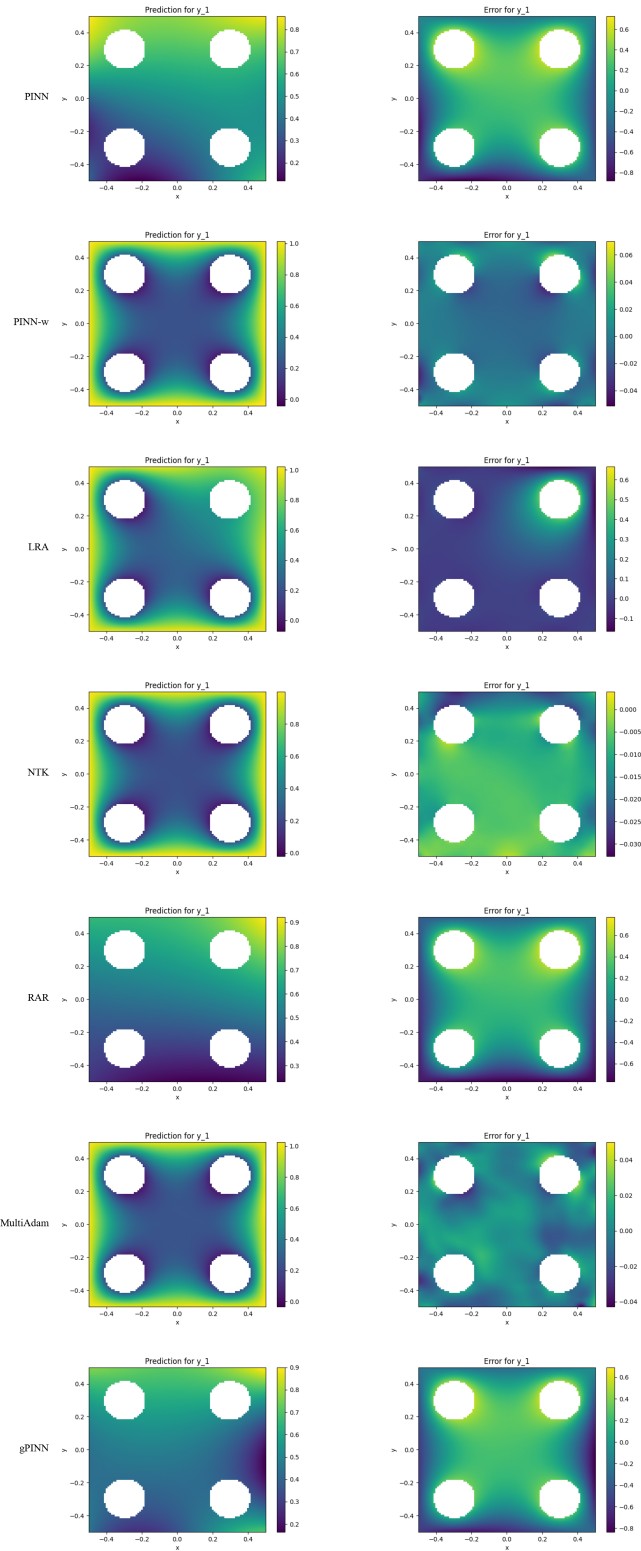

Figure 22: Visualization of Poisson2d-C. The left pictures are the prediction of PINN methods. The right pictures show the error between the prediction and the ground truth.

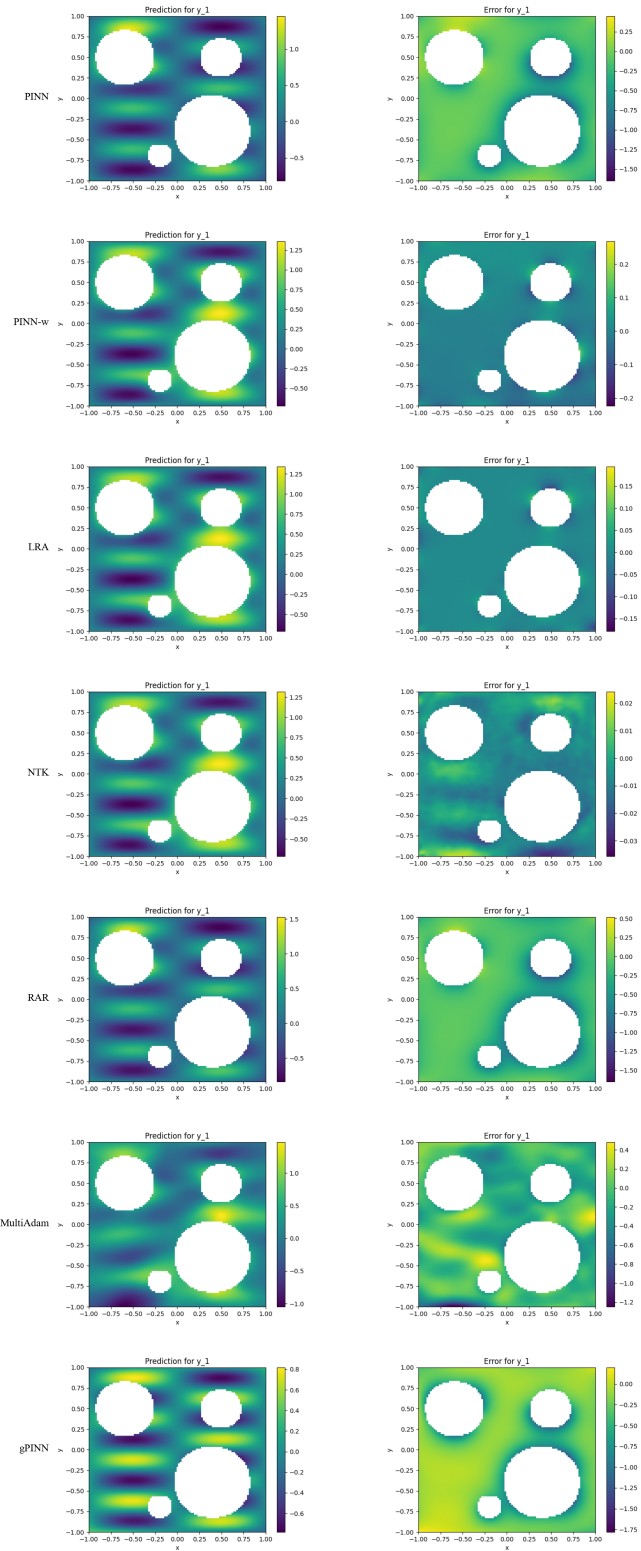

Figure 23: Visualization of Poisson2d-CG. The left pictures are the prediction of PINN methods. The right pictures show the error between the prediction and the ground truth.

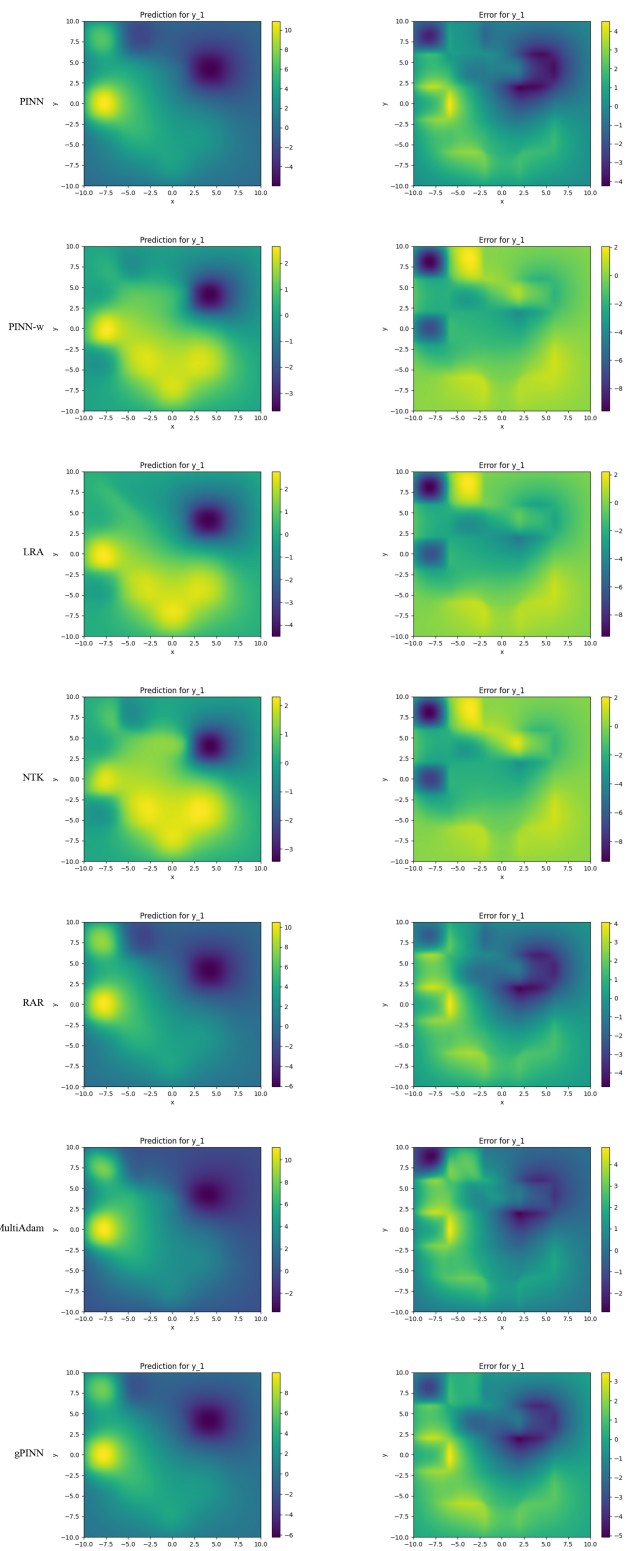

Figure 24: Visualization of Poisson2d-MS. The left pictures are the prediction of PINN methods. The right pictures show the error between the prediction and the ground truth.

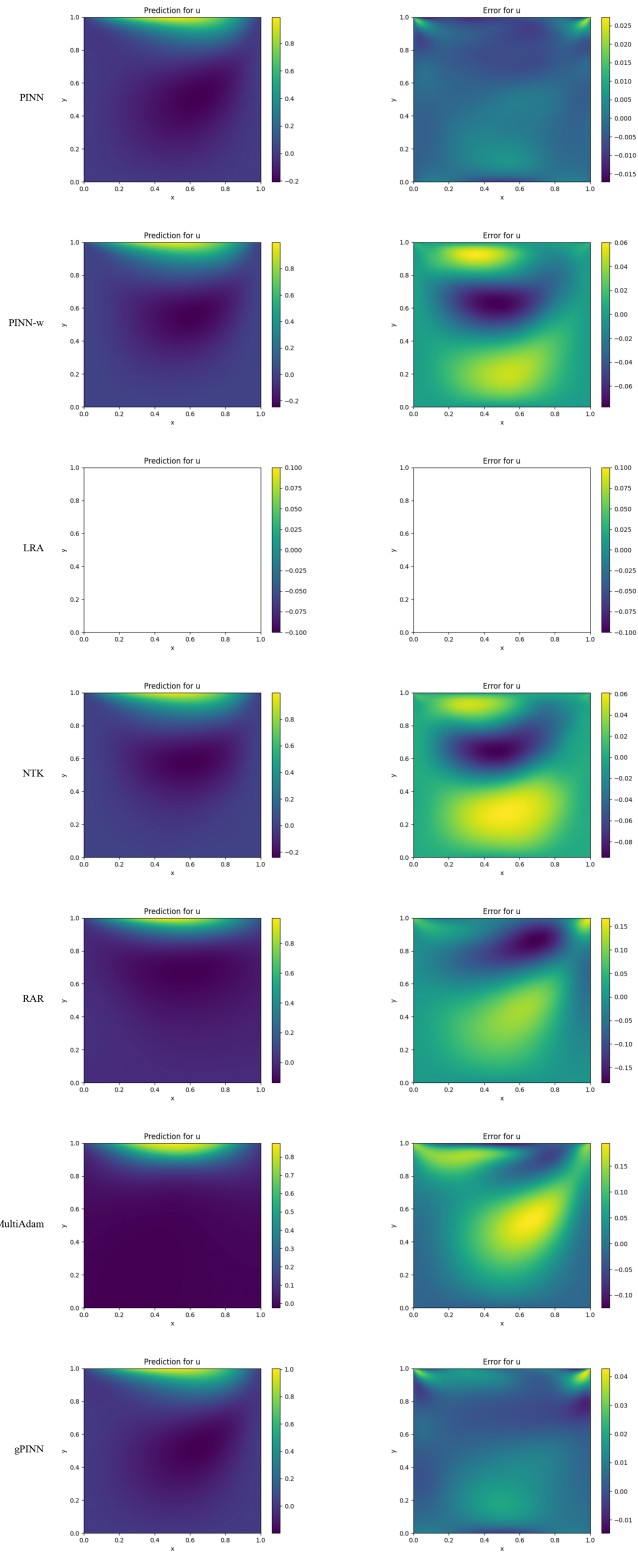

Figure 25: Visualization of NS2d-C. The left pictures are the prediction of PINN methods. The right pictures show the error between the prediction and the ground truth. Note that PINN-LRA diverged in this case.

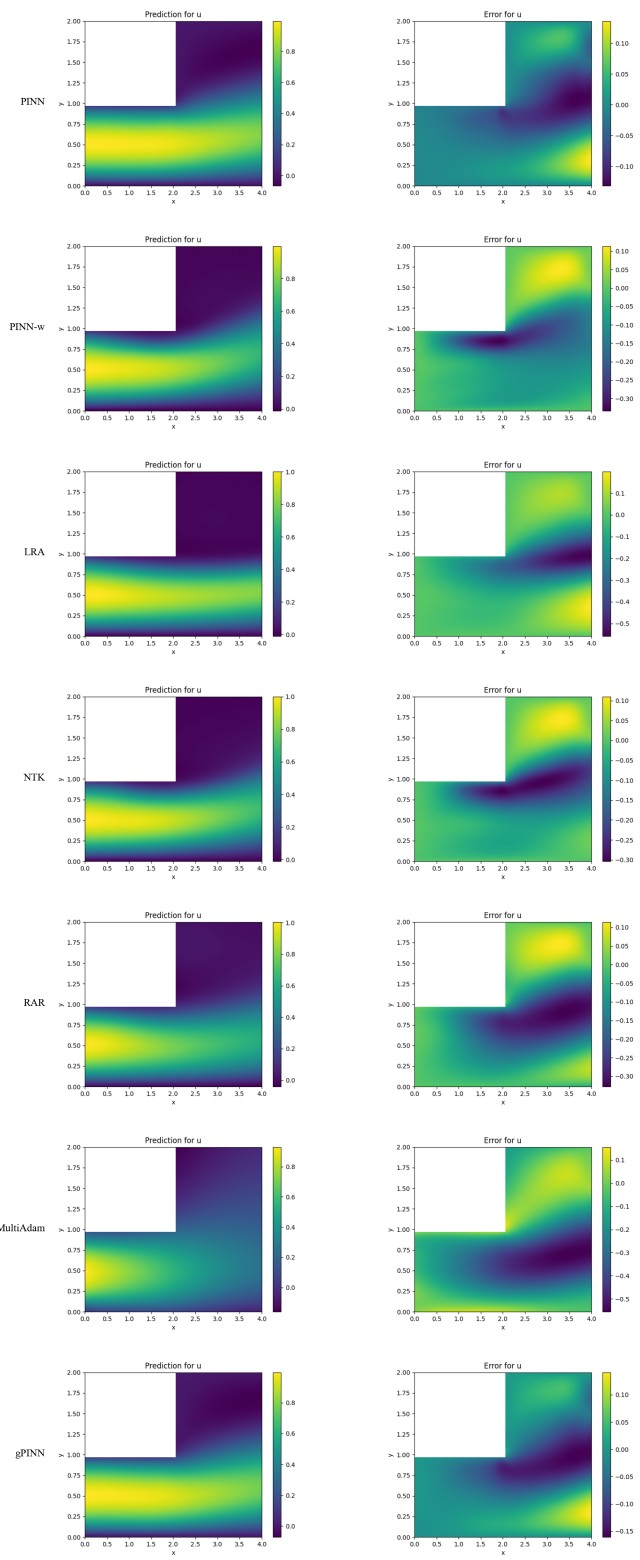

Figure 26: Visualization of NS2d-CG. The left pictures are the prediction of PINN methods. The right pictures show the error between the prediction and the ground truth.

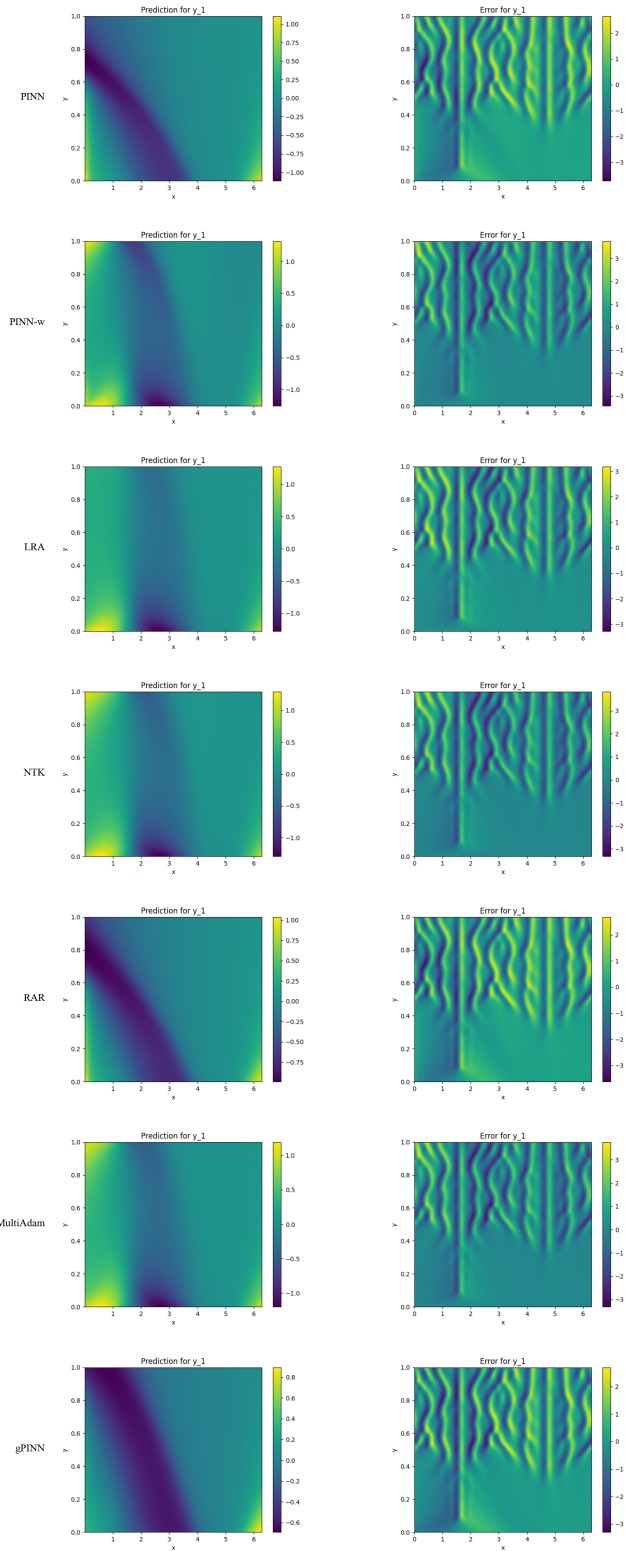

Figure 27: Visualization of KS. The left pictures are the prediction of PINN methods. The right pictures show the error between the prediction and the ground truth.